# Short-range interactions between fibrocytes and CD8+ T cells in COPD bronchial inflammatory response

Edmée Eyraud[1,2], Elise Maurat[1,2], Jean-Marc Sac-Epée[3], Pauline Henrot[1,2,4], Maeva Zysman[1,2,4], Pauline Esteves[1,2], Thomas Trian[1,2], Jean-William Dupuy[1], Alexander Leipold[4], Antoine-Emmanuel Saliba[4], Hugues Begueret[1,2,5], Pierre-Olivier Girodet[1,2,5], Matthieu Thumerel[1,2,5], Romain Hustache-Castaing[1,2,5], Roger Marthan[1,2,5], Florian Levet[6,7], Pierre Vallois[3], Cécile Contin-Bordes[8,9], Patrick Berger[1,2,5], Isabelle Dupin[2]*

[1]Univ-Bordeaux, Centre de Recherche Cardio-thoracique de Bordeaux, U1045, Département de Pharmacologie, CIC1401, Proteomics Facility, Pessac, France; [2]INSERM, Centre de Recherche Cardio-thoracique de Bordeaux, U1045, Pessac, France; [3]Univ-Lorraine, Institut Elie Cartan de Lorraine, Vandoeuvre-lès-Nancy, France; [4]Helmholtz Institute for RNA-based Infection Research (HIRI), Helmholtz-Center for Infection Research (HZI), Würzburg, Germany; [5]CHU de Bordeaux, Service d'exploration fonctionnelle respiratoire, Pessac, France; [6]Univ. Bordeaux, CNRS, INSERM, Bordeaux Imaging Center, Bordeaux, France; [7]Univ. Bordeaux, CNRS, Interdisciplinary Institute for Neuroscience, Bordeaux, France; [8]CNRS, UMR5164 ImmunoConcEpT, Université de Bordeaux, Bordeaux, France; [9]CHU de Bordeaux, Laboratoire d'Immunologie et Immunogénétique, Bordeaux, France

*For correspondence: isabelle.dupin@u-bordeaux.fr

**Abstract** Bronchi of chronic obstructive pulmonary disease (COPD) are the site of extensive cell infiltration, allowing persistent contact between resident cells and immune cells. Tissue fibrocytes interaction with CD8+ T cells and its consequences were investigated using a combination of *in situ*, *in vitro* experiments and mathematical modeling. We show that fibrocytes and CD8+ T cells are found in the vicinity of distal airways and that potential interactions are more frequent in tissues from COPD patients compared to those of control subjects. Increased proximity and clusterization between CD8+ T cells and fibrocytes are associated with altered lung function. Tissular CD8+ T cells from COPD patients promote fibrocyte chemotaxis via the CXCL8-CXCR1/2 axis. Live imaging shows that CD8+ T cells establish short-term interactions with fibrocytes, that trigger CD8+ T cell proliferation in a CD54- and CD86-dependent manner, pro-inflammatory cytokines production, CD8+ T cell cytotoxic activity against bronchial epithelial cells and fibrocyte immunomodulatory properties. We defined a computational model describing these intercellular interactions and calibrated the parameters based on our experimental measurements. We show the model's ability to reproduce histological ex vivo characteristics, and observe an important contribution of fibrocyte-mediated CD8+ T cell proliferation in COPD development. Using the model to test therapeutic scenarios, we predict a recovery time of several years, and the failure of targeting chemotaxis or interacting processes. Altogether, our study reveals that local interactions between fibrocytes and CD8+ T cells could jeopardize the balance between protective immunity and chronic inflammation in the bronchi of COPD patients.

## eLife assessment

The manuscript by Eyraud and colleagues examines the role of interactions between fibrocytes and CD8 cells as drivers of disease progression in COPD (chronic obstructive pulmonary disease). The findings that there exist bidirectional interactions between CD8 cells and fibrocytes are supported by **solid** evidence that combines histology of clinical lung samples, in vitro studies obtained from circulating blood fibrocytes and CD8 cells, as well as a computational model that predicts how bidirectional interactions could promote disease progression over the course of 20 years. The study, which is based on patient samples, thus provides **fundamental** insights on COPD progression.

## Introduction

The prevalence of COPD, one of the most common chronic diseases worldwide, has been rising in recent decades (*Mannino and Buist, 2007*; thus, prevention and treatment of COPD are important issues of global healthcare. COPD bronchi are an area of intense immunological activity and tissue remodeling, as evidenced by the extensive immune cell infiltration and changes in tissue structures such as peribronchial fibrosis. In particular, distal airways are hypothesized to constitute a 'quiet zone,' where exaggerated remodeling and inflammatory processes take place early in the history of the disease, without identifiable symptoms or lung function tests alteration (*Hogg et al., 1970*; *Mead, 1970*). In these particular areas, persistent contacts occur between resident cells and stimulated immune cells migrating from the peripheral circulation to the distal airways. The relevance of direct contact between T cells and monocyte-macrophages to potentiate the inflammatory response has been demonstrated in many chronic inflammatory diseases affecting the central nervous system, osteoarticular structures, and the lungs (*Dayer, 2003*), but remains to be fully investigated in COPD.

Fibrocytes, fibroblast-like leukocytes, produced by the bone marrow and released in the peripheral circulation (*Bucala et al., 1994*), have been implicated in lung fibrosis (*Pilling et al., 2014*). They are also recruited in the blood of COPD patients during an acute exacerbation (*Dupin et al., 2016*). High circulating fibrocyte count during a COPD exacerbation is associated with an increased risk of death, suggesting that fibrocytes could be detrimental to the evolution of this disease (*Dupin et al., 2016*). We have also demonstrated that tissue fibrocytes density increases in COPD bronchi, which was associated with a degraded lung function, increased wall thickness, and air trapping (*Dupin et al., 2019*). However, the function of these fibrocytes in COPD lungs is not yet fully understood (*Dupin et al., 2018*). Besides their role in tissue scarring matrix production (*Bucala et al., 1994*) and contraction (*Henrot et al., 2022*), recruited fibrocytes may participate in lung inflammation by virtue of their immune properties. They can function as antigen-presenting cells with T cells (*Chesney et al., 1997*), which can in turn modulate fibrocyte differentiation (*Abe et al., 2001*; *Niedermeier et al., 2009*). Fibrocyte engagement in immunomodulation has been implicated in various diseases such as thyroid-associated ophthalmopathy (*Fernando et al., 2012*) and lung cancer (*Afroj et al., 2021*). Cytotoxic CD8+ T cells are predominant in the airways of COPD patients and their number inversely correlates with lung function (*O'Shaughnessy et al., 1997*). CD8+ T cell-deficient mice are protected against lung inflammation and emphysema induced by cigarette smoke exposure (*Maeno et al., 2007*) whereas the expression of molecules linked to tissue destruction, such as perforin, granzyme B, and ADAM15, correlate with disease severity (*Freeman et al., 2010*; *Wang et al., 2020*), suggesting CD8+ T cells implication in lung inflammation and destruction in COPD. Activation of CD8+ T cells is increased in COPD lung samples (*Roos-Engstrand et al., 2009*). Other studies have shown that CD8+ T cell activation could be partially T Cell Receptor (TCR)-independent (*Freeman et al., 2010*). The absence of increased expression of cytotoxic enzymes in peripheral blood CD8+ T cells from COPD patients argues in favor of a local activation within the lungs (*Morissette et al., 2007*). CD8+ T cells express an exhausted phenotype in the COPD lung, that may result from an over-activation thus participating in the defective response to infection in COPD (*McKendry et al., 2016*). However, CD8+ T cells activation mechanism as well as their precise contribution to COPD pathogenesis remain largely unknown.

A recent study showed that fibrocytes, derived from the blood of lung adenocarcinoma patients, could strongly enhance the proliferation of CD8+ T cells (*Afroj et al., 2021*). We thus hypothesized that CD8+ T cells and fibrocytes interact in the lungs, and that this interaction is critical in COPD pathology. Multiple immunostainings in combination with specific image analysis methods allow us to determine the spatial distribution of individual CD8+ T cells and fibrocytes within bronchial tissues

of both control subjects and COPD patients. Using in vitro fibrocyte and CD8$^+$ T cell-based experiments, we studied cell interplay in terms of relative chemotaxis, dynamics, proliferation, and cytokine secretion profile. We then integrated these findings into an agent-based computational model representing airways from either healthy or COPD patients enabling us to test how local interactions shape spatial distributions of cells in both conditions. We propose that slight dysregulations of intercellular interactions induce abnormal cell organization around the bronchi, ultimately causing a breakdown of tissue homeostasis, leading to chronic inflammation and tissue remodeling.

## Results

### Direct contacts between fibrocytes and CD8$^+$ T cells are more frequent in distal bronchial tissue from COPD patients than in that of controls

We used immunohistochemistry (IHC) to assess whether fibrocytes and CD8$^+$ T cells were in close vicinity in human tissue. Sections of distal lung tissues from 17 COPD and 25 control patients were obtained, from a previously described cohort (*Dupin et al., 2019*), and labeled to detect CD8$^+$ T cells, identified as cells positive for CD8 staining and fibrocytes, identified as cells dual positive for FSP1 and CD45 double staining (*Figure 1—figure supplement 1A–D*). In agreement with previous studies (*Dupin et al., 2019*; *O'Shaughnessy et al., 1997*; *Saetta et al., 1998*), the density of both CD8$^+$ T cells and fibrocytes was increased within the subepithelial area of distal bronchi from COPD patients compared with that of control subjects (*Figure 1A–C*). Moreover, fibrocytes and CD8$^+$ T cells were frequently in close proximity (*Figure 1D*). To quantify the potential for cell-cell contacts, we determined the density of CD8$^+$ T cells in interaction with CD45$^+$ FSP1$^+$ cells (*Figure 1—figure supplement 1A–D*). Whatever the magnification used to automatically count interacting cells, the density of CD8$^+$ T cells in interaction with fibrocytes was higher in the sub-epithelial region of distal bronchi of COPD patients than in that of control subjects (*Figure 1D–F*). For subsequent analyses, we chose the dilatation size 'D8' (3.6 µm, which represents the radius of a mean ideal round cell in our analysis) to reflect the density of interacting cells. To evaluate the minimal distance between CD45$^+$ FSP1$^+$ cells and neighboring CD8$^+$ T cells, we used a CD8 distance map generated from the CD8 staining image, with the brightness of each pixel reflecting the distance from a CD8$^+$ T cell (*Figure 1—figure supplement 1E–F*). The mean minimal distance between fibrocytes and CD8$^+$ cells was significantly smaller in the sub-epithelial region of distal bronchi of COPD patients than in that of control subjects (*Figure 1G–H*). In contrast, the mean minimal distances between CD8$^+$ T cells themselves or between fibrocytes themselves were unchanged (*Figure 1—figure supplement 2A–B*). The majority of both CD8$^+$ T cells and fibrocytes were located beneath the epithelium, with their minimal distance and distribution relative to the basal membrane being similar in control and COPD patients (*Figure 1—figure supplement 2C–F*). Altogether, no difference in spatial repartition was observed within each cell population between control and COPD patients, but the relative distribution of fibrocytes and CD8$^+$ cells was affected in tissues from patients with COPD.

To further describe the relative spatial organization of both cell types, we used a method based on Delaunay triangulation computed on previously segmented cell barycenters. It is based on a custom-developed plugin to determine congregations of small groups of cells, called 'clusters' (*Figure 1—figure supplement 3*). As expected from our minimal distance analysis, we found a difference neither in the density of single cell-type clusters nor in their size, measured by the mean number of cells by cluster, between control subjects and patients with COPD (*Figure 1I–J*, left and middle panels). However, the density of clusters containing both cell types ('mixed cell clusters') was higher in the distal bronchi of COPD patients than in those of control subjects, with a median number of 5 and 6 cells in these clusters in control and COPD tissues, respectively (*Figure 1I–J*, right panels). This result indicates that fibrocytes and CD8$^+$ T cells are found within close proximity in the peribronchial area of COPD patients, with possible co-organization of CD8$^+$ T cells and fibrocytes in mixed cell clusters, indicating that direct and/or indirect fibrocyte-CD8$^+$ T cell interactions might occur in vivo.

### Relationships between the density of CD8$^+$ T cells interacting with fibrocytes and functional parameters

We determined the univariate correlation coefficients between fibrocyte density, CD8$^+$ T cell density, the three variables quantifying the interaction of CD8$^+$ T cells with fibrocytes (the interacting cell

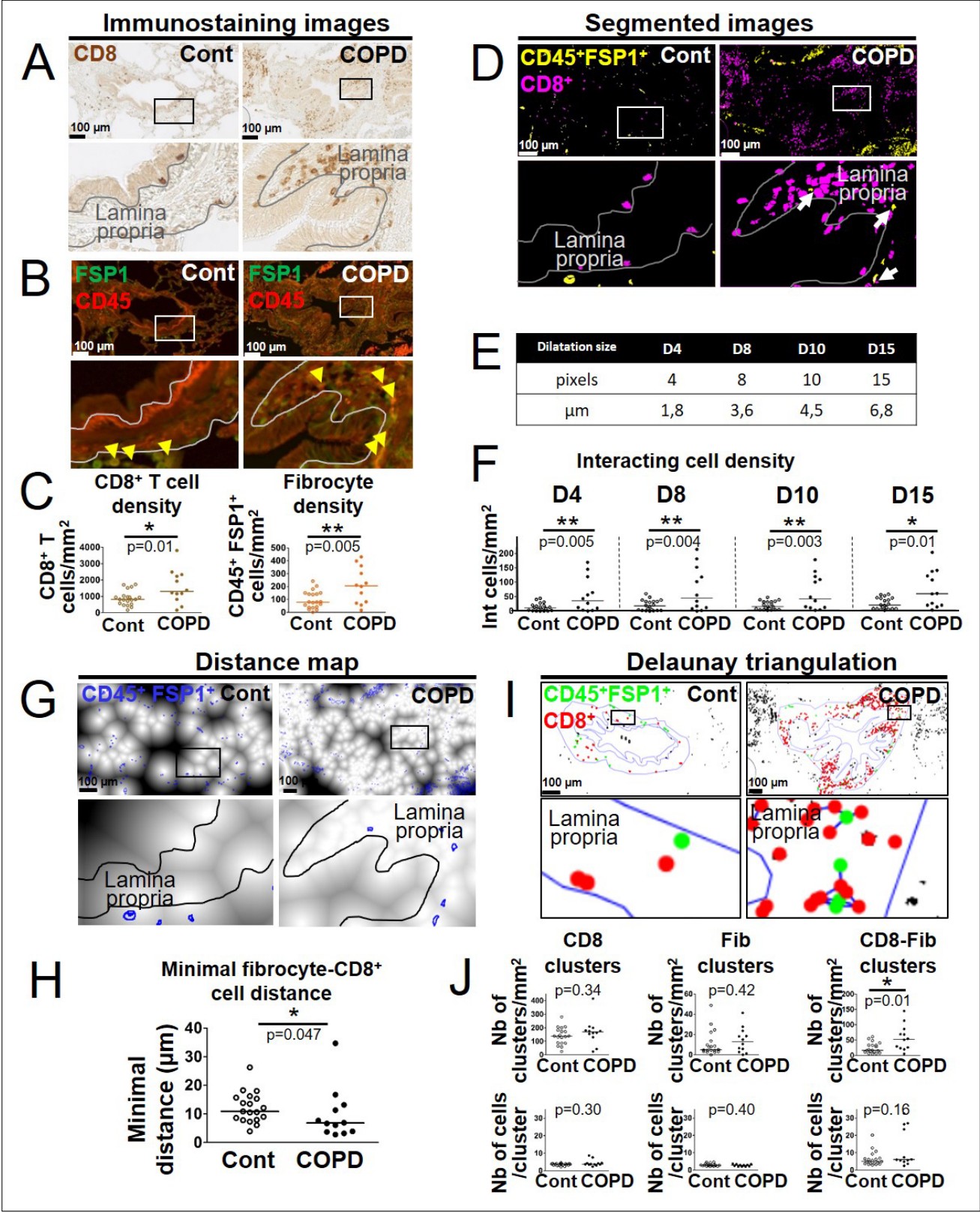

**Figure 1.** Increased interactions between CD8+ T cells, CD45+ FSP1+ cells in distal airways of chronic obstructive pulmonary disease (COPD) patients. (**A, B**) Representative stainings of CD8 (brown, **A**), CD45 (red, **B**), and FSP1 (green, **B**) in distal bronchial tissue specimens from a control subject (left) and a COPD patient (right). The yellow arrowheads indicate fibrocytes, defined as CD45+ FSP1+ cells. (**C**) Quantification of CD8+ T cells and fibrocyte densities (normalized by the sub-epithelial area) in one specimen/patient (n=20 control subjects, n=12 patients with COPD). (**D**) Merged segmented images for

*Figure 1 continued on next page*

*Figure 1 continued*

CD8 and CD45-FSP1 staining, showing CD8$^+$ T cells and CD45$^+$ FSP1$^+$ cells, respectively, in magenta and yellow. The white arrows indicate interacting cells, detected by dilatation of CD8 positive particles. (**E**) Table showing the correspondence between dilatations in pixels and µm. (**F**) Quantification of interacting cell densities (normalized by the sub-epithelial area) in one specimen/patient, using the different dilatations sizes (**E**). (**G**) Distance maps built from the binary image produced from CD8 staining, with FSP1$^+$ CD45$^+$ cells (blue outlines). (**H**) Quantification of the mean minimal distances between fibrocyte and CD8$^+$ T cells in one specimen/patient. (**I**) Cluster analysis performed by Delaunay triangulation on segmented images for CD8 and CD45-FSP1 staining, followed by the application of a threshold value (40 µm) above which connections are not kept. CD8$^+$ T cells and fibrocytes appear, respectively, with green and red dots, connections are shown in blue. (**J**) First row: densities of clusters containing exclusively CD8$^+$ T cells ('CD8 clusters'), fibrocytes ('Fib clusters'), and both cell types ('CD8-Fib clusters') normalized by the sub-epithelial area in one specimen/patient. Second row: mean number of cells by cluster. (**C, F, H, J**) The medians are represented as horizontal lines, n=20 specimens from control subjects, n=12 specimens from patients with COPD. *p<0.05, **p<0.01; ***p<0.001. unpaired t-tests or Mann-Whitney tests.

The online version of this article includes the following figure supplement(s) for figure 1:

**Figure supplement 1.** Detection of CD8$^+$ T cells, CD45$^+$ FSP1$^+$ cells, and quantification of interacting cells, and the minimal distances between the two cell types.

**Figure supplement 2.** Spatial distributions of CD8$^+$ T cells and fibrocytes in the peribronchial area.

**Figure supplement 3.** Principle of the use of Delaunay triangulation for cluster analysis.

**Figure supplement 4.** Relationships between the forced expiratory volume in 1 second (FEV$_1$)/forced vital capacity (FVC) ratio, the density of fibrocytes, the density of interacting cells, the mean minimal distance between fibrocytes and CD8$^+$ T cells, and the density of fibrocytes-CD8$^+$ T cells clusters.

density, the mean minimal distance between fibrocytes and CD8$^+$ T cells, and the density of mixed cell clusters), and various functional and CT parameters (*Supplementary files 1-5*). In particular, moderate but significant univariate correlations were found between the Forced Expiratory Volume in 1 s/Forced Vital Capacity (FEV$_1$/FVC) ratio (used to diagnose COPD if below 0.7), and the density of fibrocytes, the density of interacting cells, the mean minimal distance between fibrocytes and CD8$^+$ T cells and the density of fibrocytes-CD8$^+$ T cells clusters (*Figure 1—figure supplement 4A–D*). Variables significantly correlated with FEV$_1$/FVC were entered into stepwise regression analyses in order to find the best model fitting FEV$_1$/FVC. The best model associated the density of interacting cells and the density of mixed cell clusters. It explained 35% of the FEV$_1$/FVC variability (*Supplementary file 6*). The relationships between the FEV$_1$/FVC ratio, the density of interacting cells, and the density of mixed cell clusters were all statistically significant.

## Chemo-attraction of CD8$^+$ T cells for fibrocytes is increased in COPD tissue

To decipher the molecular mechanisms underpinning the increased cell-cell interaction in COPD bronchi, we investigated cell adhesion and chemotaxis processes in CD8$^+$ T cells of patients with COPD compared with those of controls. Using the transcriptomic profile of tissular resident memory and effector memory CD8$^+$ T cells of COPD patients compared with that of control subjects in the GSE61397 microarray dataset (http://www.ncbi.nlm.nih.gov/geo/) published elsewhere (*Hombrink et al., 2016*), we noted significative changes in the abundance of transcripts of genes related to cell adhesion. However, the changes were not consistent with clear increased or decreased adhesive properties in both tissue-resident memory CD8$^+$ CD103$^+$ T-cells (T$_{RM}$) and effector memory CD8$^+$ CD103$^-$ T-cells (T$_{EM}$) (*Figure 2—figure supplement 1*). In contrast, transcriptomic data reveal consistent changes in COPD cells *versus* controls, mostly increases, in chemokines and chemokine receptors (*Figure 2A*). Most changes in transcripts were compatible with a pro-attractive and a pro-migratory response. In particular, there were increases of CCL2, CCL26, CXCL2, and CXCL8 expression in T$_{RM}$ from patients with COPD, and CCL3L1 expression in T$_{EM}$ from patients with COPD (*Figure 2A*).

We then investigated whether tissular CD8$^+$ T cells secretion from control or COPD patients could affect fibrocytes migration in an in vitro assay (*Figure 2B*). CD8$^+$ T cells were purified from lung resection material sampled either in control subjects or in COPD patients, whose characteristics are reported (*Supplementary file 7*). Precursors of fibrocytes were purified from blood samples of a separate cohort of COPD patients (*i.e.* COBRA), whose characteristics are also reported (*Supplementary file 8*). The migration of fibrocytes was significantly increased by conditioned medium derived from tissular CD8$^+$ T cells of COPD patients compared with those from control lungs (*Figure 2C*).

The secretory profile of these tissular CD8$^+$ T cells 36 hr after culture conditions with soluble anti-CD3 and anti-CD28 antibodies were determined. The concentration of CXCL8 was increased in CD8$^+$

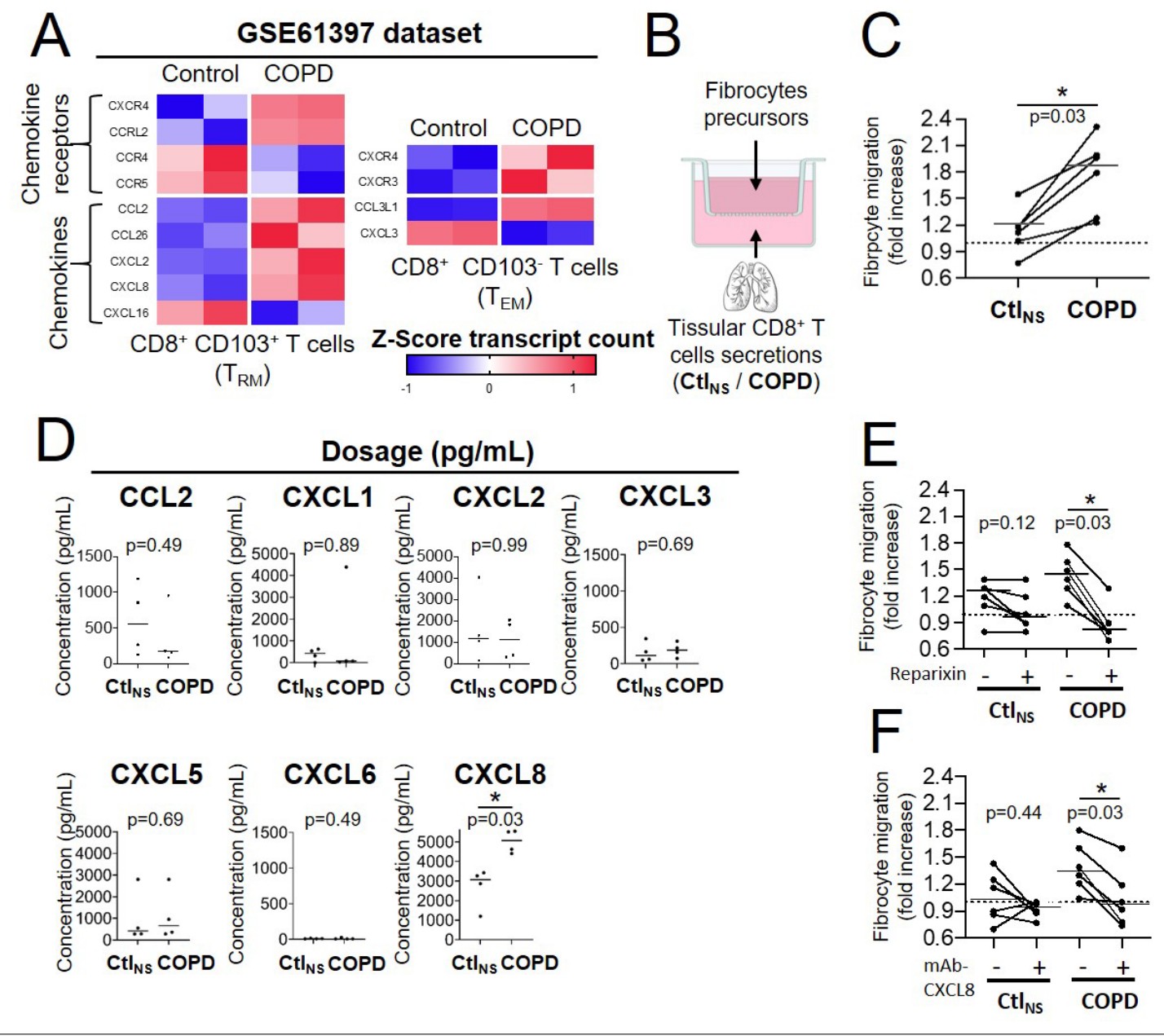

**Figure 2.** CD8+ T cells from chronic obstructive pulmonary disease (COPD) tissue have increased chemoattractive properties for fibrocytes. (**A**) Heatmaps showing the expression of differentially expressed genes with p-value <0.05 of chemokines and chemokine receptors in resting tissular tissue-resident memory T-cells ($T_{RM}$) and effector memory T-cells ($T_{EM}$) from patients with COPD (n=2 independent samples) in comparison with control subjects (n=2 independent samples) (GEO accession GSE61397). Expression values are expressed as Z-score transformed transcript count. (**B**) Migration experiment design. (**C**) Migration of fibrocytes from patients with COPD in response to CD8+ T cells supernatants from control subjects ($Ctl_{NS}$) or COPD patients (COPD). n=6 independent experiments (**D**) CCL2, CXCL2, and CXCL8 levels in CD8+ T cells supernatants from non-smoking control subjects ($Ctl_{NS}$) or patients with COPD (COPD) using BioPlex (CCL2, CXCL2) or ELISA (CXCL1, 3, 5, 6, 8). n=4 $Ctl_{NS}$ samples, n=4 COPD samples. *p<0.05, Mann-Whitney test. (**E–F**), Migration of fibrocytes from patients with COPD in response to CD8+ T cells supernatants from control subjects ($Ctl_{NS}$) or COPD patients (COPD), in the presence of 200 nM Reparixin (+) or corresponding vehicle (−) (**E**), and in the presence of 1 μg/mL blocking antibody for CXCL8 (+) or control antibody (−) (**F**). n=6 independent experiments *p<0.05, Wilcoxon matched pairs test.

The online version of this article includes the following figure supplement(s) for figure 2:

**Figure supplement 1.** Transcript levels of adhesion genes in CD8+ T cells from chronic obstructive pulmonary disease (COPD) tissues compared to control tissues.

T cells from COPD patients compared to control cells (*Figure 2D*) in good agreement with the transcriptomic analysis. By contrast, the concentration of both CCL3 and CCL3L1 was undetectable (data not shown), whereas that of CCL2, CXCL1, CXCL2, CXCL3, CXCL5, and CXCL6 remained unchanged (*Figure 2D*). Since CXCL8 is a ligand of the chemokine receptors CXCR1 and/or CXCR2, we repeated the migration assay with the addition of the drug reparixin, an antagonist of both CXCR1 and CXCR2 (*Bertini et al., 2004*). Whereas fibrocyte treatment with reparixin had no significant effect on the control CD8+ T cells-mediated migration, it did inhibit the increased migration induced by the secretions of CD8+ T cells purified from COPD tissues (*Figure 2E*). Moreover, an anti-CXCL8 blocking antibody also inhibited the increased migration induced by the secretions of CD8+ T cells purified from COPD tissues, in the same extend than the blocking of CXCR1/2 by reparixin (*Figure 2F*), suggesting that this supplementary chemotaxis is mainly due to CXCL8 and not other CXCR1/2 binding CXCL chemokines. These data indicate that tissular CD8+ T cells from patients with COPD promote fibrocyte chemotaxis via CXCL8-CXCR1/2 axis.

## CD8+ T cells repeatedly interact with fibrocytes

As fibrocytes and CD8+ T cells reside in close proximity in the subepithelial area, especially that of tissues from COPD patients, we investigated their crosstalk capacity. We developed an autologous in vitro co-culture system allowing precise control over the cell types involved. Fibrocytes and CD8+ T cells, both purified from the blood of COPD patients were co-cultured 2 days before image acquisition for the following 12 hr. CD8+ T cells were either nonactivated or activated with anti-CD3/CD28 antibodies coated microbeads, whereas fibrocytes were not stimulated. At the beginning of live imaging, nonactivated CD8+ T cells were equally allocated in fibrocyte-covered zones (41 ± 8%) and in fibrocyte-free zones (59 ± 8%) (*Figure 3A–B*). Twelve h later, most (77 ± 9%) of CD8+ T cells were present in contact with fibrocytes (*Figure 3A–B*). Activation of CD8+ T cells resulted in similar distribution (*Figure 3A–B*). These data suggest that both cell types are able to directly interact, and that these interactions progressively increase during co-culture. We tracked individual CD8+ T cells during 12 hr time-lapse to capture their spatiotemporal dynamics using multiple variables quantification (*Figure 3C* and *Video 1*). For both nonactivated and activated lymphocytes, the mean speed of CD8+ T cells decreased upon contact with fibrocytes (*Figure 3D*). Irrespective of the activation state of CD8+ T cells, a majority of intercellular contacts (49 ± 6% and 49 ± 8% for nonactivated and activated CD8+ T cells, respectively) were short-lived (<12 min) and dynamic, although some longer interactions (>32 min) could also be detected (30 ± 4% and 27 ± 7% for nonactivated and activated CD8+ T cells, respectively) (*Figure 3E*). The contact coefficient and the mean velocity of CD8+ T cells measured in the absence of contact with fibrocytes ('Mean free speed') were similar in both conditions of activation (*Figure 3F–G*). However, we observed a significant decrease in the mean speed for activated CD8+ T cells when they were in contact with fibrocytes ('Mean contact speed') compared to nonactivated CD8+ T cells (*Figure 3H*), reflecting subtle behavior changes in this condition of activation.

## Fibrocytes favor CD8+ T cell proliferation in a cell-cell contact-dependent manner

Since multiple transient contacts have been shown to be an early trigger of events leading to clonal expansion (*Obst, 2015*), we wondered whether fibrocytes could promote CD8+ T cells proliferation using total cell count and a CellTrace-based co-culture proliferation assay. We designed two different co-culture assays modeling either a direct contact between the two cell types or an indirect contact (transwell assay). The activation of CD8+ T cells by anti-CD3/CD28 antibody-coated microbeads slightly increased the basal level of dividing CD8+ T cells (comparison of the conditions 'CD8$_{NA}$' and 'CD8$_A$' without fibrocytes in *Figure 3I–P*). The presence of fibrocytes in the indirect co-culture assay did not affect the proliferation capacity of non-activated CD8+ T cells and only moderately increased the number of dividing activated CD8+ T cells (*Figure 3I–L*). The distinction between naïve (CD45RA+) and memory (CD45RA-) CD8+ T cells did not reveal any selective effect of fibrocytes on these two CD8+ subpopulations (*Figure 3—figure supplement 1A, C* and *Figure 3—figure supplement 1E–H*). In the direct co-culture model, the total number of CD8+ T cells and the percentage of dividing CD8+ T cells were far higher in the presence of fibrocytes irrespective of the activation state of CD8+ T cells (*Figure 3M–P*). This effect seemed to be particularly impressive for naïve CD8+ T cells as they demonstrated an average differential of 80 ± 14% and 70 ± 20% of dividing cells between the conditions with

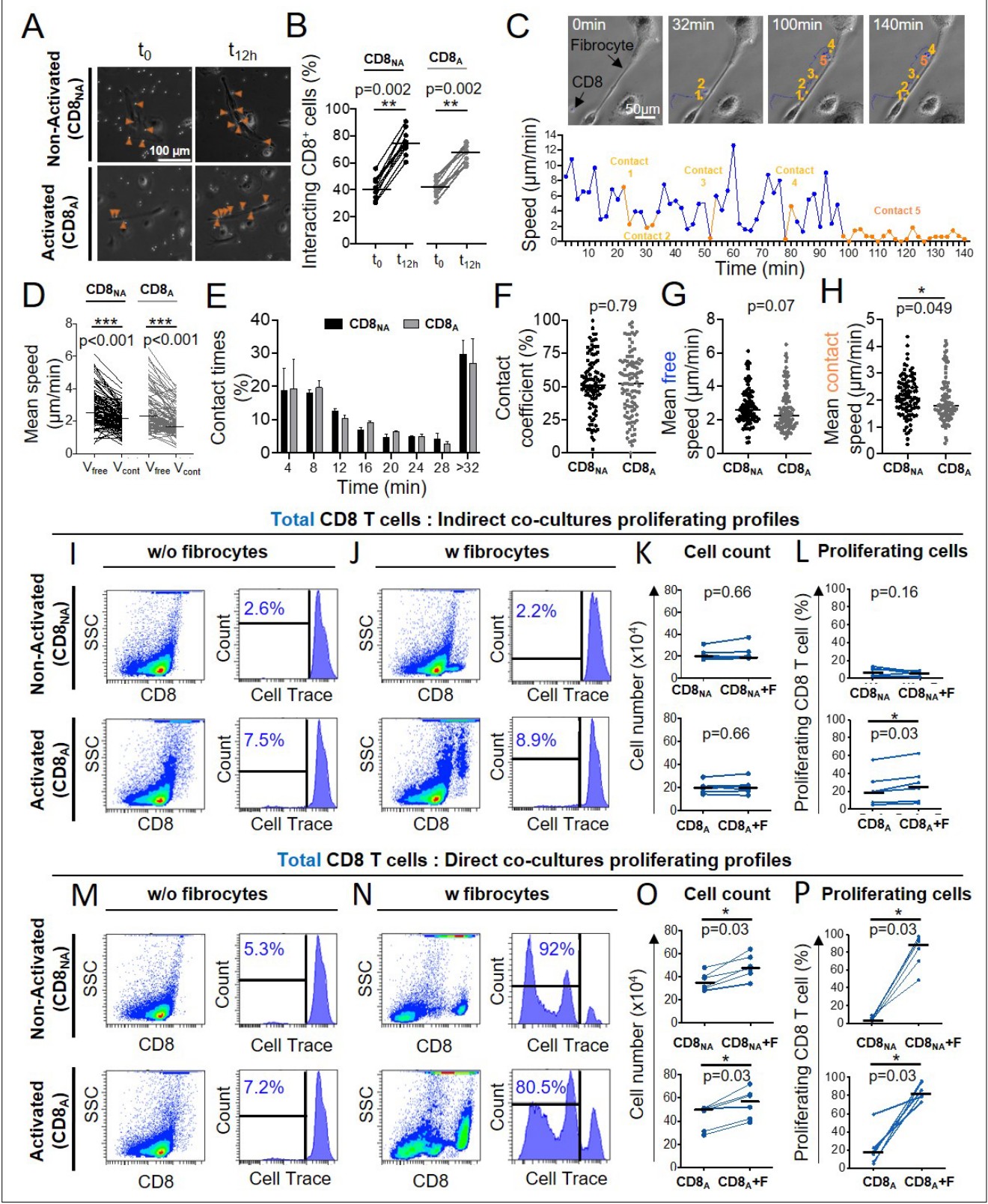

**Figure 3.** CD8[+] T cells repeatedly contacts fibrocytes and this contact greatly enhances CD8[+] T cell proliferation. Prior to co-culture, CD8[+] T cells have been either non-activated ('CD8_NA') or activated ('CD8_A'). (**A**) Representative brightfield images of co-culture between CD8[+] T cells and fibrocytes at the initial state of the acquisition (t_0) and after 12 hr (t_12hr) in both conditions of activation. The orange arrowheads indicate CD8[+] T cells (bright round-shaped cells) in contact with fibrocytes (elongated adherent cells). (**B**) Quantifications of the proportion of fibrocyte-interacting CD8[+] T cells at t_0 and

*Figure 3 continued on next page*

*Figure 3 continued*

$t_{12h}$rin both conditions of activation. (**C**) Top panel: typical CD8$^+$ T cells trajectory (blue) relative to a fibrocyte (elongated adherent cell) for a period of 140 min. Bottom panel: speed (µm/min) over time for the tracked CD8$^+$ T cell. Short-lived (<12 min, n=4) and longer-lived (>32 min, n=1) contacts are represented, respectively, in light and dark orange. (**D**) Comparison of the mean speed of individual CD8$^+$ T cells measured in the absence ('V$_{free}$') or presence ('V$_{cont}$') of contact with fibrocytes in both conditions of activation. (**E**) Mean frequency distributions of contact time duration (with 4 min binning) between CD8$^+$ T cells and fibrocytes for CD8$_{NA}$ (black) and CD8$_A$ (gray). Error bars indicate the standard error of the mean. Two-way ANOVA (**F–H**) Dot plots representing spatiotemporal variables measured for each individual CD8$^+$ T cell tracked over 12 hr. Each dot represents one cell. (**F**) Contact coefficient. (**G**) Mean speed of CD8$^+$ T cells measured in the absence of contact with fibrocytes ('Mean free speed'). (**H**) Mean speed of CD8$^+$ T cells measured in the presence of contact with fibrocytes ('Mean contact speed'). (**A–H**) n=2 independent experiments, n=10 videos by experiments, n=10 CD8$^+$T cells tracked by videos. (**I, J, M, N**) Representative gating strategy for identification of CD8$^+$ T cells without (w/o) fibrocytes (**I, M**) or with (**w**) fibrocytes (**J, N**) in indirect (**I, J**) or direct (**M, N**) co-culture. Left panels: dot plots represent representative CD8-PerCP-Cy5-5 fluorescence (y-axis) versus side scatter (SSC, x-axis) of non-adherent cells removed from the culture. Right panels: histograms represent representative cell count (y-axis) versus Cell Trace-Pacific Blue fluorescence (x-axis). The distinct fluorescence peaks correspond to the different generations of CD8$^+$ T cells. The gate and the percentage indicate cells that have proliferated. (**K, O**) Comparison of the manual count of non-adherent cells removed from co-culture without fibrocyte ('CD8') and with fibrocyte ('CD8 +F'). L, (**P**) Comparison of quantifications of CD8$^+$ T cells that have proliferated, removed from co-culture without fibrocyte ('CD8') and with fibrocyte ('CD8 +F'). (**I–P**) n=6 independent experiments. (**B, D, F, G, H, K, L, O, P**) Medians are represented as horizontal lines. *p<0.05, *p*P<0.01, ***p<0.001. (**B, D, K, L, O, P**) Wilcoxon matched pairs test. (**F, G, H**) Mann-Whitney tests.

The online version of this article includes the following figure supplement(s) for figure 3:

**Figure supplement 1.** Direct contact between fibrocytes and CD8$^+$ T cells preferentially increases proliferation of naïve CD8$^+$ T cells subset.

**Figure supplement 2.** CD4$^+$ T cells death and proliferation after 6 days in direct co-culture with fibrocytes.

**Figure supplement 3.** CD8$^+$ T cells death after 6 days of co-culture with fibrocytes.

**Figure supplement 4.** Direct contact between fibrocytes and CD8$^+$ T cells promotes phenotypic differences in CD8 expression.

**Figure supplement 5.** The CD8$^{low}$ population appears in co-culture and is distinct from the CD45$^+$ Collagen I$^+$ population.

and without fibrocytes, respectively, for nonactivated (*Figure 3—figure supplement 1I–J*, top panels) and activated CD8$^+$ T cells (*Figure 3—figure supplement 1I–J*, bottom panels), vs 67 ± 18% and 52 ± 20% for memory CD8$^+$ T cells (*Figure 3—figure supplement 1K–L*). We also performed co-cultures between fibrocytes and CD4$^+$ T cells, with the same settings than for CD8$^+$ T cells. The results from these experiments show that fibrocytes did not have any significant effect on CD4$^+$ T cells death, irrespective of their activation state (*Figure 3—figure supplement 2A–C*). Fibrocytes were able to promote CD4$^+$ T cells proliferation in the activated condition but not in the non-activated condition (*Figure 3—figure supplement 2A–D*). Altogether, this implies that a direct rather than indirect interactions between CD8$^+$ T cells and fibrocytes increased CD8$^+$ T cell proliferation, and that although fibrocyte-mediated effect on proliferation is not specific to CD8$^+$ T cells, the extend of the effect is much larger on CD8$^+$ T cells than on CD4$^+$ T cells.

Taking advantage of the staining of CD8$^+$ T cells with the death marker Zombie NIR, we have also quantified CD8$^+$ T cell death in our co-culture assay. The presence of fibrocytes in the indirect co-culture assay did not affect CD8$^+$ T cell death (*Figure 3—figure supplement 3A–B*). In direct co-culture, the death of CD8$^+$ T cells was significantly increased in the non-activated condition but not in the activated condition (*Figure 3—figure supplement 3C–D*).

After 6 days of co-culture, a cell population with a low level of CD8 expression (CD8$^{low}$) appeared, that was inversely proportional to the level of CD8$^+$ T cells strongly expressing a high level of CD8 (CD8$^{high}$, *Figure 3—figure supplement 4*). The CellTrace-based assay showed that those cells highly proliferated during co-culture, especially in the direct co-culture (*Figure 3—figure supplement 4E*), suggesting that CD8$^{high}$ cells disappeared in favor of CD8$^{low}$ cells. As fibrocytes could have contaminated the cell suspension harvested from the direct co-culture, we did check that those CD8$^{low}$ cells were not CD45$^+$ Collagen I$^+$ (*Figure 3—figure supplement 5*). Phenotypic analysis of this CD8$^{low}$ population indicated that cells were mostly CD45RA$^-$ cells (*Figure 3—figure supplement 4A–B*, S9D-E), with a low level of cytokine expression (*Figure 3—figure supplement 4C, F*). Since CD8$^{low}$ cells may thus represent a population of exhausted T cells, we focused on CD8$^{high}$ cells in the following, especially regarding the secretion profile characterization. As CD86 and CD54 co-stimulatory molecule and adhesion molecules, respectively, pivotal in immunological synapse formation, are both expressed by fibrocytes (*Afroj et al., 2021*; *Balmelli et al., 2005*), we tested the effects of anti-CD54 and anti-CD86 blocking antibodies on fibrocyte-induced proliferation of CD8$^+$ T cells. The inhibition of CD86 and CD54 significantly reduced the proliferation of nonactivated CD8$^+$ T cells in the direct co-culture with

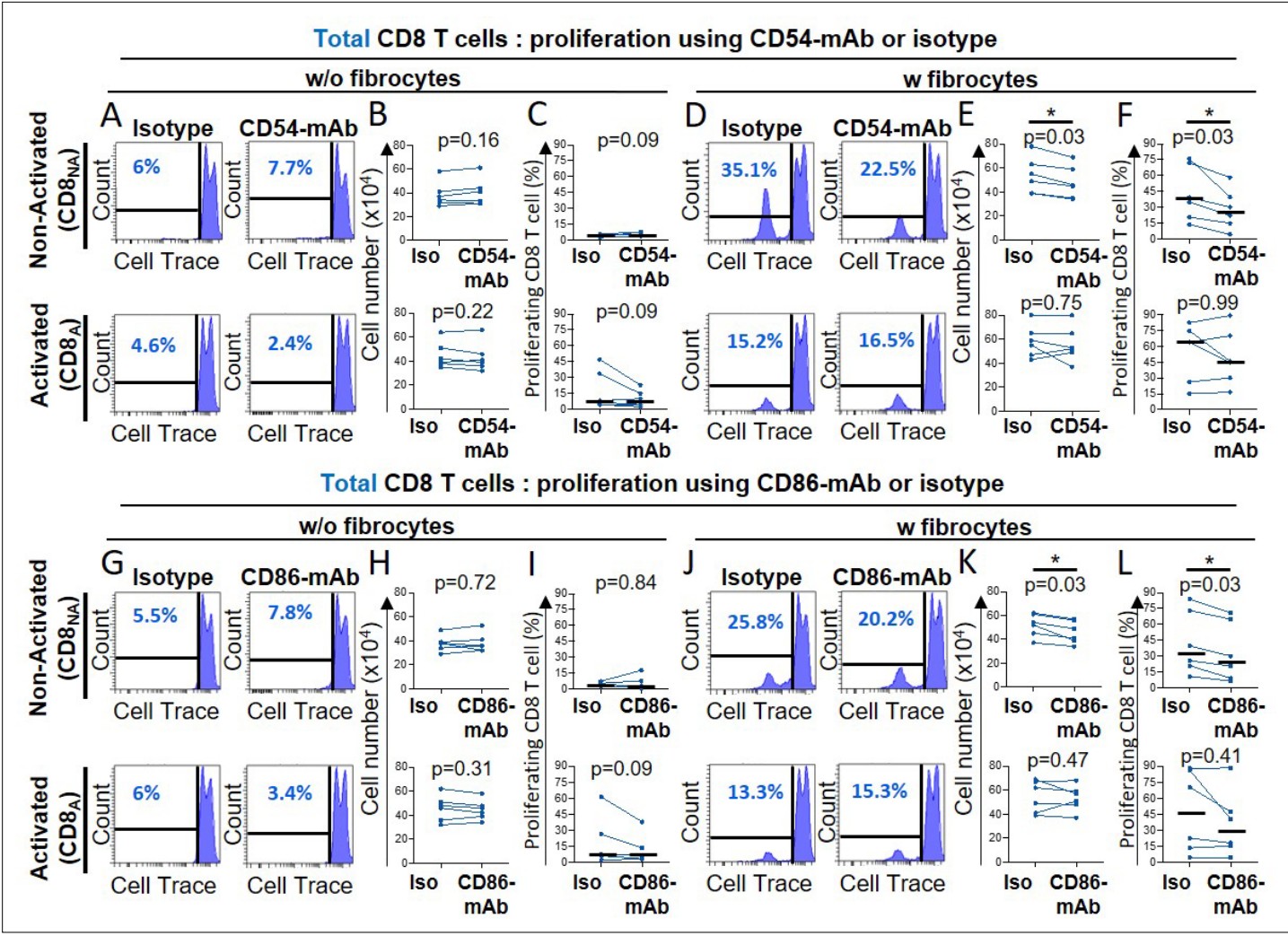

**Figure 4.** Fibrocytes act as a major promoter of CD8+ T cell proliferation in a CD54 and CD86-dependent manner. Prior to co-culture, CD8+ T cells have been either non-activated ('CD8$_{NA}$') or activated ('CD8$_A$'). (A, D, G, J) Representative gating strategy for identification of proliferating CD8+ T cells without (w/o) fibrocytes (A, G) or with (w) fibrocytes (D, J) using neutralizing CD54-mAb (A, D) or neutralizing CD86-mAb (G, J) and respective control isotype. Histograms represent representative cell count (y-axis) versus Cell Trace-Pacific Blue fluorescence (x-axis). The distinct fluorescence peaks correspond to the different generations of CD8+ T cells. The gate and the percentage indicate cells that have proliferated. (B, E, H, K) Comparison of the manual count of non-adherent cells removed from co-culture treated with neutralizing CD54-mAb or control isotype (Iso) (B, E) and neutralizing CD86-mAb or control isotype (Iso) (H, K). (C, F, I, L) Comparison of quantifications of CD8+ T cells that have proliferated, removed from co-culture treated with neutralizing CD54-mAb (C, F) or neutralizing CD86-mAb (I, L) and respective control isotype. n=6 independent experiments. Medians are represented as horizontal lines. *p<0.05, Wilcoxon matched paired tests.

The online version of this article includes the following figure supplement(s) for figure 4:

**Figure supplement 1.** LFA-1 or CD44 blockade is not sufficient to decrease CD8+ T cells proliferation induction.

fibrocytes (*Figure 4*). However, these antibodies failed to alter the stimulatory activity of lymphocyte division by fibrocytes, when CD8+ T cells were previously activated (*Figure 4*). Blocking LFA-1 did not affect the fibrocyte-mediated CD8+ T cell division (*Figure 4—figure supplement 1A–D*), suggesting the existence of compensatory integrins at the surface of the lymphocyte, such as CD11b/CD18, to mediate the interaction with CD54. The inhibition of CD44, a receptor for hyaluronan which has been shown to be produced by fibrocytes (*Bianchetti et al., 2012*), did not impair the proliferation of CD8+ T cells irrespective of their activation state (*Figure 4—figure supplement 1E–H*).

In total, these results indicate that direct contacts between fibrocytes and CD8+ T cells, such as those mediated by CD54 and CD86, were strong positive signals to trigger CD8+ T cell proliferation with the induction of CD8$^{high}$ and CD8$^{low}$ phenotypes.

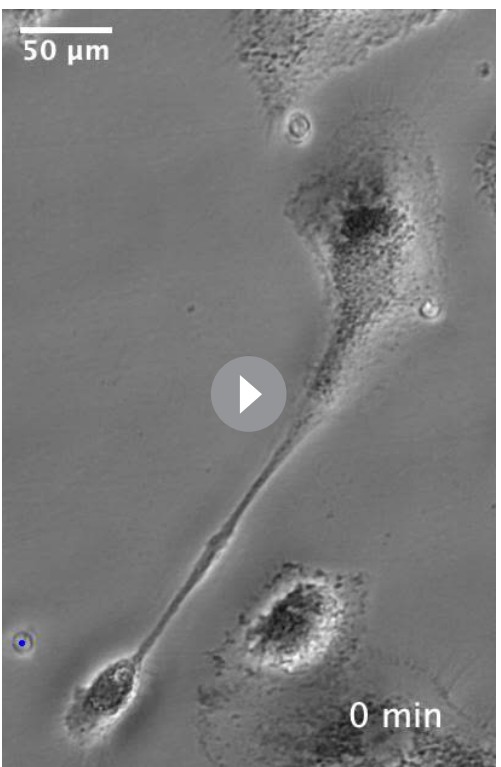

**Video 1.** Two days after adding non-activated CD8+ T cells (bright round cells) on fibrocytes (adherent elongated cells), phase-contrast images of co-culture taken were recorded every 2 min. A tracked lymphocyte is indicated by a blue dot and its trajectory is shown by a blue line dot (Manual Tracking plugin, Fiji software). https://elifesciences.org/articles/85875/figures#video1

## Fibrocyte-CD8+ T cell interactions alter cytokine production and promote CD8+ T cell cytotoxicity

Multiparametric flow cytometry was used to characterize the cytokine expression profile of CD8+ T cells in the indirect and direct co-culture with fibrocytes. When nonactivated CD8+ T cells were indirectly co-cultured with fibrocytes, the expression of TNF-α, IFN-γ by CD8+ T cells was slightly increased (*Figure 5A–B*). IL-10, IL-17, and Granzyme B were not detected (*Figure 5A–B*). When CD8+ T cells were activated with anti-CD3/CD28, the level of TNF-α and IFN-γ further increased, and the expression of granzyme B and IL-10 was slightly induced (*Figure 5A–B*). Upon direct co-culture, we observed a massive induction of TNF-α, IFN-γ, granzyme B, IL-10, and IL-17, irrespective of the activation state of CD8+ T cells (*Figure 5C–D*). Altogether, these results show that soluble factors and direct contacts between fibrocytes and CD8+ T cells might have an additive effect on CD8+ T cell cytokine production. The concentration of TNF-α measured in culture supernatant increased significantly upon co-culture between fibrocytes and non-activated CD8+ T cells at day 4, confirming that TNF-α was secreted in the medium upon direct interactions with fibrocytes (*Figure 5E*). This shows that both soluble factors produced by fibrocytes and direct contacts influence the CD8+ T cell secretion profile.

We then wondered whether glucocorticoid drugs (*i.e.* budesonide or fluticasone propionate) could reverse the fibrocyte-induced proliferation and differentiation of CD8+ T cells. Treatment with glucocorticoid drugs significantly decreased fibrocyte-induced TNF-α secretion by non-activated CD8+ T cells, without affecting the proliferation (*Figure 5—figure supplement 1*). Collectively, these results underline the importance of the interaction with fibrocytes for CD8+ T cell activation, possibly by favoring cellular proliferation and local cytokine production.

Having shown that fibrocytes promoted CD8+ T cells expression of cytotoxic molecules such as granzyme B, we decided to investigate the cytotoxic capacity of CD8+ T cells against primary basal bronchial epithelial cells (see *Supplementary file 9* for patient characteristics). Direct co-culture with fibrocytes increased total and membrane expression of the cytotoxic degranulation marker CD107a, which was only significant in non-activated CD8+ T cells (*Figure 6A–E*). A parallel increase of cytotoxicity against primary epithelial cells was observed in the same condition (*Figure 6F–H*). This demonstrates that following direct interaction with fibrocytes, CD8+ T cells have the ability to kill target cells such as bronchial epithelial cells.

## Direct contact with CD8+ T cells triggers fibrocyte engagement toward immunologic signaling

To analyze the effect of the interaction on the fibrocyte, we performed proteomic analyses on fibrocytes, alone or in co-culture during 6 days with CD8+ T cells either non-activated or activated (*Figure 7A*). Of the top ten pathways that were most significantly activated in co-cultured versus mono-cultured fibrocytes, the largest upregulated genes were those of the dendritic cell maturation box, the multiple sclerosis signaling pathway, the neuroinflammation signaling pathway, and the macrophage classical signaling pathway, irrespective of the activation state of CD8+ T cells (*Figure 7B*). The changes

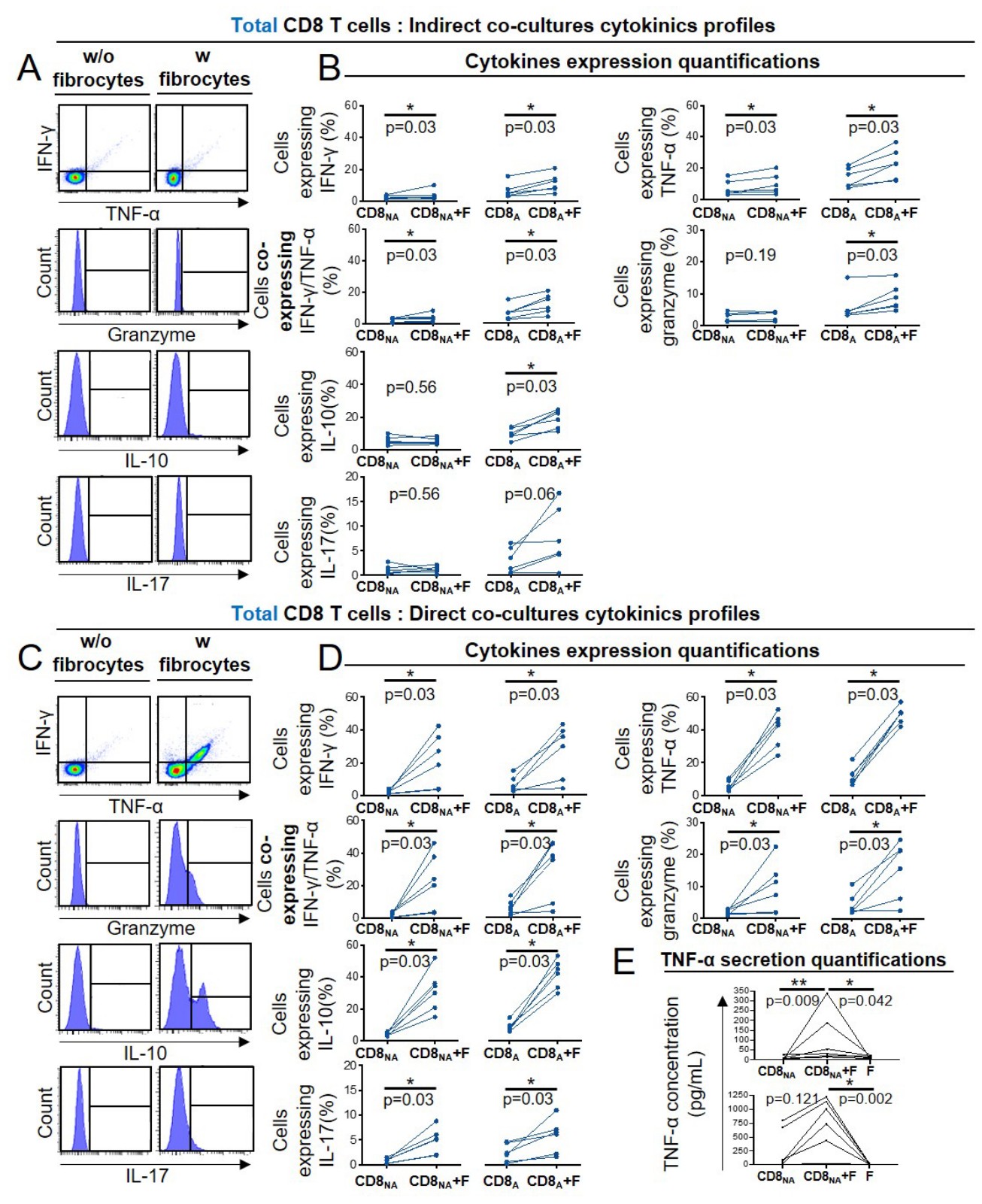

**Figure 5.** Fibrocyte-CD8+ T cell interactions alter cytokine production. Prior to co-culture, CD8+ T cells have been either non-activated ('CD8$_{NA}$') or activated ('CD8$_A$'). (**A, C**) Representative gating strategy for identification of CD8+ T cells expressing IFN-γ, TNF-α, granzyme, IL-10, and IL17 without (w/o) fibrocytes (left panel) or with (w) fibrocytes (right panel) in indirect (**A**) or direct (**C**) co-culture. (**B, D**) Quantifications of CD8+ T cells expressing IFN-γ, TNF-α, both, granzyme and IL-10 after co-culture without fibrocytes (CD8$_{NA}$/CD8$_A$) or with fibrocytes (CD8$_{NA}$/CD8$_A$ +F) in indirect (**B**) or direct

*Figure 5 continued on next page*

Figure 5 continued

(D) co-culture. (E) TNF-α concentrations in supernatants from co-cultures without fibrocytes (CD8$_{NA}$/CD8$_A$), with fibrocyte (CD8$_{NA}$/CD8$_A$ +F), and only with fibrocytes (F) as control, for direct co-cultures. n=6 independent experiments. *p<0.05, Wilcoxon matched paired tests or Friedman tests.

The online version of this article includes the following figure supplement(s) for figure 5:

**Figure supplement 1.** Glucocorticoid drugs significantly decrease fibrocyte-induced TNF-α secretion by CD8$^+$ T cells but not the proliferation induction.

were globally identical in the two conditions of CD8$^+$ T cell activation, with some upregulation more pronounced in the activated condition. They were mostly driven by up-regulation of a core set of Major Histocompatibility Complex class I (HLA-B, C, F) and II (HLA-DMB, DPA1, DPB1, DRA, DRB1, DRB3) molecules, co-simulatory and adhesion molecules (CD40, CD86, and CD54). Another notable proteomic signature was that of increased expression of IFN signaling-mediators IKBE and STAT1, and the IFN-responsive genes GBP2, GBP4, and RNF213. We also observed a strong downregulation of CD14, suggesting fibrocyte differentiation, and an upregulation of the matrix metalloproteinase-9 (MMP9) in the non-activated condition only. These changes suggest that the interaction between CD8$^+$ T cells and fibrocytes promotes the development of fibrocyte immune properties, which could subsequently impact the activation of CD4$^+$ T cells activation.

## Stochastic mathematical model taking into account intercellular interactions describes the evolution over time of cell populations in control and COPD bronchi

All the above mentioned results led us to hypothesize that fibrocyte infiltration into the lung, differential migration of fibrocytes towards CD8$^+$ T cells, and subsequent CD8$^+$ T cell proliferation, could result in a distinct spatial cellular repartition observed in tissues obtained from patients with COPD, compared to control tissues. To investigate this hypothesis, which could not be experimentally tested, we developed an agent-based (cellular automata) model with local and random cellular interactions. We considered the lamina propria (*i.e.* the peribronchial zone), located between the bronchial epithelium and the smooth muscle layer, which contains fibrocytes and CD8$^+$ T cells. In line with the present analysis, the computational domain (*i.e.* the lamina propria), corresponds to a zone of 179,000 μm$^2$. Fibrocytes and CD8$^+$ T cell are considered as individual objects that can move, divide, die, and infiltrate the lamina propria in a stable state and during exacerbation. Their individual behaviors and interactions are supposed to be stochastic and the value of the probabilities has been established from literature (*Afroj et al., 2021*; *Bivas-Benita et al., 2013*; *Dupin et al., 2016*; *Dupin et al., 2019*; *Ely et al., 2006*; *Freeman et al., 2007*; *Gribben et al., 1995*; *Hurst et al., 2010*; *Ling et al., 2019*; *McMaster et al., 2015*; *Mrass et al., 2017*; *Saetta et al., 1999*; *Scheipers and Reiser, 1998*; *Schmidt et al., 2003*; *Schyns et al., 2019*; *Siena et al., 2011*; *Takamura et al., 2016*; *Zenke et al., 2020*) and the present in vitro data, as summarized in the method section and in *Supplementary files 9 and 10*, and exhaustively described in the Appendix 1 and (*Dupin et al., 2023*). Initial cell densities were scaled with respect to reference values, corresponding to the mean densities measured in non-smoking subjects. Simulations started with these initial densities and ended 20 years later, to reflect the average time between the beginning of cigarette smoke exposure and COPD onset (*Løkke et al., 2006*).

All the biological processes are governed by probabilities (*Figure 8A*). CD8$^+$ T cells, but not fibrocytes, are able to proliferate, based on our own unpublished observations and other studies (*Ling et al., 2019*; *Schmidt et al., 2003*). The presence of fibrocytes in the local neighborhood of a CD8$^+$ T cell can trigger CD8$^+$ T cell division with an increased probability, based on the present in vitro experiments showing that the contact between those two cell types greatly enhanced CD8$^+$ T cell proliferation. When a CD8$^+$ T cell has many other T cells in its neighborhood, it can die with an increased probability, in agreement with (*Zenke et al., 2020*) and our in vitro results. Fibrocytes and CD8$^+$ T cells movements depend on the local neighborhood of cells, reflecting their relative chemo-attractive properties. We then simulated the evolution over 20 years, with two sets of parameters, respectively, for the control and COPD cases (see Appendix 1).

We first tested the results of simulations against our experimental data from patients' tissues. First, we compared cell densities, experimentally measured in tissue samples, with theoretical predictions

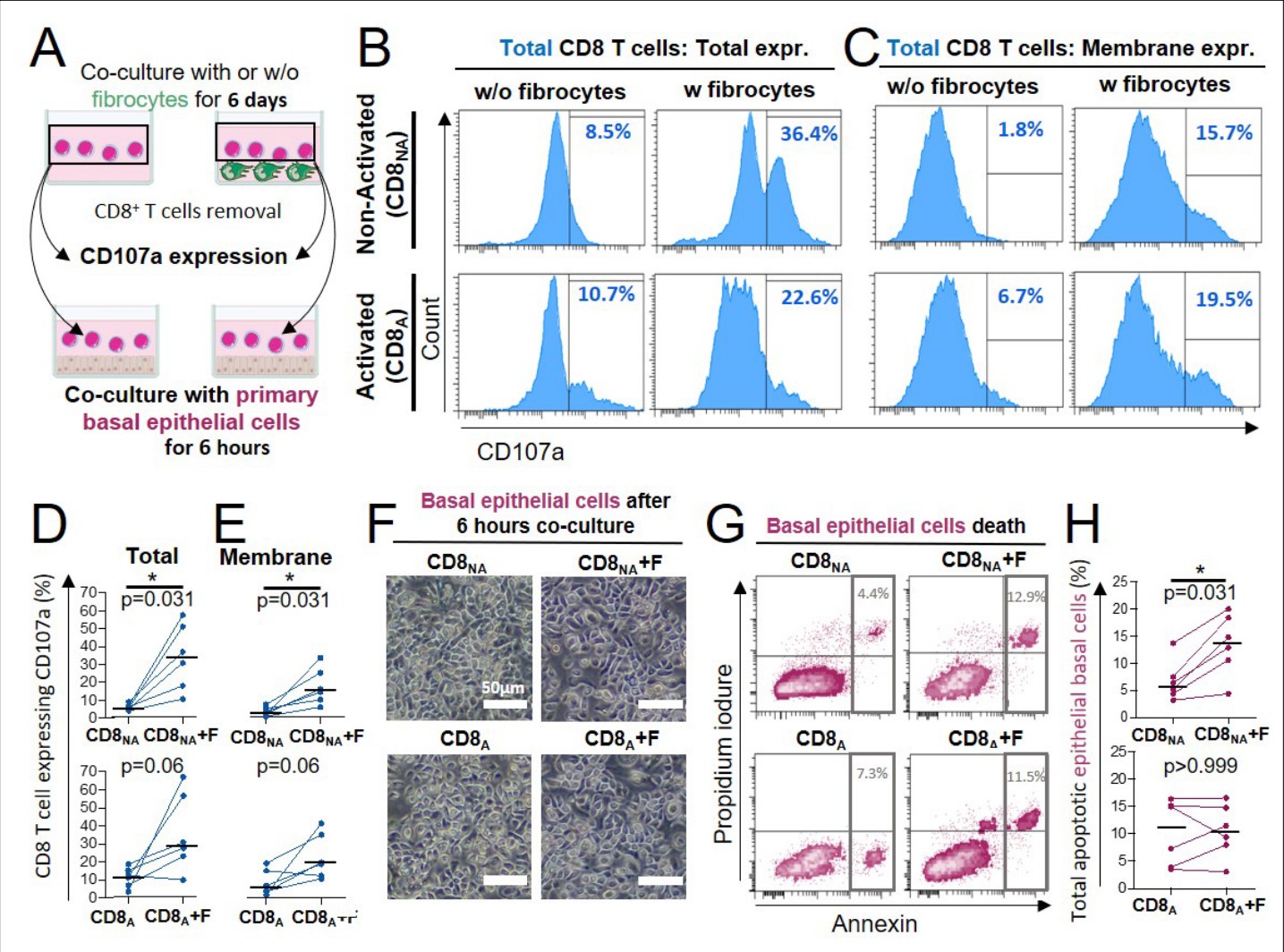

**Figure 6.** Direct contact between fibrocytes and CD8+ T cells triggers CD8+ T cell cytotoxicity against primary bronchial basal epithelial cells. (**A**) Experiment design: CD8+ T cells have been either non-activated ('CD8$_{NA}$') or activated ('CD8$_A$') before being co-cultured with fibrocytes. Six days after fibrocytes co-culture, CD8+ T cells were transferred and cultured with primary bronchial basal epithelial cells for 6 hr. (**B, C**) Representative gating strategy for identification of CD8 T cells expressing CD107a without (w/o) fibrocytes (left panels) or with (w) fibrocytes (right panels). Histograms represent representative cell count (y-axis) versus CD107a fluorescence (x-axis) for total (**B**) and extracellular expression (**C**). (**D, E**) Comparison of quantifications of CD8+ T cells expressing the CD107a, removed from co-culture without fibrocyte ('CD8') and with fibrocyte ('CD8 +F') for total (**D**) and extracellular (**E**) expressions. (**F**) Representative brightfield images of primary basal epithelial cells co-cultured with CD8+ T cells for 6 hr, following their previous co-culture without (CD8) or with fibrocytes (CD8 +F) for 6 days. (**G**) Representative gating strategy for identification of apoptotic primary bronchial epithelial basal cells exposed to CD8+ T cells which were previously co-cultured without (CD8, left panel) or with fibrocytes (CD8 +F, right panel). Dot plots represent representative Propidium iodure fluorescence (y-axis) versus Annexin fluorescence (x-axis). (**H**) Comparison of quantifications of apoptotic primary bronchial epithelial basal cells exposed to CD8 T cells which were previously co-cultured without (CD8) or with fibrocytes (CD8 +F). (**D, E, H**) Medians are represented as horizontal lines. *p<0.05, Wilcoxon matched paired tests.

at the final state. Snapshots of the peribronchial area at the end of the simulations show that the densities of cells as well as their relative distribution were different between healthy and COPD situations (**Figure 8B**). From the simulations (n=160 in each condition), we found a median of 754 CD8+ T cells/mm$^2$ (95% CI, 748–763) and 106 fibrocytes/mm$^2$ (95% CI, 101–108) in the control situation, and 1187 CD8+ T cells/mm$^2$ (95% CI, 1169–1195) and 212 fibrocytes/mm$^2$ (95% CI, 206–216) in the COPD situation. These values are in very good agreement with our experimental findings, and the simulations were also able to reproduce the statistical increase of cell densities in COPD situations compared to that of controls (**Figure 8C**). Next, we tested if our theory accounted for the experimental relative distribution of CD8+ T cells and fibrocytes. The densities of CD8+ T cells in interaction with fibrocytes

(*Figure 8D*), the mean minimal distances between fibrocytes and CD8+ cells (*Figure 8E*), the distribution of mean minimal distances (*Figure 8—figure supplement 1*), and the mean number of mixed cell clusters (*Figure 8F*) were in good agreement with tissular analyses and mimicked the variations observed between control subjects and patients with COPD. The densities of mixed cell clusters predicted by simulations (control simulations:median = 17 clusters/mm² (95% CI, 18–21), COPD simulations:median = 45 clusters/mm² (95% CI, 46–51), p<0.001) agreed perfectly with experimental measurements (*Figure 8G*) and were, therefore, chosen as a readout of intercellular interactions in the following analyses. If purely random, the density of mixed clusters was expected to be 28 clusters/μm² (95% CI, 25–29) and 73 clusters/μm² (95% CI, 70–74) in control and COPD situations, respectively (*Figure 8—figure supplement 2*). These random densities as well as the other parameters quantifying the relative distribution of cells were statistically different from the distributions obtained in both simulations and in situ analyses (*Figure 8—figure supplement 2*). We conclude that the relative organization of CD8+ T cells and fibrocytes in control and COPD bronchi did not result from a pure stochastic mechanism but implicates chemotaxis processes.

One of the strengths of the model is to allow the monitoring of the temporal evolution of the different cellular processes and the numerical detection of a change of regime (*Figure 8H–I*). CD8+ T cells infiltration remained identical in control and COPD situation. Fibrocyte-induced T cell proliferation, that represents the minor part of the total proliferation in control situations, quickly increased in COPD situations over time to reach a plateau after approximately 4 years. As the basal proliferation of CD8+ T cells remained similar in healthy and diseased situations, the resulting total proliferation in CD8+ T cells over time was higher in the COPD situation compared to the control one. COPD dynamics also affected CD8+ T cell death, with a concomitant increase of T cell-induced death. In total, the net balance between the gain and loss of CD8+ T cells was around zero for control dynamics and strictly positive for COPD dynamics, explaining the increased CD8+ T cell density in COPD simulations. Fibrocytes infiltration remained very similar in control and COPD dynamics (*Figure 8I*). Fibrocytes death was initially lower in COPD simulations before increasing and reaching a stationary state after approximately 7 years, resulting in a net expansion of fibrocytes population in COPD bronchi after 20 years.

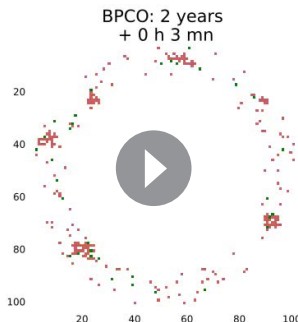

**Video 3.** Cell dynamics within the peribronchial area, 2 years after the initial time, with chronic obstructive pulmonary disease (COPD) dynamics. Images of the simulations were recorded every 3 min for 24 hr. CD8+ T cells and fibrocytes are represented, respectively, by pink and green squares. control (resp. COPD) situation.
https://elifesciences.org/articles/85875/figures#video3

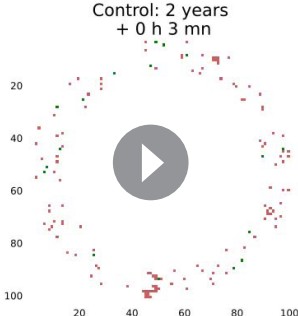

**Video 2.** Cell dynamics within the peribronchial area, 2 years after the initial time, with control dynamics. Images of the simulations were recorded every 3 min for 24 hr. CD8+ T cells and fibrocytes are represented, respectively, by pink and green squares. control (resp. COPD) situation.
https://elifesciences.org/articles/85875/figures#video2

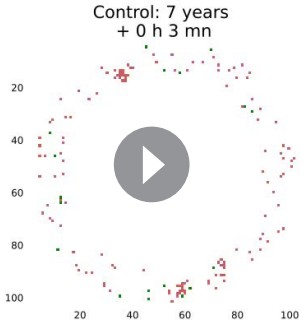

**Video 4.** Cell dynamics within the peribronchial area, 7 years after the initial time, with control dynamics. Images of the simulations were recorded every 3 min for 24 hr. CD8+ T cells and fibrocytes are represented, respectively, by pink and green squares. control (resp. COPD) situation.
https://elifesciences.org/articles/85875/figures#video4

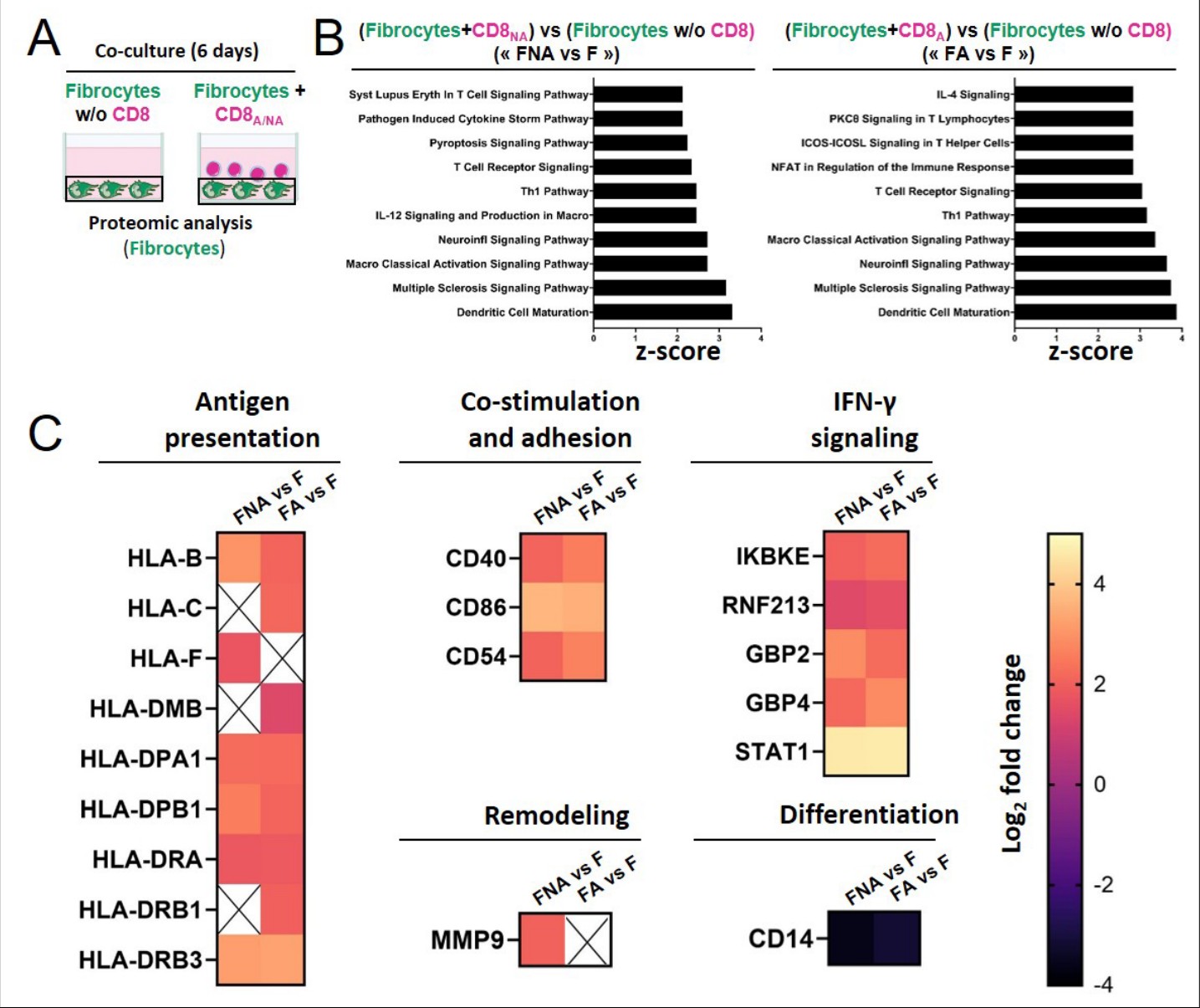

**Figure 7.** Direct contact between fibrocytes and CD8⁺ T cells favors the acquisition of fibrocyte immune properties. (**A**) Experiment design: fibrocytes have been either cultured alone, or with CD8⁺ T cells that have been previously non-activated ('CD8_NA') or activated ('CD8_A'). After 6 days of (co)-culture, fibrocyte proteins have been extracted for proteomic analyses. (**B**) Top 10 Canonical Ingenuity Pathways significantly altered in fibrocytes co-cultured with non-activated CD8⁺ T cells or activated CD8⁺ T cells vs fibrocytes cultured alone ('FNA versus F,' left graph, 'FA versus F,' right graph, respectively, n=4 for each condition), ranked by Z-score, obtained by Gene Set Enrichment Analysis. (**C**) Heatmaps of significantly differentially regulated proteins in FNA versus F and FA versus F, including proteins related to antigen presentation, co-stimulation and adhesion, remodeling, IFN-γ signaling, and differentiation. The color scale indicates the log₂ fold changes of abundance for each protein.

Moreover, the simulations allowed us to monitor the interactions between fibrocytes and CD8⁺ T cells. The density of mixed cell clusters gradually increased in the first years of the COPD simulation before reaching a stationary state after approximately 6 years (*Figure 8J*, *Video 2*, *Video 3*, *Video 4*, *Video 5*). Altogether, the theory of the influence of local interactions tested by our agent-based (cellular automata) model correctly accounts for the shift of the absolute and relative distribution of CD8⁺ T cells and fibrocytes in peribronchial areas from control subjects to patients with COPD.

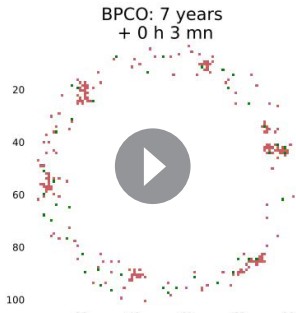

**Video 5.** Cell dynamics within the peribronchial area, 7 years after the initial time, with chronic obstructive pulmonary disease (COPD) dynamics. Images of the simulations were recorded every 3 min for 24 hr. CD8+ T cells and fibrocytes are represented, respectively, by pink and green squares. control (resp. COPD) situation.
https://elifesciences.org/articles/85875/figures#video5

## Simulations help to predict the outcomes of therapeutic strategies

We performed additional simulations to investigate the outcomes of possible therapeutic interventions. First, we applied COPD dynamics for 20 years, to generate the COPD states, that provide the basis for treatment implementation. Then, we applied COPD dynamics for 7 years, that mimics the placebo condition (*Figure 9A*), and we compared it to a control dynamics ('Total inhibition'), that mimics an ideal treatment able to restore all cellular processes. As expected the populations of fibrocytes and CD8+ T cells, as well as the density of mixed clusters, decreased. These numbers reached levels similar to healthy subjects after approximately 2.5 years, and this time point can, therefore, be considered as the steady state (*Figure 9B–E*). Monitoring of the different processes revealed that these effects were mainly due to a reduction in fibrocyte-induced CD8+ T duplication, and a transient or more prolonged increase in basal fibrocyte and CD8+ T death (*Figure 9C–D*). Then, three possible realistic treatments were considered (*Figure 9A*). We tested the effect of directly inhibiting the interaction between fibrocytes and CD8+ T cells by blocking CD54. This was implemented in the model by altering the increased probability of a CD8+ T cell to divide when a fibrocyte is in its neighborhood, as shown by the co-culture results (*Figure 4*). We also chose to reflect the effect of a dual CXCR1/2 inhibition by setting the displacement function of fibrocyte similar to that of control dynamics, in agreement with the in vitro experiments (*Figure 2E*). Blocking CD54 only slightly reduced the density of CD8+ T cells compared to the placebo condition, and had no effect on fibrocyte and mixed cluster densities (*Figure 9B*). CXCR1/2 inhibition was a little bit more potent in the reduction of CD8+ T cells than CD54 inhibition, and it also significantly decreased the density of mixed clusters (*Figure 9B*). As expected, this occurred through a reduction of fibrocyte-induced duplication, which was affected more strongly by CXCR1/2 blockage than by CD54 blockage (*Figure 9C–E*). Combining both therapies (CD54 and CXCR1/2 inhibition) did not strongly major the effects (*Figure 9B–E*). In all the conditions tested, the size of the fibrocyte population remained unchanged, suggesting that other processes such as fibrocyte death or infiltration should be targeted to expect broader effects.

## Discussion

The present study aimed at identifying the role and mechanism of fibrocyte-CD8+ T cells cross-talk in COPD. A previous study had pointed out a pivotal role for fibrocyte to activate CD8+ T cells proliferation (*Afroj et al., 2021*). However, whether and how both cell types could interact in bronchi, as well as their implication in COPD was completely unknown. Quantitative image analysis provided crucial insight into the relative distribution of fibrocytes and CD8+ T cells in distal bronchial specimens from control subjects and COPD patients. In addition to data from previous studies demonstrating that the densities of both fibrocytes (*Dupin et al., 2019*) and CD8+ T cells (*Saetta et al., 1999*) are increased within the distal bronchi of COPD patients, we found that fibrocyte and CD8+ T cells are localized in close proximity in peribronchial areas, especially in tissues from patients with COPD. We deciphered the spatiotemporal characteristics of these cell–cell contacts by live imaging in an in vitro autologous co-culture assay, and showed that the duration of the contacts was compatible with activation through the establishment of dynamic synapses. On the one hand, CD8+ T cells induced fibrocyte chemotaxis through CXCL8/CXCR1/2 axis and engagement towards immunologic signaling, and, on the other hand, fibrocytes directly induced CD8+ T cell proliferation, cytokine production, and cytotoxic activity against bronchial epithelial cells (*Figure 10*). The strength of our work relies on the integration of findings from the present in vitro experiments and other studies into a comprehensive computational

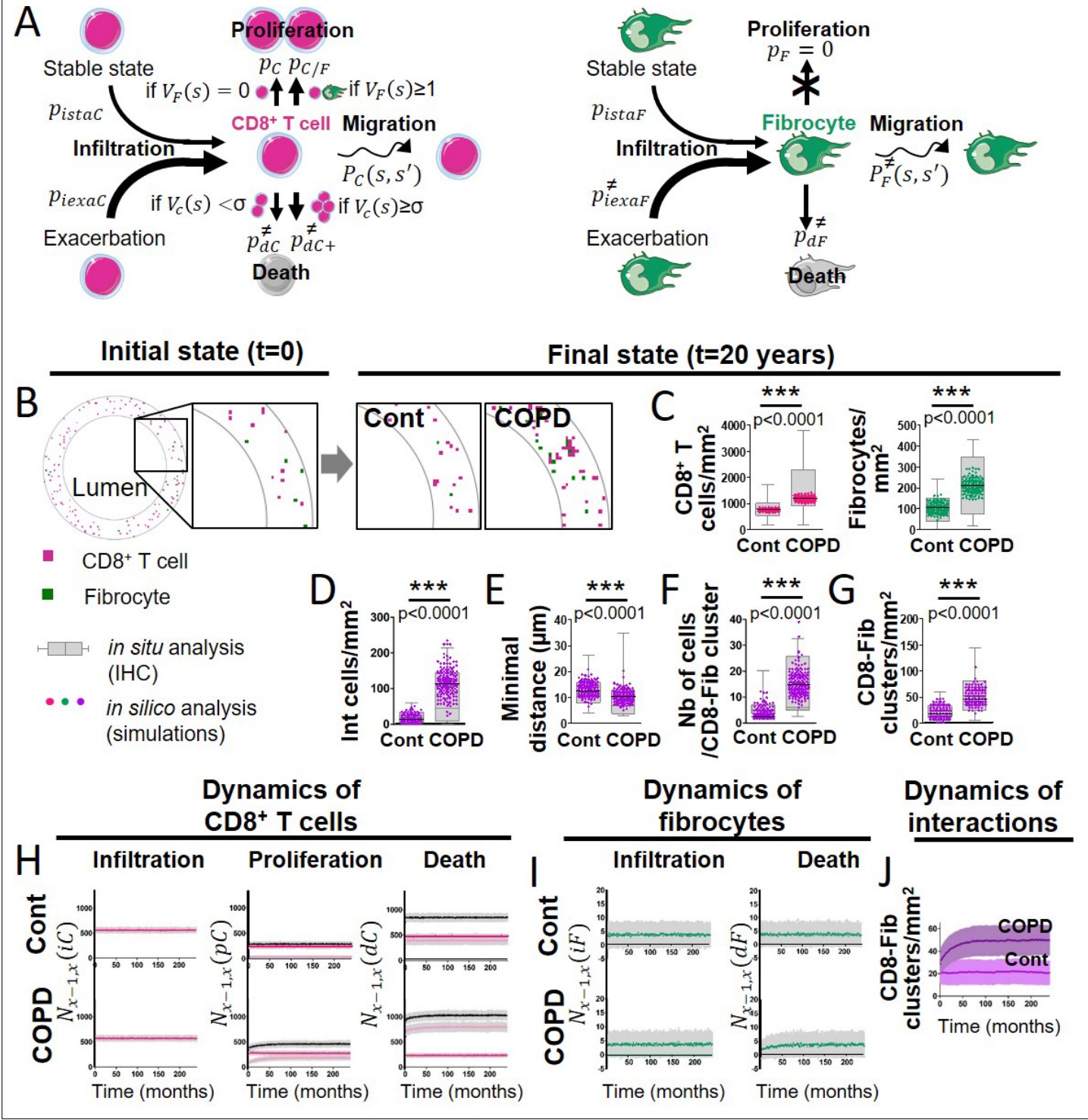

**Figure 8.** A probabilistic cellular automata-type model captures the features of the normal and pathological patterns of cell organization observed in the tissues. (**A**) Schematic representation of the probabilities associated with CD8+ T cells (left panel) and fibrocytes (right panel). For each CD8+ T cell, we define a 'basal' probability $p_{dC}$ of dying, an increased probability $p_{dC+}$ of dying when the CD8+ T cell has many other CD8+ T cells in its neighborhood, a 'basal' probability $p_C$ of dividing, an increased probability $p_{C/F}$ of dividing when the CD8+ T cell has fibrocytes in its neighborhood, a probability $P_C\left(s, s^{'}\right)$ of moving from a site s to a neighboring site s', a probability $p_{istaF}$ to be infiltrated at the stable state and a probability $p_{iexaC}$ to be infiltrated during exacerbation. For each fibrocyte, we define a probability $p_{dF}$ of dying, a probability $p_F$ of dividing, a probability $P_F\left(s, s^{'}\right)$ of moving from a site s to a neighboring site s', a probability $p_{istaF}$ to be infiltrated at the stable state and a probability $p_{iexaF}$ to be infiltrated

*Figure 8 continued on next page*

*Figure 8 continued*

during exacerbation. The ≠ symbol indicates parameters whose numerical value differs from control to COPD situation. (**B**) Selected representative pictures for initial state and final states after 20 years of control and COPD dynamics. Images surrounded by black squares: higher magnifications of peribronchial area. CD8+ T cells and fibrocytes are represented, respectively, by pink and green squares. (**C**) CD8+ T cells (left) and fibrocyte (right) densities. (**D**) Interacting cells densities of interacting cells. (**E**) Mean minimal distances between fibrocyte and CD8+ T cells. (**F**) CD8+ T cells-fibrocytes-containing clusters ('CD8-Fib clusters') densities. (**G**) mean number of cells by CD8-Fib clusters. (**C–G**), n=160 simulations for each situation. The medians are represented as horizontal lines. The equivalent measurements measured on the patient's tissues are represented by gray boxes (25th to the 75th percentile) and whiskers (min to max). ***p<0.001. unpaired t-tests or Mann-Whitney tests. (**H, I**) Mean kinetics of the populations of CD8+ T cells and fibrocytes in control and COPD situation in silico. Standard deviations are indicated in gray, n=160 simulations. Left panels: $N_{x-1,\ x}\left(iC\right)$ and $N_{x-1,\ x}\left(iF\right)$ are the number of CD8+ T cells (resp. fibrocytes) that have infiltrated the peribronchial area for the month $x$, relatively to the surface of interest. For fibrocytes, the infiltration at the stable state and during exacerbation are indicated, respectively, in green and light green. For the control situation, there is no infiltration by exacerbation. Midde panels: $N_{x-1,\ x}\left(pC\right)$ is the number of CD8+ T cells that have proliferated for the month $x$, relatively to the surface of interest. Basal duplication, fibrocyte-induced duplication and total duplication are indicated, respectively, in pink, light pink, and black. Right panels: $N_{x-1,\ x}\left(dC\right)$ and $N_{x-1,\ x}\left(dF\right)$ are the number of CD8+ T cells (resp. fibrocytes) that have died for the month $x$, relatively to the surface of interest. For CD8+ T cells, basal death, T cell-induced death, and total death are indicated, respectively, in pink, light pink, and black. (**J**) Graphs showing the variations of the mean densities of CD8-Fib clusters over time in control (light purple) and COPD situation (dark purple).

The online version of this article includes the following figure supplement(s) for figure 8:

**Figure supplement 1.** Minimal intercellular distance distributions are similar between in situ analyses and simulations.

**Figure supplement 2.** Spatial cellular repartition of cells obtained at the final state of simulations are distinct from random distributions.

model that provides an accurate prediction of histological ex vivo characteristics and the possibility to figure out the in vivo effect of drugs. Altogether, our data suggest a pivotal role for fibrocytes to activate CD8+ T cell deleterious functions in the context of COPD.

We analyzed the relationship between these histological parameters and clinical data and found associations between fibrocyte presence, fibrocyte-CD8+ T cell interaction and the alteration of lung function. We have demonstrated using stepwise and multivariate regressions that the density of inter-acting cells and the density of mixed cell clusters were the two best-correlated parameters with the FEV$_1$/FVC ratio, supporting a potential role for the interplay between both cell types in COPD. Since regions of microscopic emphysematous destruction of terminal bronchioles have been associated with increased infiltration of CD8+ T cells and immune response activation, such as the up-regulation of IFN-γ signaling (*Xu et al., 2022*), and we have evidenced fibrocyte-mediated cytotoxic activity in CD8+ T cells, it is tempting to speculate that fibrocyte-CD8+ T cell interplay could be implicated in early changes leading to tissue remodeling and chronic inflammation in COPD. Of note, the gene signature obtained by tissue microarray associated with this site also indicates the modification of two genes associated with the tissue repair process, FGF10 and TGFB2 (*Xu et al., 2022*). Considering the possible effect of CD8+ T cells on fibrocyte differentiation, it could be worthwhile to focus on these genes in further studies.

We also addressed the potential mechanisms explaining these increased interactions of CD8+ T cells and fibrocytes in the tissues of COPD patients. Chemotaxis could guide CD8+ T cells towards fibrocytes and reciprocally, as it has been proposed for T cells towards dendritic cells (*Mackay, 2001*; *Ngo et al., 1998*; *von Andrian and Mackay, 2000*). Stronger or longer interactions could also explain the differential spatial distribution between healthy and diseased tissues. On the other hand, the contact between both cell types could also occur through a stochastic mechanism, as shown for CD4+ T cells and dendritic cells in lymph nodes, without any implication of chemotactic processes (*Miller et al., 2004*). Although we cannot totally rule out a role for fibrocyte-CD8+ T cell adhesion to explain the increased interactions, our findings rather suggest a central role for the CXCL8-CXCR1/2 axis in promoting encounters between CD8+ T cells and fibrocytes in COPD patients. Importantly, this is further supported by the results of computational modelization, which only integrates chemotaxis and not adhesion processes, revealing a final spatial repartition of cells in the COPD situation distinct from a random distribution. Altogether, our data suggest that the likelihood of interactions between fibrocytes and CD8+ T cells could be increased in tissues from patients with COPD through the CXCL8-CXCR1/2 axis thus participating in cluster fibrocytes and CD8+ T cells in diseased tissues. Importantly, dual blockers of CXCR1-CXCR2 have been suggested as therapeutic targets in COPD (*Henrot et al., 2019*). Although reparixin, a dual blocker of CXCR1-CXCR2, was efficient in our in vitro experiments to block the increased chemotaxis of fibrocytes towards secretion of COPD CD8+ T cells, the in vivo

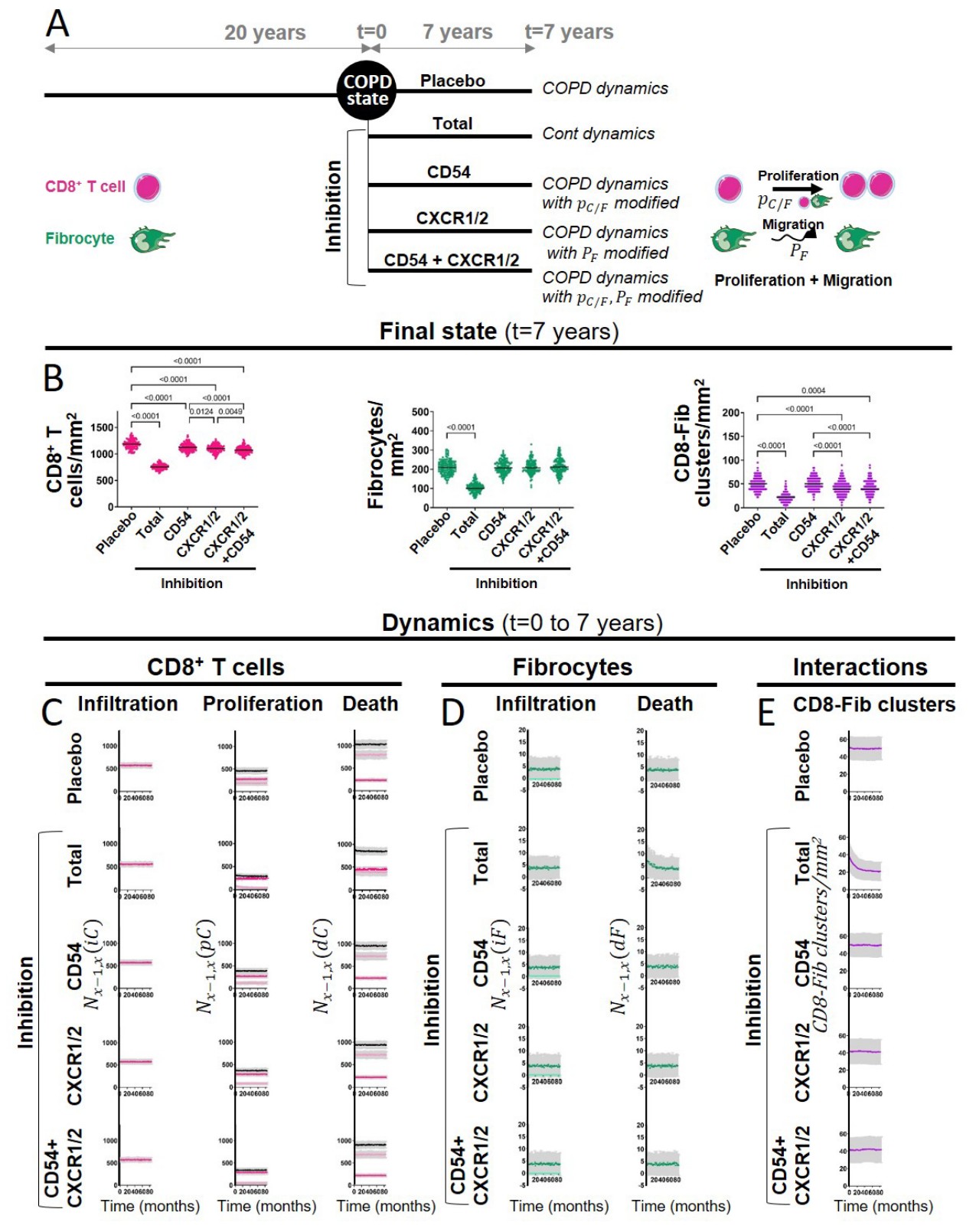

**Figure 9.** The outcomes of therapeutic interventions are predicted by simulations. (**A**) Schematic representation of the design used to test therapeutic strategies. Chronic obstructive pulmonary disease (COPD) states were first generated by applying COPD dynamics for 20 years (n=144 simulations). Then, different dynamics were applied for 7 years: COPD dynamics (corresponding to the placebo condition), control dynamics (corresponding to an ideal treatment able to restore all cellular processes, 'Total inhibition'), and modified COPD dynamics (corresponding to CD54, CXCR1/2, and dual

*Figure 9 continued*

inhibitions with alterations of the probability $p_{C/F}$ of dividing when the CD8[+] T cell has fibrocytes in its neighborhood, the probability $P_F$ for a fibrocyte to move, and both, respectively). (**B**) CD8[+] T cells (left), fibrocyte (middle), and CD8[+] T cells-fibrocytes-containing clusters ('CD8-Fib clusters,' right) densities at the final state (t=7 years). The medians are represented as horizontal lines. One-way ANOVA with Tukey's post-tests. p-values below 0.05 with Tukey's post-tests are indicated on the graphs, except for the comparisons between the condition 'Total inhibition' and the other conditions of inhibition, that are not indicated. (**C–D**) Mean kinetics of the populations of CD8[+] T cells and fibrocytes in the different conditions (t=0–7 years). Left panels: $N_{x-1,\ x}\left(iC\right)$ and $N_{x-1,\ x}\left(iF\right)$ are the number of CD8[+] T cells (resp. fibrocytes) that have infiltrated the peribronchial area for the month $x$, relatively to the surface of interest. For fibrocytes, the infiltration at the stable state and during exacerbation are indicated, respectively, in green and light green. For the total inhibition situation, there is no infiltration by exacerbation. Midde panels: $N_{x-1,\ x}\left(pC\right)$ is the number of CD8[+] T cells that have proliferated for the month $x$, relatively to the surface of interest. Basal duplication, fibrocyte-induced duplication, and total duplication are indicated, respectively, in pink, light pink, and black. Right panels: $N_{x-1,\ x}\left(dC\right)$ and $N_{x-1,\ x}\left(dF\right)$ are the number of CD8[+] T cells (resp. fibrocytes) that have died for the month $x$, relatively to the surface of interest. For CD8[+] T cells, basal death, T cell-induced death, and total death are indicated, respectively, in pink, light pink, and black. (**E**) Mean kinetics of the densities of CD8-Fib clusters in the different conditions (t=0–7 years). (**C–E**) Standard deviations are indicated in gray, n=144 simulations.

effect of this therapy predicted using our computational model was moderate, highlighting the importance of this integrated approach.

We show that fibrocytes act as a major promoter of CD8[+] T cell proliferation, thus confirming, in an autologous co-culture system, what has been previously found in the context of cancer-related immunity (*Afroj et al., 2021*). This is consistent with the present in situ analyses, showing the presence of clusters containing both cell types in the peribronchial area, especially in the tissues of patients with COPD. The mean numbers of cells in those clusters remained relatively low, suggesting that these structures are distinct from inducible bronchus-associated lymphoid tissue (iBALT) (*Conlon et al., 2020*). Although a previous report has demonstrated that fibrocytes, exposed to viral antigens, could induce the proliferation of naïve CD8[+] T cells (*Balmelli et al., 2005*), the pro-proliferative effect exerted by fibrocytes on CD8[+] T cells occurred without antigen exposure in our in vitro study. This antigen-independent T cell proliferation driven by fibrocytes was also found in the context of sepsis (*Nemzek et al., 2013*), suggesting that fibrocytes generally impact T cells expansion with a mechanism independent of the traditional antigen-driven clonal proliferation. This is also in agreement with our findings showing that contacts between CD8[+] T cells and fibrocytes were relatively short and dynamic, and that the dynamics of the interaction did not depend on the activation state of CD8[+] T cells. The spatiotemporal behavior of CD8[+] T cells was consistent with the establishment of dynamic synapse, also called 'kinapse' (*Dustin, 2008*), which is associated with the induction of relatively weak TCR signals (*Moreau et al., 2012*). We have evidenced the requirement for cellular contacts, implicating the surface receptors CD86 and CD54. The lack of effect of the anti-CD86 and CD54 in preactivated CD8[+] T cells might indicate potential changes in the expression of molecules belonging to the immunological synapse upon activation, that could make the lymphocytes more responsive to other signals. The well-known inhaled glucocorticoids (*i.e.* budesonide and fluticasone propionate) also failed to significantly inhibit fibrocyte-induced CD8[+] T cell proliferation. This is consistent with their lack of activity in lymphocytes obtained from patients with COPD (*Kaur et al., 2012*). In contrast, we propose that targeting the interaction between structural and immune cells and/or its consequences should reveal robust candidates for future pharmacotherapeutic strategies to treat COPD. Of note, the stimulatory activity of CD8[+] T cells by fibrocytes was also found to be enhanced by the blockade of the PD-1/PD-L1 pathway in a previous study (*Afroj et al., 2021*). As this latter property of fibrocytes may be beneficial in tumor microenvironment (*Henrot et al., 2021*), especially when cancer patients were treated with anti–PD-1/PD-L1 antibody, it might be rather detrimental in COPD patients, by promoting tissue damages and chronic immune inflammation.

Fibrocytes skewed CD8[+] T-cell populations towards both CD8[high] and CD8[low] phenotypes in a cell–cell contact-independent manner. It has been described that, following contact between an antigen-presenting cell and a lymphocyte, asymmetric division can occur generating a memory cell, weakly expressing CD8, and an effector cell strongly expressing CD8 (*Backer et al., 2018*; *Chang et al., 2007*). The asymmetry is reduced but still present even without specific recognition of foreign antigens by T cells (*Chang et al., 2007*). It is tempting to speculate that the induced proliferation we observed in our experiments generates, via asymmetric division, unequal CD8 inheritance in daughter cells. The low level of cytokine expression in CD8[low] cells is compatible with an exhausted phenotype, while CD8[high] cells express higher levels of cytokines, a profile consistent with an effector commitment.

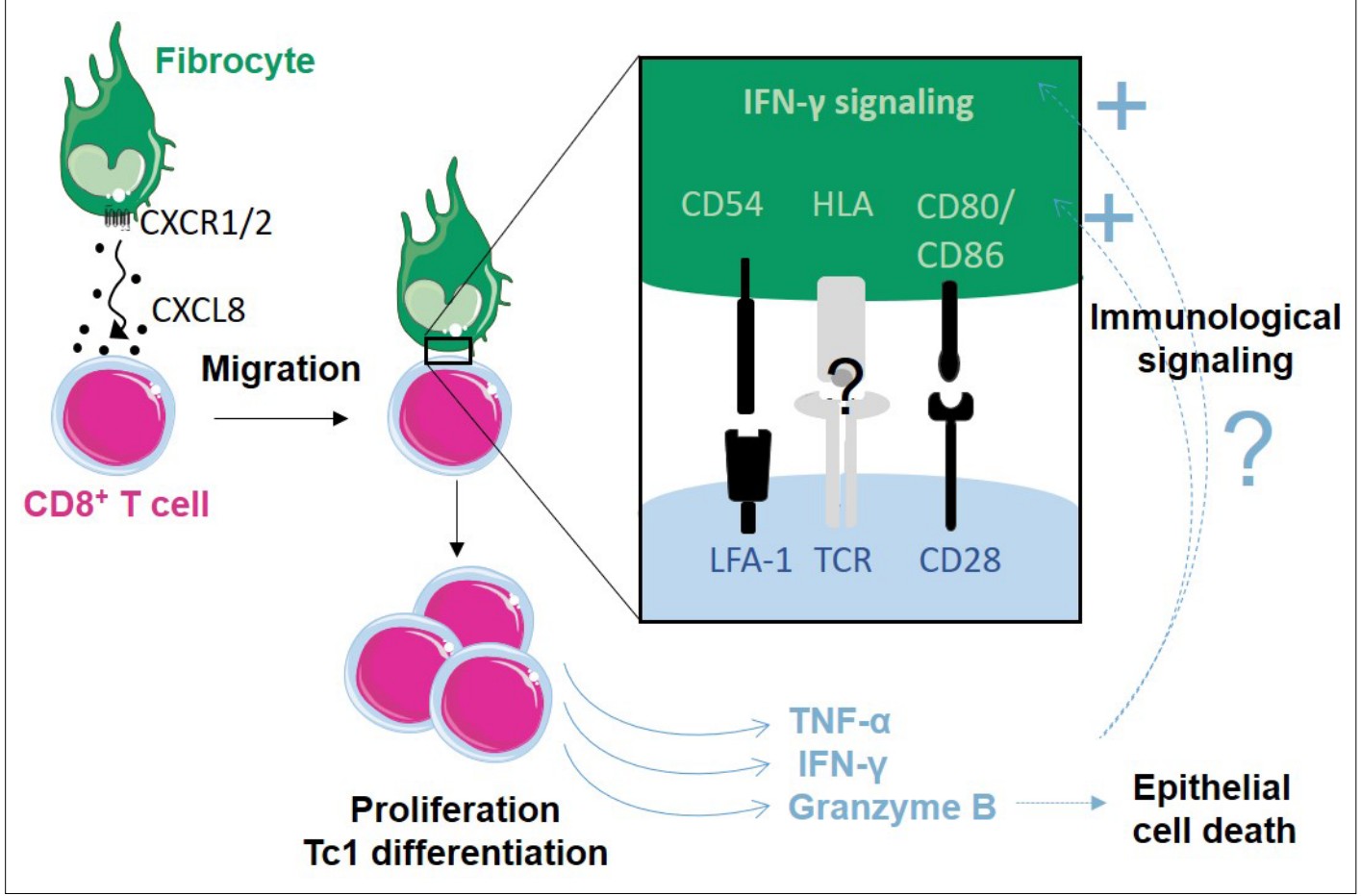

**Figure 10.** Proposed model of how fibrocytes interact with CD8+ T cells in the context of chronic obstructive pulmonary disease (COPD). Fibrocyte chemotaxis towards CD8+ T cells is mainly due to an increased CXCL8 secretion by CD8+ T cells in COPD lungs, and promotes direct contact between both cell types. This interaction triggers CD8+ T cell proliferation, cytokine production, and cytotoxic activity. The interaction and its consequences might be further increased by a reinforcement of IFN-γ signaling and expression of molecules belonging to the immune synapse, from the fibrocyte side.

Although the presence of the CD8high and CD8low subsets remains to be evidenced in the tissues, we suspect that the relative expansion of the CD8high and CD8low subset triggered by fibrocytes could have functional implications. Reiterative rounds of CD8+ T cells division induced by frequent interactions with fibrocytes might induce defective immune response by exhausted CD8low T cells (*Grundy et al., 2013*; *McKendry et al., 2016*), and tissue destruction by cytotoxic CD8high cells (*Chrysofakis et al., 2004*; *Maeno et al., 2007*).

In COPD, outside of exacerbations, factors triggering pro-inflammatory cytokines production are still elusive. Here, we demonstrate that fibrocytes exert a strong effect through soluble factors and direct cellular contacts with CD8+ T cells, inducing a massive upregulation of TNF-α, IFN-γ, and granzyme B production, all implicated in COPD pathophysiology (*Barnes, 2016*). Greater production of TNF-α, IFN-γ, and granzyme B by CD8+ T cells triggered by the interaction with fibrocytes is consistent with previous studies showing enhanced production of Tc1 cytokines and cytotoxic molecules by CD8+ T cells purified from patients with COPD (*Freeman et al., 2010*; *Hodge et al., 2007*; *Lethbridge et al., 2010*), suggesting that local interactions with cells such as fibrocytes may play a pivotal role in CD8 polarization in COPD. In particular, TNF-α has proinflammatory and prooxidative actions (*Mukhopadhyay et al., 2006*), and its overexpression has been associated with emphysema (*Lundblad et al., 2005*). TNF-α can directly contribute to cytolysis, together with the cytotoxic granzyme B (*Velotti et al., 2020*) and in synergy with IFN-γ (*Williamson et al., 1983*). TNF-α can also indirectly participate in extracellular matrix degradation through the induction of matrix metalloproteinases

(*Wright et al., 2007*). Simultaneously, the production of the pro-fibrotic IL-17 was also induced upon co-culture with fibrocytes, raising the possibility that the interaction between CD8[+] T cells and fibrocyte participates in the generation of IL-17-secreting CD8[+] T cells in airways of patients with COPD (*Chang et al., 2011*). Interestingly, IL-17 is able to simulate matrix component synthesis in other cell types, including fibrocytes, and promotes CD40-mediated IL-6 production by fibrocytes (*Hayashi et al., 2013*). Up-regulated pathways identified in the proteomic profile of fibrocytes co-cultured with CD8[+] T cells are very consistent with a shift towards a proinflammatory phenotype rather than towards a reparative role. The activation of IFN-γ signaling could be triggered by CD8[+] T cell secretion of IFN upon fibrocyte interaction, suggesting the existence of a positive feedback loop (*Figure 10*). Additionally, the priming of fibrocytes by CD8[+] T cells could also induce CD4[+] T cell activation. Cooperative interactions between fibrocytes and CD8[+] T cells, through tissue destruction and abnormal inflammation, may thus directly contribute to the loss of normal lung function. On the other hand, CD8[+] T cell production of anti-inflammatory cytokines such as IL-10, was also stimulated upon co-culture with fibrocytes. In total, rather than the net production of each cytokine, it is probably the balance or imbalance between pro-inflammatory and anti-inflammatory molecules that will dictate the outcome of the inflammatory process.

Whereas the field of respiratory research is rapidly moving towards an exhaustive description of modifications of molecular and cellular components in diseased lungs, the actual transition between a healthy to a diseased state, although critical, remains very difficult to investigate. We developed here a probabilistic cellular automata-type model to explore dynamic behaviors and interactions between fibrocytes and CD8[+] T cells. Previous agent-based computational approaches have been used to describe the switch from normal to allergic response (*Pothen et al., 2015*) and airway remodeling in asthma (*Saunders et al., 2019*), but, to our knowledge, this type of modeling was never applied to COPD. Qualitative estimates of probabilities that govern cell death, proliferation, infiltration, and displacement are derived from experimental data from our study and others. We could simulate spatiotemporal behaviors of cells in the lamina propria over a long period of time (*i.e.* 20 years) and we showed that this model can accurately reproduce the absolute and relative repartition of fibrocytes and CD8[+] T cells in both control and COPD situations.

Although simulated and in situ data were close, the variances of in silico data were smaller than the in situ measurements, which can be probably explained by the fact that cell diversity and interactions are far more complex than those considered in this model. Nevertheless, it appears that (i) our model captures important aspects of reality, and (ii) modifications of specific cellular processes and local interactions, *i.e.*, fibrocyte-induced CD8[+] T cell proliferation and fibrocyte attraction towards CD8[+] T cells, are sufficient to reproduce the shift of histological composition between the control and COPD situations. This theoretical approach and associated simulations allowed us to validate the key hypothesis of modification of local interactions, and to show that the specific values of the COPD parameters led to an increased cell density and the spatial patterns observed in patients with COPD. The simulations made it possible to follow over time various quantities of interest and to empirically determine the time when the stationary state is achieved, that would be difficult to reveal in any other way. Given the consistency of our results with those from the literature, our model provides a unique opportunity to decipher the dynamics of increased interactions between the two cell types as well as the infinite possibility to investigate therapeutic strategies. Using the simulations, we were indeed able to estimate the characteristic time to reach a stationary state reminiscent of a resolution of the COPD condition. This time of approximately 2.5 years was totally unpredictable by in vitro experiments, and indicates that a treatment aiming at restoring these cellular processes should be continued for several years to obtain significant changes.

Regarding potential therapeutic strategies, chemokine-receptor based therapies could be used to inhibit fibrocyte recruitment into the lungs, such as CXCR4 blockade. We have very recently shown that using the CXCR4 antagonist, plerixafor, alleviates bronchial obstruction, and reduces peri-bronchial fibrocytes density (*Dupin et al., 2023*). Because CXCR4 expression in human fibrocytes is dependent on mTOR signaling and is inhibited by rapamycin in vitro (*Mehrad et al., 2009*), alternative strategies consisting of targeting fibrocytes via mTOR have been proposed. This target has proven effective in bronchiolitis obliterans, idiopathic pulmonary fibrosis, and thyroid-associated ophthalmopathy, using rapamycin (*Gillen et al., 2013*; *Mehrad et al., 2009*), sirolimus (*Gomez-Manjarres et al., 2023*) or an insulin-like growth factor-1 (IGF-I) receptor blocking antibody (*Douglas et al., 2020*; *Smith et al.,*

2017). Inhibiting mTOR is also expected to have effects on CD8+ T cells, ranging from an immuno-stimulatory effect by activation of memory CD8+ T-cell formation, to an immunosuppressive effect by inhibition of T cell proliferation (*Araki et al., 2010*). Last, chemokine-receptor base therapies could also include strategies to inhibit the CD8+-induced fibrocyte chemotaxis, such as dual CXCR1-CXCR2 blockade, which we were able to test in our mathematical model. Immunotherapies directly targeting the interaction between fibrocytes and CD8+ T cells could also be considered, such as CD86 or CD54 blockade. The use of abatacept and belatacept, that interfere with T cell co-stimulation, is effective in patients with rheumatoid arthritis (*Pombo-Suarez and Gomez-Reino, 2019*) and in kidney-transplant recipients (*Vincenti, 2016*), respectively. Targeting the IGF-I receptor by teprotumumab in the context of thyroid-associated ophthalmopathy also improved disease outcomes, possibly by altering fibrocyte-T cell interactions (*Bucala, 2022*; *Fernando et al., 2021*). Of note, the outcomes of CXCR1/2 and CD54 blocking strategy for COPD treatment were tested by our simulations, with limited beneficial effects. It suggests that such treatments may be more effective when used in combination with other drugs *e.g.*, those affecting fibrocyte infiltration and/or death. Such therapies should be used with caution as they may favour adverse events such as infections, particularly in the COPD population (*Rozelle and Genovese, 2007*). Additionally, the fibrocytes-lymphocytes interaction has recently been shown to promote anti-tumoral immunity via the PD1-PDL1 immunological synapse (*Afroj et al., 2021*; *Mitsuhashi et al., 2023*). Therefore, care should be taken in the selection of patients to be treated and/or timing of treatment administration with regards to the increased risk of lung cancer in COPD patients.

The present in vitro model has limitations, including the use of circulating cells for some in vitro experiments, that were obtained exclusively from patients with COPD. In particular, it was not possible to test the hypothesis of a different baseline activation state in the blood of patients with COPD in comparison with healthy subjects, that could participate to initiate or maintain the vicious cycle of inflammation. One should also mention the difficulty in extrapolating results obtained from these in vitro assays to in vivo processes. However, we took this limit into account in our modelization approach, by using a combination of our experiments and measurements obtained in tissues, to accurately determine the dedicated parameters (*Dupin et al., 2023*). Even if computational modelization was done in 2D, whereas the bronchi are 3D structures, we believe that our model is representative as it mimics the cellular distribution of normal and pathological airways, that was also quantified in 2D lung sections. Besides this, some quantitative features of our approach are still valid in 3D, such as the probabilities that govern cell death, proliferation, and infiltration, whereas others are expected to change with dimensionality, such as displacement rules.

From our study and others (*Hufford et al., 2011*; *Takamura et al., 2019*), it is now clear that the fate of CD8+ T cells in distal airways may depend on multiple successive interactions with different cell types, including fibrocytes. We believe that targeting the interaction between structural and immune cells should be considered in future drug discovery programs and that computational modelization should help to refine drug priority.

## Materials and methods

**Key resources table**

| Reagent type (species) or resource | Designation | Source or reference | Identifiers | Additional information |
|---|---|---|---|---|
| Antibody | Anti-CD8 (rabbit monoclonal) | Fisher Scientific | Cat. #:MA5-14548, RRID:AB_10984334 | IHC (1:200) |
| Antibody | Anti-rabbit-HRP (goat polyclonal) | Nichirei Biosciences | Cat. #:414141 F, RRID:N/A | IHC (1:200) |
| Antibody | Anti-CD45 (mouse monoclonal) | BD Biosciences | Cat. #:555480, RRID: AB_395872 | IHC (1:50) |
| Antibody | Anti-FSP1 (rabbit polyclonal) | Agilent | Cat. #:A5114, RRID: AB_2335679 | IHC (1:200) |
| Antibody | Anti-mouse-Alexa568 (goat polyclonal) | Fisher Scientific | Cat. #:A-11004, RRID:AB_2534072 | IHC (1:50) |
| Antibody | Anti-rabbit-Alexa488 (goat polyclonal) | Fisher Scientific | Cat. #:A-11008, RRID:AB_143165 | IHC (1:200) |

*Continued on next page*

*Continued*

| Reagent type (species) or resource | Designation | Source or reference | Identifiers | Additional information |
|---|---|---|---|---|
| Antibody | Anti-LFA-1 (mouse monoclonal) | BioLegend | Cat. #:301233, RRID:AB_2832576 | Blocking experiment (1 μg/mL) |
| Antibody | Anti-CD54 (mouse monoclonal) | Fisher Scientific | Cat. #:15247027, RRID:N/A | Blocking experiment (10 μg/mL) |
| Antibody | Anti-CD86 (mouse monoclonal) | Fisher Scientific | Cat. #:15297097, RRID:N/A | Blocking experiment (10 μg/mL) |
| Antibody | Anti-CD44 (rabbit monoclonal) | Fisher Scientific | Cat. #:15266957, RRID:N/A | Blocking experiment (10 μg/mL) |
| Antibody | Anti-CXCL8 (mouse monoclonal) | BioTechne | Cat. #:MAB208-100, RRID:N/A | Blocking experiment (1 μg/mL) |
| Antibody | Anti-CD4-PerCP-Vio700 (human recombinant monoclonal) | Miltenyi Biotec | Cat. #:130-113-228, RRID:AB_2726039 | FC (1:50) |
| Antibody | Anti-CD8-PerCP-Vio700 (human recombinant monoclonal) | Miltenyi Biotec | Cat. #:130-110-682, RRID:AB_2659249 | FC (1:50) |
| Antibody | Anti-CD45RA-FITC (human recombinant monoclonal) | Miltenyi Biotec | Cat. #:130-113-365, RRID:AB_2726135 | FC (1:50) |
| Antibody | Anti-granzyme-APC (human recombinant monoclonal) | Miltenyi Biotec | Cat. #:130-099-780, RRID:AB_2651900 | FC (1:20) |
| Antibody | Anti-TNF-α-PE (human recombinant monoclonal) | Miltenyi Biotec | Cat. #:130-110-066, RRID:AB_2654213 | FC (1:20) |
| Antibody | Anti IFN-γ-APC (human recombinant monoclonal) | Miltenyi Biotec | Cat. #:130-113-496, RRID:AB_2751119 | FC (1:20) |
| Antibody | Anti-IL-17-PE-Cy7 (mouse monoclonal) | Miltenyi Biotec | Cat. #:130-120-413, RRID:AB_2752086 | FC (1:20) |
| Antibody | Anti-IL-10-PE (human recombinant monoclonal) | Miltenyi Biotec | Cat. #:130-112-728, RRID:AB_2652318 | FC (1:20) |
| Antibody | Anti-Collagen Type I-FITC (mouse monoclonal) | Sigma Aldrich | Cat. #:FCMAB412F, RRID:AB_11204160 | FC (1:50) |
| Antibody | Anti-CD45-APC (mouse monoclonal) | BD Pharmingen | Cat. #:555485, RRID:AB_398600 | FC (1:10) |
| Antibody | Anti-CXCR1-PE (human recombinant monoclonal) | Miltenyi Biotec | Cat. #:130-115-879, RRID:AB_2727234 | FC (1:50) |
| Antibody | Anti-CXCR2-APC-Cy7 (human recombinant monoclonal) | Miltenyi Biotec | Cat. #:130-119-571, RRID:AB_2733103 | FC (1:50) |

## Study populations

Lung tissues for the in situ study were obtained from a previously described cohort (*Dupin et al., 2019*). The study was registered at ClinicalTrials.gov with the identifier NCT01692444 ('Fibrochir' study). The study protocol was approved by the research ethics committee ('CPP') and the French National Agency for Medicines and Health Products Safety ('ANSM'). Briefly, subjects more than 40 years of age were eligible for enrolment if they required thoracic lobectomy surgery for cancer (pN0), lung transplantation, or lung volume reduction. A total of 17 COPD patients with a clinical diagnosis of COPD according to the GOLD guidelines (*Global Initiative for Chronic Obstructive Lung Disease, 2023*) and 25 non-COPD subjects ('control subjects') with normal lung function testing (*i.e.* $FEV_1$/FVC>0.70) and no chronic symptoms (cough or expectoration) were recruited from the University Hospital of Bordeaux. Due to low quality of some tissue sections, distal bronchial identification or fibrocyte and CD8+ T cell quantification was impossible in five control specimens and 5 COPD specimens, which were excluded from the in situ analysis.

Lung tissues for the purification of tissular CD8+ T cells and basal bronchial epithelial cells were obtained from a separate cohort of patients requiring thoracic lobectomy surgery for nodule or cancer (pN0) (*i.e.* TUBE, sponsored by the University Hospital of Bordeaux, which includes its own local

ethic committee (CHUBX 2020/54)). According to the French law and the MR004 regulation, patients received an information form, allowing them to refuse the use of their surgical samples for research. For the purification of tissular CD8+ T cells, a total of 20 patients with COPD and 26 nonsmokers were prospectively recruited from the University Hospital of Bordeaux, according to the GOLD guidelines (*Global Initiative for Chronic Obstructive Lung Disease*, s. d.) (*Supplementary file 7*). For the purification of basal bronchial epithelial cells, a total of two patients were prospectively recruited from the University Hospital of Bordeaux (*Supplementary file 9*).

To study fibrocyte- CD8+ T cells interplay in vitro, blood samples were obtained from a separate cohort of COPD patients, (*i.e.* COBRA (Bronchial Obstruction and Asthma Cohort; sponsored by the French National Institute of Health and Medical Research, INSERM, Ethics committee number: 2008-A00294-51/1)), as outpatients in the Clinical Investigation Centre of the University Hospital of Bordeaux (*Supplementary file 8*).

All subjects gave their written informed consent to participate in the studies. The studies received approval from the local or national ethics committees.

## Identification of bronchial fibrocytes and CD8+ T cells

Fragments of distal parenchyma were obtained from macroscopically normal lung resection or transplantation material. The samples were embedded in paraffin and sections of 2.5 μm thick were cut, as described previously (*Dupin et al., 2019*). Sections were deparaffinized through three changes of xylene and through graded alcohols to water. Heat-induced antigen retrieval was performed using citrate buffer, pH 6 (Fisher Scientific, Illkirch, France) at 96 °C in a Pre-Treatment Module (Agilent, Les Ulis, France). Endogenous peroxidases were blocked for 10 min using hydrogen peroxide treatment (Agilent). Nonspecific binding was minimized by incubating the sections with 4% Goat Serum (Agilent) for 30 min, before CD8 staining, and before the double staining for CD45 and FSP1. First, the sections were stained with rabbit anti-CD8 monoclonal antibody (Fisher Scientific) for 45 min, and then incubated with HRP anti-Rabbit (Nichirei Biosciences). Immunoreactivity was detected by using the DAB System (Agilent). Second, the same sections were stained with mouse anti-CD45 monoclonal antibody (BD Biosciences, San Jose, CA) overnight and then with rabbit anti-FSP1 polyclonal antibody (Agilent) for 45 min. They were incubated with Alexa568–conjugated anti-Mouse and with Alexa488–conjugated anti-Rabbit (Fisher Scientific) antibodies. Immunoreactivity was detected by fluorescence for FSP1 and CD45 staining.

The sections were imaged using a slide scanner Nanozoomer 2.0HT with fluorescence imaging module (Hamamatsu Photonics, Massy, France) using objective UPS APO 20 X NA 0.75 combined to an additional lens 1.75 X, leading to a final magnification of 35 X. Virtual slides were acquired with a TDI-3CCD camera. Fluorescent acquisitions were done with a mercury lamp (LX2000 200 W - Hamamatsu Photonics) and the set of filters adapted for DAPI, Alexa 488, and Alexa 568. Brightfield and fluorescence images were acquired with the NDP-scan software (Hamamatsu) and processed with ImageJ.

Quantification of CD8+ T cells was performed, as described in *Figure 1—figure supplement 1A, C*. A color deconvolution plugin was used on brightfield images to isolate the signal corresponding to DAB staining. A binary threshold was applied to this grayscale image, followed by a watershed transformation to the segmented image to separate potential neighboring cells (*Figure 1—figure supplement 1C*). CD8+ T cells were then automatically counted by recording all the positive particles with an area greater than 64 μm$^2$. This threshold was empirically determined on our images to select positive cells. Quantification of dual positive cells for FSP1 and CD45 was performed, as described in *Figure 1B and D*. A binary threshold was applied to fluorescence images corresponding to FSP1 and CD45 stainings. These images were combined using the 'AND' function of the Fiji 'Image Calculator' to select cells dual positive for FSP1 and CD45 double staining (*Figure 1—figure supplement 1D*). This was followed by a watershed transformation to separate potential neighboring cells. These CD45+ FSP1+ cells were then automatically counted by recording all the positive particles with an area greater than 64 μm$^2$.

## Quantification of the density of CD8+ T cells, FSP1+ CD45+ cells, and CD8+ T cells in interaction with CD45+ FSP1+ cells

This latter segmented image was then used to quantify CD8+ T cells in interaction with CD45+ FSP1+ cells as described in *Figure 1E*: each CD8 positive particle with an area greater than 64 µm² was enlarged using the dilatation function (4, 8, 10, and 15 pixels dilatation: used to count the cells, respectively, less than 1.8, 3.6, 4.5, and 6.8 µm apart). This modified image was combined with the segmented image for dual CD45 FSP1 positive staining using the 'AND' function of the Fiji 'Image Calculator' to select CD8+ T cells in interaction with CD45+ FSP1+ cells. These interacting cells were automatically counted by recording all the positive particles. The lamina propria contour was manually determined on a brightfield image and the area was calculated. For distal bronchi, the lumen area was also determined and only bronchi less than 2 mm in diameter were analyzed as described previously (*Hogg et al., 2004*). The densities of CD8+ T cells, FSP1+ CD45+ cells, and interacting cells were defined by the ratio between the number of positive cells in the lamina propria divided by the lamina propria area. Tissue area and cell measurements were all performed in a blinded fashion for patients' characteristics.

## Quantification of the minimal distances between CD45+ FSP1+ cells and CD8+ T cells

The segmented image produced from the DAB staining image was inverted, and a CD8 distance map was built from the latter image (*Figure 1—figure supplement 1F*). As a result, the brighter the pixel, the closer the distance from a CD8+ T cell. Conversely, the darker the pixel, the farther away the distance from a CD8+ T cell. On the binary image produced from FSP1 and CD45 staining images, dual positive cells for FSP1 and CD45 were selected in the lamina propria. Each area corresponding to a FSP1+ CD45+ cell was reported on the CD8 distance map, and the minimal gray value in each area was measured and converted to a distance, allowing to measure the minimal distance between the CD45+ FSP1+ cell and neighboring CD8+ T cells. For each patient, a frequency distribution of all minimal distances (with 7 µm binning) and the mean minimal distance were calculated.

## Quantification of cell clusters

On the segmented image with dual CD45 FSP1 positive staining combined with CD8 positive staining, centroids from positive particles located in the lamina propria were connected by a Delaunay triangulation, using a custom freely available ImageJ plugin (*Schneider et al., 2012*; *Figure 1—figure supplement 3A–C*, https://github.com/flevet/Delaunay_clustering_ImageJ; copy archived at *Levet, 2022*). All triangles sharing one edge with the ROI defining the lamina propria were removed (*Figure 1—figure supplement 3C*, left panel). On the remaining triangulation a distance threshold, corresponding to the minimal mean distance between fibrocytes and CD8+ T cells (40 µm) was applied, allowing to select the connections with a distance lower than the threshold distance (*Figure 1—figure supplement 3C*, right panel). The number of clusters and their composition were then automatically recorded.

## Dataset transcriptomic analysis

The microarray data of tissular CD8+ T cells were downloaded from the Gene Expression Omnibus (http://www.ncbi.nlm.nih.gov/geo/) using a dataset under the accession code GSE61397. Differential expression analysis between patients with COPD and control subjects was performed using the GEO2R interactive web tool. Heatmaps of the expression profiles for genes related to cell adhesion and chemotaxis were visualized with GraphPad Prism 6 software.

## Tissular CD8+ T cell purification, culture, and secretion profile analyses

After lung parenchyma resection from control or COPD patients (*Supplementary file 7*), samples were finely chopped at room temperature using scissors and then enzymatically dissociated with 40 IU/mL of collagenase (ThermoFisher) in DMEM medium for 45 min at 37 °C. The enzymatic reaction was stopped by adding HBSS medium (Hank's Balanced Salt Solution) without calcium and supplemented with 2 mM EDTA (Invitrogen, Cergy Pontoise, France). The cell suspension was filtered twice using 100 µm gauze and 70 µm cell strainer (Fisher Scientific). Tissular CD8+ T cells were purified by positive selection using CD8 microbeads (Miltenyi Biotech, Paris, France). Then, tissular CD8+ T cells

were resuspended in DMEM supplemented with 8% fetal calf serum, soluble anti-CD3 and anti-CD28 antibodies (respectively, 1 µg and 3 µg for $10^6$ cells) for a final concentration of $0.5 \times 10^6$ cells/mL. After 36 hr, supernatants from tissular CD8$^+$ T cells were collected and frozen, for migration experiments or for further analyses. Supernatants from different samples obtained either from non-smoking subjects or patients with COPD were pooled for migration experiments. Supernatant concentrations of CXCL1, CXCL3, CXCL5, CXCL6, and CXCL8 were measured using ELISA (BioTechne for CXCL1, 5, 6, 8, Abcam for CXCL3). CCL26, CXCL2, and CCL2 concentrations were measured by using a customized Bio-Plex Assay (BioRad, Hercules, CA), using a special plate reader (Bio-Plex 200 Systems, BioRad) and software (Bio-Plex manager), according to the manufacturer's instruction.

### Fibrocyte migration

Fibrocytes precursors were isolated from peripheral blood as described previously (*Dupin et al., 2016*). Fibrocyte migration was assessed using a modified Boyden chamber assay. The transwell inserts (pore size 8 µm, Dutscher) and the wells were coated for 1 hr at room temperature with poly-lysine-ethylene glycol (PEG-PLL, SuSoS, Dübendorf, Switzerland) to prevent cell adhesion. A total of $0.3 \times 10^6$ NANT cells resuspended in 0.2 ml DMEM, containing 4.5 g/l glucose and L-glutamine, supplemented with penicillin/streptomycin and MEM non-essential amino acid solution were added to the upper compartment of each well. When indicated, NANT cells were pretreated for 30 min at 37 °C with 200 nM reparixin (MedChem Express), an antagonist of CXCR1-2. Supernatants of tissular CD8$^+$ T cells from non-smoking control subjects or COPD patients were added to the bottom compartment of each well. When indicated, supernatants were pretreated for 30 min at 37 °C with blocking Ab against CXCL8 (clone 6217, BioTechne, 1 µg/mL) or respective control Ab. After 12 hr, the content of the bottom compartment was removed and DAPI staining was performed to exclude dying cells. Cells were then fixed, permeabilized, and stained with anti-Collagen Type I-FITC (Sigma Aldrich), anti-CD45-APC (BD Pharmingen), anti-CXCR1-PE, and anti-CXCR2-APC-Cy7 (Miltenyi Biotec, Paris, France). Fibrocyte migration was assessed by flow cytometry using double labeling CD45-Collagen I. To obtain absolute values of migratory cells, flow cytometric counts for each condition were obtained during a constant predetermined time period (1 min). The fraction of migratory fibrocytes was defined as the number of CD45$^+$ Col1$^+$ cells counted in the bottom chamber divided by the number of total added cells. These values were normalized to the fraction of migratory fibrocytes obtained in the control condition.

### Fibrocyte CD8$^+$ and CD4$^+$ T cell purification

Peripheral blood mononuclear cells (PBMCs) were first separated from the whole blood by Ficoll-Hypaque (Eurobio Scientific, Les Ulis, France) density gradient centrifugation. Cells were washed twice in cold PBS containing 0.5% bovine serum albumin (BSA, Sigma-Aldrich, Saint Quentin-Fallavier, France) and 2 mM Ethylene Diamine Tetra-acetic Acid (EDTA, Invitrogen, Cergy Pontoise, France). CD8$^+$ and CD4$^+$ T cells were purified by positive selection using CD8 and CD4 microbeads, respectively (Miltenyi Biotech, Paris, France). CD8$^+$/CD4$^+$ T cells were washed in a buffered solution ('CTL-Wash,' Cellular Technology Limited, Bonn, Germany) and resuspended in a serum-free freezing media ('CTL-Cryo Medium,' Cellular Technology Limited, Bonn, Germany) for cryopreservation of freshly-isolated CD8$^+$/CD4$^+$ T cells during fibrocyte differentiation. The CD8$^+$ or CD4$^+$ T cells-depleted cell fraction was then depleted from CD3$^+$ cells using CD3 microbeads (Miltenyi Biotec). Cell suspension containing fibrocyte precursors was cultured during at least 14 days to induce fibrocyte differentiation: a total of $2 \times 10^6$ cells resuspended in 1 ml DMEM (Fisher Scientific, Illkirch, France), containing 4.5 g/l glucose and glutaMAX, supplemented with 20% fetal calf serum (Biowest, Riverside, USA), penicillin/streptomycin and MEM non-essential amino acid solution (Sigma-Aldrich), was added to each well of a 12 well plate. After one week in culture, fibrocyte differentiation was induced by changing the medium to a serum-free medium. Mediums were changed every 2–3 days.

### Fibrocyte/CD8$^+$ T cells and fibrocyte/CD4$^+$ T cells co-culture assay

One day before co-culture, CD8$^+$ T or CD4$^+$ cells were thawed. A buffer solution previously heated to 37 °C (PBS 1 X with 0.5% BSA and 2 mM EDTA) was added to the cell suspension. CD8$^+$/CD4$^+$ T cells were washed with PBS and resuspended in DMEM supplemented with 8% fetal calf serum for a final concentration of $0.5 \times 10^6$ cells/mL. CD8$^+$/CD4$^+$ T cells were either stimulated with a low

dose of CD3 antibody (3 µg/10⁶ cells) to promote cell survival without stimulating cell proliferation ('non-activated' condition), or stimulated overnight with anti-CD3/CD28 coated microbeads (Fisher Scientific) with a bead-to-cell ratio of 1:1 ('activated' condition). At day 0 (co-culture), these beads were removed, CD8⁺/CD4⁺ T cells were stained with 5 µM CellTrace Violet (Fisher Scientific) in case of proliferation experiments, before being added to fibrocyte cultures (0.5 × 10⁶ CD8⁺/CD4⁺ T cells/ well). In blocking experiments, the antibodies (Abs) directed against LFA-1 (clone HI111, BioLegend, 1 µg/mL), CD54 (clone HA58, eBioscience), CD86 (clone IT2.2, eBioscience, 10 µg/mL) or CD44 (clone 82102, BioTechne, 10 µg/mL) were used with their respective control Abs, mIgG1 κ (clone MOPC-21, BioLegend), mIgG2b κ (eBM 2b, eBioscience), mIgG2B (133303, BioTechne). In LFA-1 and CD44 blocking experiments as well as in glucocorticoid drugs experiments, CD8⁺ T cells were preincubated, respectively, with corresponding Abs, budesonide, or fluticasone propionate ($10^{-8}$M, MedChemExpress) at 37 °C for 1 hr before being added to fibrocytes. In CD54 and CD86 blocking experiments, fibrocytes were preincubated with corresponding Abs at 37 °C for 1 hr before adding CD8⁺ T cells. For indirect co-culture, CD8⁺ T cells were cultured in 0.4 µm transwell inserts (Sigma-Aldrich) for 12-well plates.

## Live imaging

For time-lapse microscopy, cells were imaged after 2 days of co-culture, at 37 °C and with 5% $CO_2$ on an inverted DMi8 stand microscope (Leica, Microsystems, Wetzlar, Germany) equipped with a Flash 4.0 sCMOS camera (Hamamatsu, Japan). The objective used was an HC PL FL L 20 X dry 0.4 NA PH1. The multi-positions were done with an ASI MS-2000–500 motorized stage (Applied Scientific Instrumentation, Eugene, USA). The 37 °C/5%CO2 atmosphere was created with an incubator box and a gaz heating system (Pecon GmbH, Erbach, Germany). This system was controlled by MetaMorph software (Molecular Devices, Sunnyvale, USA). Phase contrast images were collected every 2 min for 12 hr. Image analysis and measurements were performed with the ImageJ software. Using the plugin 'Cell counter' of the Fiji software, the number of CD8⁺ T cells in direct contact with a fibrocyte as well as the number of free CD8⁺ T cells were manually counted at the beginning of the acquisition and after 12 hr of acquisition. Cell tracking was performed using the 'Manual Tracking' plugin of the Fiji software to determine the durations of contacts between tracked CD8⁺ T cell with fibrocytes and the frequency of contact. A contact was defined manually by a direct interaction between CD8⁺ T cell and fibrocyte. Five numerical variables were collected to characterize CD8⁺ T cell dynamic over time. The mean speed corresponded to the track length divided by the time of tracking duration. The mean free speed corresponded to the length of the track when the T cell was not interacting with any other cell, divided by the time spent free. The mean contact speed corresponded to the length of the track when the T cell is in contact with a fibrocyte, divided by the time spent in contact. For each T cell and for each contact, a contact time was defined as the time spent in contact until the T cell becomes free again. Then, each T cell can have many contact times with fibrocytes. The contact coefficient was defined by the proportion of time the T cell was in contact with a fibrocyte divided by the time of tracking duration.

## CD8⁺ and CD4⁺ T cell characterization by flow cytometry

Four or 6 days after co-culture, CD8⁺/CD4⁺ T cells were harvested and manually counted before being processed for FACS analysis. Dead cells were excluded by using DAPI or the Zombie NIR fixable viability kit (BioLegend). Intracellular cytokines were assessed following stimulation with PMA (25 ng/ml, Sigma-Aldrich), ionomycin (1 µM, Sigma-Aldrich) for 4 hr, and brefeldin A (5 µg/ml, Sigma-Aldrich) for the last 3 hr. Cells were stained with anti-CD8-PerCP-Vio700 or anti-CD4-PerCP-Vio700, anti-CD45-RA-FITC, and then fixed, permeabilized using the IntraPrep Permeabilization Reagent Kit (Beckman Coulter) and stained with anti-Granzyme-APC, anti-TNF-α-PE, anti-IFN-γ-APC, anti-IL-17-PE-Cy7, anti-IL-10-PE or isotype controls (Miltenyi Biotech, Paris, France). The percentage of cell proliferation was estimated using Cell Trace Violet fluorescence loss. FACS data were acquired using a Canto II 4-Blue 2-Violet 2-Red laser configuration (BD Biosciences). Flow cytometry analyses were performed using Diva 8 (BD Biosciences). Human TNF-α concentration levels were quantified using ELISA following the manufacturer's recommendations (BioTechne). Values below the detection limit were counted as zero.

## CD107a expression of CD8+ T cells

Six days after co-culture, CD8+ T cells were harvested and stained to assess CD107a expression following stimulation with PMA (25 ng/ml, Sigma-Aldrich) and ionomycin (1 µM, Sigma-Aldrich) for 3 hr. Cells were stained with anti-CD8-PerCP-Vio700, anti-CD45-RA-FITC, or isotype controls (Miltenyi Biotech, Paris, France). For membrane expression analysis, cells were then stained with anti-CD107a-PE or isotype control (BD Biosciences, San Jose, CA). For total expression analysis, cells were fixed, permeabilized using the IntraPrep Permeabilization Reagent Kit (Beckman Coulter), and stained with anti-CD107a-PE or isotype control. FACS data were acquired using a Canto II 4-Blue 2-Violet 2-Red laser configuration (BD Biosciences).

## Bronchial epithelial cell culture

Human basal bronchial epithelial cells were derived from bronchial specimens (*Supplementary file 9*) as described previously (*Trian et al., 2015*). Bronchial epithelial tissue was cultured in PneumaCult-Ex medium (Stemcell Technologies, Vancouver, Canada) under a water-saturated 5% $CO_2$ atmosphere at 37 °C until epithelial cells reached 80–90% confluence. For the cytotoxic assay, basal bronchial epithelial cells were plated on a 24-wells plate in PneumaCult-Ex medium.

## Cell trace violet-based cytotoxic activity of CD8+ T cells

After 6 days of co-culture, CD8+ T cells were resuspended ($2.10^6$ cells/ml), and labeled with 5 µM Cell Trace Violet (Fisher Scientific) following the manufacturer's recommendations. Cell Trace Violet-labeled CD8+ T cells as effector cells and primary basal bronchial epithelial cells as target cells were co-incubated at effector:target ratio of 4:1 in PneumaCult-Ex medium. After 6 hr, CD8+ T cells were kept with supernatant, and basal epithelial cells were harvested by using 0.05% trypsin-EDTA (Fisher Scientific). The entire content of each well was stained using Annexin V- Propidium iodide (Fisher Scientific) for the detection of apoptotic and living cells by flow cytometry, according to the manufacturer's instructions. CD8+ T cells exclusion was performed based on the Cell Trace Violet-based fluorescence performed 6 hr before.

## Label-free quantitative proteomics

Four independent biological replicates on total protein extracts from fibrocytes cultured alone (control condition), or co-cultured with CD8+ T cells that have been previously non-activated or activated during 6 days. 10 µg of proteins were loaded on a 10% acrylamide SDS-PAGE gel and proteins were visualized by Colloidal Blue staining. Migration was stopped when samples had just entered the resolving gel and the unresolved region of the gel was cut into only one segment. Each SDS-PAGE band was cut into 1 mm × 1 mm gel pieces. Gel pieces were destained in 25 mM ammonium bicarbonate ($NH_4HCO_3$), 50% Acetonitrile (ACN), and shrunk in ACN for 10 min. After ACN removal, gel pieces were dried at room temperature. Proteins were first reduced in 10 mM dithiothreitol, 100 mM $NH_4HCO_3$ for 60 min at 56 °C then alkylated in 100 mM iodoacetamide, 100 mM $NH_4HCO_3$ for 60 min at room temperature and shrunk in ACN for 10 min. After ACN removal, gel pieces were rehydrated with 50 mM $NH_4HCO_3$ for 10 min at room temperature. Before protein digestion, gel pieces were shrunk in ACN for 10 min and dried at room temperature. Proteins were digested by incubating each gel slice with 10 ng/µl of trypsin (V5111, Promega) in 40 mM $NH_4HCO_3$, rehydrated at 4 °C for 10 min, and finally incubated overnight at 37 °C. The resulting peptides were extracted from the gel in three steps: a first incubation in 40 mM $NH_4HCO_3$ for 15 min at room temperature and two incubations in 47.5% ACN, 5% formic acid for 15 min at room temperature. The three collected extractions were pooled with the initial digestion supernatant, dried in a SpeedVac, and resuspended with 0.1% formic acid for a final concentration of 0.02 µg/µL. NanoLC-MS/MS analysis was performed using an Ultimate 3000 RSLC Nano-UPHLC system (Thermo Scientific, USA) coupled to a nanospray Orbitrap Fusion Lumos Tribrid Mass Spectrometer (Thermo Fisher Scientific, California, USA). Each peptide extract were loaded on a 300 µm ID × 5 mm PepMap $C_{18}$ precolumn (Thermo Scientific, USA) at a flow rate of 10 µL/min. After a 3 min desalting step, peptides were separated on a 50 cm EasySpray column (75 µm ID, 2 µm $C_{18}$ beads, 100 Å pore size, ES903, Thermo Fisher Scientific) with a 4–40% linear gradient of solvent B (0.1% formic acid in 80% ACN) in 115 min. The separation flow rate was set at 300 nL/min. The mass spectrometer operated in positive ion mode at a 1.9 kV needle voltage. Data were acquired using Xcalibur 4.4 software in a data-dependent mode. MS scans (m/z 375–1500) were

recorded at a resolution of $R$=120,000 (@ m/z 200), a standard AGC target, and an injection time in automatic mode, followed by a top speed duty cycle of up to 3 s for MS/MS acquisition. Precursor ions (2–7 charge states) were isolated in the quadrupole with a mass window of 1.6 Th and fragmented with HCD@28% normalized collision energy. MS/MS data were acquired in the Orbitrap cell with a resolution of $R$=30,000 (@m/z 200), a standard AGC target, and a maximum injection time in automatic mode. Selected precursors were excluded for 60 s. Protein identification and Label-Free Quantification (LFQ) were done in Proteome Discoverer 2.5. MS Amanda 2.0, Sequest HT, and Mascot 2.5 algorithms were used for protein identification in batch mode by searching against a Uniprot *Homo sapiens* database (81,856 entries, released January 20th, 2023). Two missed enzyme cleavages were allowed for the trypsin. Mass tolerances in MS and MS/MS were set to 10 ppm and 0.02 Da. Oxidation (M) and acetylation (K) were searched as dynamic modifications and carbamidomethylation (C) as static modifications. Peptide validation was performed using the Percolator algorithm and only 'high confidence' peptides were retained corresponding to a 1% false discovery rate at the peptide level (*Käll et al., 2007*). Minora feature detector node (LFQ) was used along with the feature mapper and precursor ions quantifier. The normalization parameters were selected as follows: 1: Unique peptides, 2: Precursor abundance based on intensity, 3: Normalization mode:total peptide amount, 4: Protein abundance calculation:summed abundances, 5: Protein ratio calculation:pairwise ratio based, 6: Imputation mode:Low abundance resampling and 7: Hypothesis test:t-test (background based). Quantitative data were considered for master proteins, quantified by a minimum of two unique peptides, a fold changes above 2, and a statistical p-value adjusted using Benjamini-Hochberg correction for the FDR lower than 0.05. The mass spectrometry proteomics data have been deposited to the ProteomeXchange Consortium via the PRIDE (*Deutsch et al., 2020*) partner repository with the dataset identifier PXD041402. Proteins were clusterized according to their functions by using the Kyoto Encyclopedia of genes and genome analysis in the search tool for retrieval of interaction between genes and proteins (STRING) database. More global analysis of the data was performed via the use of Ingenuity Pathway Analysis (IPA; Qiagen). We used the 'Core Analysis' package to identify relationships, mechanisms, functions, and pathways relevant to a dataset.

## The mathematical model

Exhaustive description of the mathematical model is provided in the Appendix 1.

To understand the interaction between fibrocyte and CD8[+] T cells in the spatial cellular organization in the peribronchial area, we constructed a discrete time cellular automata model. Two agent types are introduced - CD8[+] T cell agents and fibrocytes agents, denoted C and F, respectively. C and F cells evolve on a lattice in two-dimensions. We take as the surface of interest a zone with a crown shape, containing 3 652 lattice sites corresponding to a total area of approximately 179,000 μm², which is in agreement with our in situ measurements. Reflecting (zero-flux) boundary conditions are imposed at the external and internal borders. On each site, there is at most one cell. The lattice is initially randomly seeded with both F and C cells at densities corresponding to the mean distribution of non-smokers subjects, reflecting the 'healthy' situation: $n_0(C)$ = 660 cells/mm², and $n_0(F)$ = 106 cells/mm². This corresponds to an average value of $N_0(C)$ = 118 C cells and $N_0(F)$ = 19 F cells.

We assumed that for a healthy subject as for a patient with COPD, the same model can be applied but with different parameters. These parameters are estimated thanks to experiments and data from the literature (see Appendix 1 and *Dupin et al., 2023* for a complete description, *Supplementary file 11* for numerical values).

The notations and parameters of the mathematical model are summarized in *Supplementary file 10* and their numerical values are given in *Supplementary file 11*. We now describe the behavior of the cells and their interactions. F and C cells infiltrate into the peribronchial area at the stable state with the respective probabilities $p_{istaF}$ and $p_{istaC}$, and during exacerbation, a supplementary infiltration can occur, each year, with the probability $p_{iexaF}$ (resp. $p_{iexaC}$). In the model, C cells can proliferate with a very low probability $p_C$, but the presence of F cells in the local neighborhood of a C cell can induce C cell division with an increased probability $p_{C/F}$, based on our own results and another study (*Afroj et al., 2021*). We suppose that fibrocytes do not proliferate, as shown by our own in vitro observations (data not shown) and other studies (*Ling et al., 2019*; *Schmidt et al., 2003*). F and C cells can move, with probabilities which are determined by the results from chemotaxis experiments (*Figure 2*). F and C cells die with a 'basal' probability $p_{dC}$ (respectively, $p_{dF}$). C cells also die with an

increased probability $p_{dC+}$ when the considered C cell has many other C cells in its neighborhood, in agreement with previous data (*Zenke et al., 2020*). Some of the probabilities are independent of the local environment ($p_{istaF}$, $p_{istaC}$, $p_{iexaF}$, $p_{iexaC}$, $p_C$), the other ones being dependent of the local environment ($p_{C/F}$, $p_{dC+}$, and displacement probabilities) (*Figure 8A*).

Each simulation represents a total duration of 20 years and is divided into 3,504,000 iterations, of 3 min each. Each type of simulation is performed 160 times. This time period allowed the investigation of COPD development. At the final state (20 years), the total numbers of F and C cells, the densities of C cells in interaction with F cells, the minimal distances between C and F cells, and the number and composition of clusters were quantified in the control and COPD situations. For therapeutic interventions, different dynamics were applied during 7 years, starting from COPD states that have been generated using the application of COPD dynamics during 20 years. At the final state (7 years after the application of therapeutic dynamics), the same outcomes than at 20 years were quantified.

### Definition of biological and technical replicates

Biological replicates are samples purified from different patients. Technical replicates are repeated measurements in the same biological samples. For in situ analyses, both technical replicates (measurements performed on at least two bronchi) and biological replicates have been performed. For most of in vitro analyses, only biological replicates have been obtained, because our cells of interest have a very limited lifespan in culture and experiments can, therefore, be performed only once. Exceptions include migration assays and ELISA measurements, with both technical duplicates and biological replicates. For simulations, technical replicates have been performed, with each type of simulation being repeated 160 times.

### Statistical analyses

Statistical significance, defined as $p < 0.05$, was analyzed by t-tests and MANOVA for variables with parametric distribution, and by Kruskal-Wallis with multiple comparison z-tests, Mann-Whitney tests, Wilcoxon tests, and Spearman correlation coefficients for variables with non-parametric distribution and by two-way ANOVA for distribution tests, using GraphPad Prism 6 software. RStudio software was used to perform stepwise regression and multivariate regression analyses.

## Acknowledgements

We thank the study participants and the staff of the Thoracic Surgery, Radiology, Pathology, Respiratory, Lung Function Testing departments from the University Hospital of Bordeaux (Bordeaux, France), Isabelle Goasdoue, Isabelle Bernis, Natacha Robert, Virginie Niel, and Marine Servat from the clinical investigation center for technical assistance, and Atika Zouine and Vincent Pitard for technical assistance at the Flow cytometry facility (CNRS UMS 3427, INSERM US 005, Univ. Bordeaux, F-33000 Bordeaux, France), Christel Poujol, Sébastien Marais and Fabrice Cordelières for help with imaging and image analysis at the Bordeaux Imaging Centre (BIC; Bordeaux, France). Microscopy was performed at BIC, a service unit of the CNRS-INSERM and Bordeaux University, a member of the national BioImaging infrastructure of France supported by the French National Research Agency (ANR-10-INBS-04). Funding: The project was funded by: the 'Fondation de l'Université de Bordeaux' (Fonds pour les maladies chroniques nécessitant une assistance médico-technique FGLMR/AVAD) (ID) the 'Agence Nationale de la Recherche' (ANR-21-CE18-0001-01) (ID) AstraZeneca (an unrestricted grant to PB). The COBRA cohort was funded by AstraZeneca, Chiesi, Glaxo-SmithKline, Novartis and Roche.

## Additional information

### Competing interests

Maeva Zysman: MZ reports personal fees from AstraZeneca, Boehringer Ingelheim, Novartis, Chiesi, GlaxoSmithKline and non-financial support Lilly outside the submitted work. Pierre-Olivier Girodet: POG has a patent (EP 3050574: Use of plerixafor for treating and/or preventing acute exacerbations of chronic obstructive pulmonary disease) granted. POG reports grants, personal fees and non-financial support from AstraZeneca, personal fees and non-financial support from Chiesi, personal fees and

non-financial support from GlaxoSmithKline, personal fees and non-financial support from Novartis, personal fees and non-financial support from Sanofi, outside the submitted work. Patrick Berger: PB has a patent (EP N3050574: Use of plerixafor for treating and/or preventing acute exacerbations of chronic obstructive pulmonary disease) granted. PB reports grants from AstraZeneca, Glaxo-Smith-Kline, Novartis, Chiesi, which support COBRA during the conduct of the study; grants and personal fees from AstraZeneca, BoehringerIngelheim, Novartis, personal fees and non-financial support from Chiesi, Sanofi, Menarini, outside the submitted work. Isabelle Dupin: ID has a patent (EP 3050574: Use of plerixafor for treating and/or preventing acute exacerbations of chronic obstructive pulmonary disease) granted. The other authors declare that no competing interests exist.

## Funding

| Funder | Grant reference number | Author |
| --- | --- | --- |
| Fondation Bordeaux Université | Fonds pour les maladies chroniques nécessitant une assistance médico-technique FGLMR/AVAD | Isabelle Dupin |
| Fondation Bordeaux Université | Assistance Ventilatoire à Domicile" (AVAD), Fédération Girondine de Lutte contre les Maladies Respiratoires (FGLMR) | Pauline Henrot Maeva Zysman |
| Agence Nationale de la Recherche | ANR-21-CE18-0001-01 | Isabelle Dupin |
| AstraZeneca France | | Patrick Berger |
| AstraZeneca France | | Pierre-Olivier Girodet |
| Boehringer Ingelheim | | Patrick Berger |
| Novartis | | Patrick Berger |

The funders had no role in study design, data collection and interpretation, or the decision to submit the work for publication.

## Author contributions

Edmée Eyraud, Formal analysis, Validation, Investigation, Visualization, Methodology, Writing – original draft; Elise Maurat, Formal analysis, Validation, Investigation, Visualization, Methodology; Jean-Marc Sac-Epée, Resources, Software, Formal analysis, Investigation, Visualization; Pauline Henrot, Validation, Investigation, Visualization, Methodology, Writing – review and editing; Maeva Zysman, Resources, Investigation, Methodology, Writing – review and editing; Pauline Esteves, Thomas Trian, Jean-William Dupuy, Alexander Leipold, Antoine-Emmanuel Saliba, Methodology; Hugues Begueret, Pierre-Olivier Girodet, Romain Hustache-Castaing, Resources; Matthieu Thumerel, Resources, Investigation; Roger Marthan, Writing – review and editing; Florian Levet, Resources, Software, Methodology; Pierre Vallois, Conceptualization, Software, Formal analysis, Validation, Investigation, Methodology, Writing – review and editing; Cécile Contin-Bordes, Conceptualization, Validation, Investigation, Methodology, Writing – review and editing; Patrick Berger, Conceptualization, Resources, Funding acquisition, Validation, Methodology, Writing – review and editing; Isabelle Dupin, Conceptualization, Formal analysis, Supervision, Funding acquisition, Validation, Investigation, Visualization, Methodology, Writing – original draft, Project administration, Writing – review and editing

## Author ORCIDs

Edmée Eyraud http://orcid.org/0000-0003-3807-8359
Isabelle Dupin https://orcid.org/0000-0002-4992-9625

## Ethics

Clinical trial registration NCT01692444.
All subjects gave their written informed consent to participate to the studies. The studies received approval from the local or national ethics committees:Lung tissues for the in situ study were obtained from a previously described cohort (Dupin et al., 2019). The study was registered at ClinicalTrials.gov with identifier NCT01692444 ('Fibrochir' study). The study protocol was approved by the research

ethics committee ('CPP') and the French National Agency for Medicines and Health Products Safety ('ANSM'). Lung tissues for the purification of tissular CD8+ T cells and bronchial epithelial cells were obtained from a separate cohort of patients requiring thoracic lobectomy surgery for nodule or cancer (pN0) (i.e. TUBE, sponsored by the University hospital of Bordeaux, which includes its own local ethic committee (CHUBX 2020/54)). According to the French law and the MR004 regulation, patients received an information form, allowing them to refuse the use of their surgical samples for research. To study fibrocyte- CD8+ T cells interplay in vitro, blood samples were obtained from a separate cohort of COPD patients, (i.e. COBRA (Bronchial Obstruction and Asthma Cohort; sponsored by the French National Institute of Health and Medical Research, INSERM, Ethics committee number: 2008-A00294-51/1), as outpatients in the Clinical Investigation Centre of the University Hospital of Bordeaux.

Reviewer #1 (Public Review): https://doi.org/10.7554/eLife.85875.3.sa1
Reviewer #2 (Public Review): https://doi.org/10.7554/eLife.85875.3.sa2
Reviewer #3 (Public Review): https://doi.org/10.7554/eLife.85875.3.sa3
Author Response https://doi.org/10.7554/eLife.85875.3.sa4

---

# Additional files

## Supplementary files

• Supplementary file 1. Association between the density of fibrocytes and clinical characteristics. $FEV_1$, forced expiratory volume in 1 s; FVC, forced vital capacity; LFT, lung function test; RV, residual volume; TLCO, Transfer Lung capacity of Carbon monoxide, $PaO_2$, partial arterial oxygen pressure, $PaCO_2$, partial arterial carbon dioxide pressure; WA, mean wall area; LA, mean lumen area, WA%, mean wall area percentage; WT, wall thickness; LAA, low-attenuation area; MLA E or I, mean lung attenuation value during expiration or inspiration. MLA I-E, the difference between inspiratory and expiratory mean lung attenuation value. $\%CSA_{<5}$, percentage of total lung area taken up by the cross-sectional area of pulmonary vessels less than 5 $mm^2$; $\%CSA_{5-10}$, percentage of total lung area taken up by the cross-sectional area of pulmonary vessels between 5 and 10 $mm^2$; $CSN_{<5}$, number of vessels less than 5 $mm^2$ normalized by total lung area; $CSN_{5-10}$, number of vessels between 5 and 10 $mm^2$ normalized by total lung area; NR: not relevant. The correlation coefficient (r), 95% confidence interval, and significance level (p value), were obtained by using nonparametric Spearman analysis.

• Supplementary file 2. Association between the density of CD8+ T cells and clinical characteristics. $FEV_1$, forced expiratory volume in 1 second; FVC, forced vital capacity; LFT, lung function test; RV, residual volume; TLCO, Transfer Lung capacity of Carbon monoxide, $PaO_2$, partial arterial oxygen pressure, $PaCO_2$, partial arterial carbon dioxide pressure; WA, mean wall area; LA, mean lumen area, WA%, mean wall area percentage; WT, wall thickness; LAA, low-attenuation area; MLA E or I, mean lung attenuation value during expiration or inspiration. MLA I-E, the difference between inspiratory and expiratory mean lung attenuation value. $\%CSA_{<5}$, percentage of total lung area taken up by the cross-sectional area of pulmonary vessels less than 5 $mm^2$; $\%CSA_{5-10}$, percentage of total lung area taken up by the cross-sectional area of pulmonary vessels between 5 and 10 $mm^2$; $CSN_{<5}$, number of vessels less than 5 $mm^2$ normalized by total lung area; $CSN_{5-10}$, number of vessels between 5 and 10 $mm^2$ normalized by total lung area; NR: not relevant. The correlation coefficient (r), 95% confidence interval, and significance level (p value), were obtained by using nonparametric Spearman analysis.

• Supplementary file 3. Association between interacting cell density and clinical characteristics. $FEV_1$, forced expiratory volume in 1 s; FVC, forced vital capacity; LFT, lung function test; RV, residual volume; TLCO, Transfer Lung capacity of Carbon monoxide, $PaO_2$, partial arterial oxygen pressure, $PaCO_2$, partial arterial carbon dioxide pressure; WA, mean wall area; LA, mean lumen area, WA%, mean wall area percentage; WT, wall thickness; LAA, low-attenuation area; MLA E or I, mean lung attenuation value during expiration or inspiration. MLA I-E, the difference between inspiratory and expiratory mean lung attenuation value. $\%CSA_{<5}$, percentage of total lung area taken up by the cross-sectional area of pulmonary vessels less than 5 $mm^2$; $\%CSA_{5-10}$, percentage of total lung area taken up by the cross-sectional area of pulmonary vessels between 5 and 10 $mm^2$; $CSN_{<5}$, number of vessels less than 5 $mm^2$ normalized by total lung area; $CSN_{5-10}$, number of vessels between 5 and 10 $mm^2$ normalized by total lung area; NR: not relevant. The correlation coefficient (r), 95% confidence interval, and significance level (p value), were obtained by using nonparametric Spearman analysis.

• Supplementary file 4. Association between the mean minimal distance between fibrocytes and CD8+ T cells and clinical characteristics. $FEV_1$, forced expiratory volume in 1 s; FVC, forced vital

capacity; LFT, lung function test; RV, residual volume; TLCO, Transfer Lung capacity of Carbon monoxide, $PaO_2$, partial arterial oxygen pressure, $PaCO_2$, partial arterial carbon dioxide pressure; WA, mean wall area; LA, mean lumen area, WA%, mean wall area percentage; WT, wall thickness; LAA, low-attenuation area; MLA E or I, mean lung attenuation value during expiration or inspiration. MLA I-E, the difference between inspiratory and expiratory mean lung attenuation value. $\%CSA_{<5}$, percentage of total lung area taken up by the cross-sectional area of pulmonary vessels less than 5 $mm^2$; $\%CSA_{5-10}$, percentage of total lung area taken up by the cross-sectional area of pulmonary vessels between 5 and 10 $mm^2$; $CSN_{<5}$, number of vessels less than 5 $mm^2$ normalized by total lung area; $CSN_{5-10}$, number of vessels between 5 and 10 $mm^2$ normalized by total lung area; NR: not relevant. The correlation coefficient (r), 95% confidence interval, and significance level (p value), were obtained by using nonparametric Spearman analysis.

• Supplementary file 5. Association between the density of mixed cell clusters and clinical characteristics. $FEV_1$, forced expiratory volume in 1 s; FVC, forced vital capacity; LFT, lung function test; RV, residual volume; TLCO, Transfer Lung capacity of Carbon monoxide, $PaO_2$, partial arterial oxygen pressure, $PaCO_2$, partial arterial carbon dioxide pressure; WA, mean wall area; LA, mean lumen area, WA%, mean wall area percentage; WT, wall thickness; LAA, low-attenuation area; MLA E or I, mean lung attenuation value during expiration or inspiration. MLA I-E, the difference between inspiratory and expiratory mean lung attenuation value. $\%CSA_{<5}$, percentage of total lung area taken up by the cross-sectional area of pulmonary vessels less than 5 $mm^2$; $\%CSA_{5-10}$, percentage of total lung area taken up by the cross-sectional area of pulmonary vessels between 5 and 10 $mm^2$; $CSN_{<5}$, number of vessels less than 5 $mm^2$ normalized by total lung area; $CSN_{5-10}$, number of vessels between 5 and 10 $mm^2$ normalized by total lung area; NR: not relevant. The correlation coefficient (r), 95% confidence interval, and significance level (p value), were obtained by using nonparametric Spearman analysis.

• Supplementary file 6. Multivariate analysis of $FEV_1/FVC$. $FEV_1$, forced expiratory volume in 1 s; FVC, forced vital capacity.

• Supplementary file 7. Patient characteristics (for tissular $CD8^+$ T cells purification). Plus–minus values are means ± SD. PFT, pulmonary function test; $FEV_1$, forced expiratory volume in 1 s; FVC, forced vital capacity.

• Supplementary file 8. Patient characteristics (for circulating $CD8^+/CD4^+$ T cells and fibrocyte precursors purification). Plus–minus values are means ± SD. PFT, pulmonary function test; $FEV_1$, forced expiratory volume in 1 s; FVC, forced vital capacity; $PaO_2$, partial arterial oxygen pressure, $PaCO_2$, partial arterial carbon dioxide pressure.

• Supplementary file 9. Patient characteristics (for basal bronchial epithelial cell purification). Plus–minus values are means ± SD. PFT, pulmonary function test; $FEV_1$, forced expiratory volume in 1 s; FVC, forced vital capacity.

• Supplementary file 10. Definition of the notations and parameters of the mathematical model.

• Supplementary file 11. Numerical values of parameters depending in control and COPD situations.

• MDAR checklist

• Source code 1. Program to simulate CD8+ T cells and fibrocytes evolution in the peribronchial area.

## Data availability

All data needed to evaluate the conclusions are present in the paper, the Appendix 1, and/or the deposited data. The customized ImageJ plugin used to perform Delaunay triangulation and cluster quantification is available here: https://github.com/flevet/Delaunay_clustering_ImageJA (copy archived at *Levet, 2022*). The complete version of the code for launching the simulations associated to control and COPD dynamics is included as *Source code 1*. The mass spectrometry proteomics data have been deposited to the ProteomeXchange Consortium (http://proteomecentral.proteomexchange.org) via the PRIDE partner repository with the dataset identifier PXD041402.

The following dataset was generated:

| Author(s) | Year | Dataset title | Dataset URL | Database and Identifier |
|---|---|---|---|---|
| Jean-William D, Isabelle D | 2023 | Short-range interactions between fibrocytes and CD8+ T cells in COPD bronchial inflammatory response | https://proteomecentral.proteomexchange.org/cgi/GetDataset?ID=PXD041402 | ProteomeXchange, PXD041402 |

The following previously published dataset was used:

| Author(s) | Year | Dataset title | Dataset URL | Database and Identifier |
|---|---|---|---|---|
| Hombrink P, Jongejan A, Moerland PD, van Lier RA | 2016 | The transcriptional profile of human CD8+ lung resident memory T-cells | https://www.ncbi.nlm.nih.gov/geo/query/acc.cgi?acc=GSE61397 | NCBI Gene Expression Omnibus, GSE61397 |

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

# Appendix 1

The mathematical model is fully described below.

## Model description

### 1. Model definition

#### 1.1. Surface of interest

Fibrocytes and CD8$^+$ T cells are noted, respectively, F and C. They evolve on a lattice in the x-y plane of dimension 103 × 103 where the area of each square is determined by the size of a cell (*Appendix 1—figure 1*) and has a side length of 7 μm, which has been calculated from our in situ analyses to match the mean cellular area (around 50 μm²). Indeed, a C cell can be approximated by a disk with a diameter of about 8 μm (*Mrass et al., 2017* and our unpublished observations), giving a surface area of 50 μm². The size is roughly equivalent to a F cell in lung tissue. Thus, the cells are modeled by squares with a side length $x_0$=7 μm, which correspond to the units of the lattice. Each element of the lattice is defined by two coordinates, where the point on the upper left (resp. lower right) corner has the coordinates (1; 1) (resp. (103; 103)). The coordinates of the center of the lattice are (52; 52). The geometry of bronchi corresponds to the transverse section of a cylinder, then we model our surface of interest, the lamina propria, by a zone with a crown shape. The internal radius has been calculated from the mean area of the lumen area combined with the epithelium surface (216,567 μm²), i.e., 38 lattice sites; the external radius has been determined from the mean area of the peribronchial surface (396,436 μm²), i.e., 50 lattice sites. Then the lamina propria $L$ is the set of points with coordinates $(i, j)$ such that:

$$38 \leq \hat{d}\,(i,j),\,(52,\,52) \leq 50$$

where $\hat{d}$ is the pseudo-distance: $\hat{d}\left((i,j),\,\left(i',j'\right)\right) = \left\lfloor\sqrt{\left(i-i'\right)^2 + \left(j-j'\right)^2}\right\rfloor$

and $\lfloor x \rfloor$ stands for the integer part of the real number $x$. We thus obtain a working surface containing 3652 lattice sites (potential cells) corresponding to an area of approximately 179,000 μm², which is in agreement with our in situ measurements. In other words, the number $|L|$ of elements of $L$ equals 3652. Reflecting (zero-flux) boundary conditions are imposed at the external and internal borders. On each site, there is at most one cell.

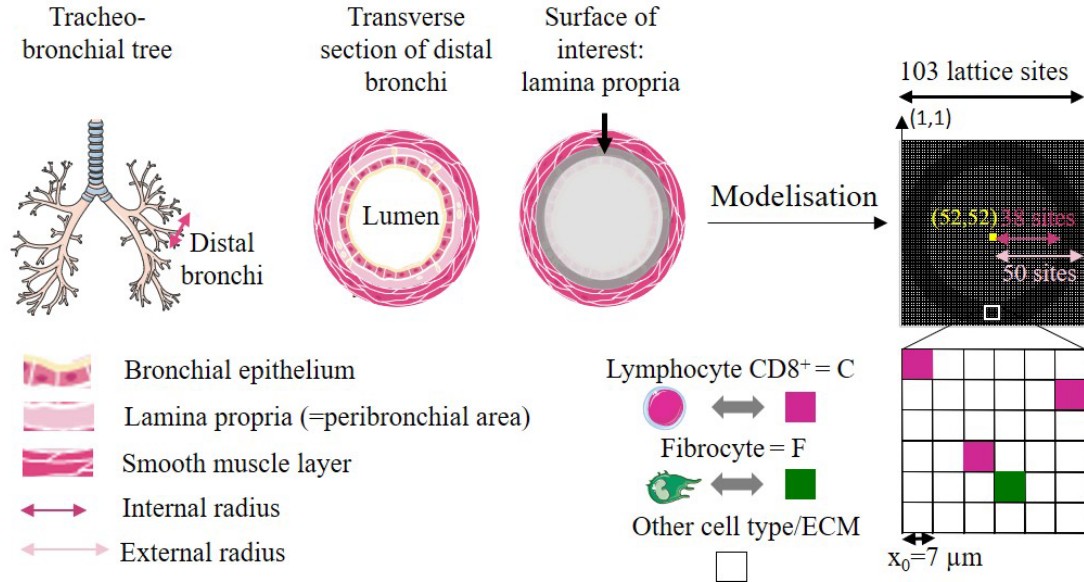

**Appendix 1—figure 1.** The lamina propria $L$ forms two 2-dimensional crown shapes in the bronchial wall, between the bronchial epithelium and the smooth muscle layer. Adapted from *Dupin et al., 2023*.

In the literature, it is described that bronchial wall thickness is increased in COPD patients, (*Hasegawa et al., 2006*; *Hogg et al., 2004*) but we did not observe this increase in our tissue

measurements. We will consider the area of the lamina propria to be the same for healthy subjects and patients with COPD.

## 1.2. Neighborhood

For any site $(i,j) \in L$, $M$, $M(i,j)$ is the neighborhood of $(i,j)$, it is the set of $(i-1, j-1)$, $(i-1, j)$, $(i-1, j+1)$, $(i, j-1)$, $(i, j+1)$, $(i+1, j-1)$, $(i+1, j)$, $(i+1, j+1)$ belonging to $L$ (*Appendix 1—figure 2*). For a site inside the lamina propria, the cardinal of $M(i,j)$ is 8 and lower if this site is at the edge of $L$. We will note the following $|M(i,j)|$ the number of elements of $M(i,j)$. In the literature, Moore's neighborhood is $M(i,j) \cup \{(i,j)\}$.

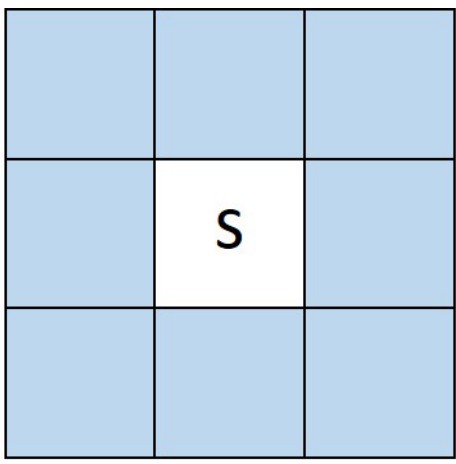

## Neighbourhood $M(s)$

**Appendix 1—figure 2.** Schematic representation showing the neighborhood $M(s)$ at the site s (shaded blue).

A site of L has the code 1 (resp. 2) if it contains an F (resp. C) cell. If the site is empty it will be coded 0. This state corresponds either to another cell type (mainly mesenchymal, that was hypothesized to interact minimally with fibrocyte and CD8[+] T cells) or to the extracellular matrix, which again does not play any role in the cellular cross-talk. This is the predominant state, as the bronchial wall contains structural and immune cells sparsely embedded in the extracellular matrix.

A configuration is an element $x = x(i,j)_{(i,j) \in L}$ where $x(i,j)$ belongs to $\{0, 1, 2\}$ and $x(i,j) = 1$ (resp. $x(i,j) = 2$) means that an F cell (resp. C) occupies the site $(i,j)$ and $x(i,j) = 0$ when the site $(i,j)$ is empty. The set of configurations is $\{0, 1, 2\}^L$ and is identified with $\{0, 1, 2\}^{|L|}$.

For any $s = (i,j)$, $V(F)(s)$ (resp. $V(C)(s)$) denotes the number of F cells (resp. C) near s:
$V(F)(s) = \sum_{s' \in M(s)} 1_{\{x(s')=1\}}$, $V(C)(s) = \sum_{s' \in M(s)} 1_{\{x(s')=2\}}$
$V(s)$ is the number of F and C cells close to $s$:
$V(s) = V(F)(s) + V(C)(s) = \sum_{s' \in M(s)} 1_{\{x(s')=1 \, or \, 2\}} = \sum_{s' \in M(s)} 1_{\{x(s') \neq 0\}}$

## 1.3. Initial distribution of cells

The lattice is initially randomly seeded with both F and C cells at the densities described below. For a given patient, $n(F)$ and $n(C)$ are, respectively, the number of F and C cells divided by the area of the lamina propria (quantified by image analysis). Thus, $n(F)$ and $n(C)$ represent numbers of cells per mm². We define the initial density $n_0(F+C)$, the initial F cells density $n_0(F)$ and the initial C cells density $n_0(C)$. For the initial distribution, we used the mean distribution of non-smokers subjects, reflecting the 'healthy' situation:

$n_0(F+C)$ = 766 cells/mm², *i.e.*, 0.766.10⁻³ cells/µm²
$n_0(C)$ = 660 cells/mm², *i.e.*, 0.660.10⁻³ cells/µm²
$n_0(F)$ = 106 cells/mm², *i.e.*, 0.106.10⁻³ cells/µm²

Since the area of the lattice is approximately 179,000 µm², To have an initial density of 0.766 × 10⁻³ cells/µm², we will, therefore, need to add an average value of 0.766.10⁻³ × 179,000 = 137 cells in the grid of 3652 cells. Starting from an initial situation with 137 cells in the grid, this corresponds to an average value of $N_0(C)$ = 118 C cells and $N_0(F)$ = 19 F cells.

## 1.4 The different time scales

The median speed of a C cell measured in lung tissue for 15 min is $v_0$ = 2,3 μm/min (*Mrass et al., 2017*). For a cell modeled by a square with a side length $x_o$ = 7 μm, $v_0$ represents a movement of one square (lattice square) every 3 min. Since we have no information on the in vivo speed of an F cell, we will assume that its speed is identical to that of a C cell. Thus, we choose a time step of 3 min for each iteration.

COPD results from a progressive phenomenon: the disease is usually diagnosed starting from the age of 40, but results from exposure to cigarette smoke for several years (*Løkke et al., 2006*). The time scale that interests us is, therefore, 10–30 years. We thus choose a time period $T$ of 20 years for the simulations. This corresponds to 20 (years)=365 (days) × 24(hr) × 20 (time steps of 3 min)=3 504000 iterations

## 2. The probabilistic interaction model

The mathematical model that we have chosen aims to take into account the interaction between F and C cells and its consequences. We assumed that for a healthy subject as for a patient with COPD, the same model is applied with the same parameters but with different values. These parameters are estimated thanks to experiments and data from the literature and the mathematical results obtained with the streamlined model (*Dupin et al., 2023*). The notations and parameters of the model are described in *Supplementary file 10*.

First, we first describe the general behavior of a cell, then dissociate the specific cases of F and C. What we call 'behavior' includes the description of four cellular processes: cell death, displacement, proliferation, and infiltration. In the second step, we describe the behavior of all the cells.

## 2.1 Cell death rules

C and F cells have a limited lifespan which varies from cell to cell. When they are alive, they will be able to move or duplicate as explained in the following sections. In our algorithm (see section 2.5), when a cell dies, it stays in place for a while and then disappears.

### 2.1.1 F cell death rules

We define for each F cell a probability $p_{dF}$ of dying (*Appendix 1—figure 3*).

### 2.1.2 C cell death rules

For each C cell, we define a 'basal' probability $p_{dC}$ of dying, and an increased probability $p_{dC+}$ of dying when the C cell has many other C cells in its neighborhood. This latter probability is justified as a recent study has shown the existence of CD8[+] T cell-population-intrinsic mechanisms regulating cellular behavior, with induction of apoptosis to avoid an excessive increase in T cell population (*Zenke et al., 2020*). So, we introduce the threshold number σ of neighboring C cells, above which the probability of dying for a C cell is increased from $p_{dC}$ to $p_{dC+}$ .

two cases are distinguished (*Appendix 1—figure 3*):

- Case 1: if the C cell has few C neighbors ($V\left(C\right)\left(s\right) < σ$, where σ is an unknown integer), then the C cell attempts to die with the probability $p_{dC}$
- Case 2: if the C cell has many C neighbors C ($V\left(C\right)\left(s\right) \geq σ$), then the cell C attempts to die with the probability $p_{dC+}$

The numerical values of $p_{dC}$, $p_{dC+}$, and $p_{dF}$ will be presented and justified in section 3.1.

## F cell death

**No condition on the neighborhood**

**F cell die, with the probability $p_{dF}$**

## C cell death

**Few C neighbours ($V(C)(s) < \sigma$)**

**C cell die, with the probability $p_{dC}$**

**Many C neighbours ($V(C)(s) \geq \sigma$)**

**C cell die, with the probability $p_{dC+}$**

**Appendix 1—figure 3.** Cell death rules. σ has to be taken equal to three neighbors. F and C cells are indicated by, respectively, green and pink squares. Adapted from *Dupin et al., 2023*.

## 2.2 Cell proliferation rules

Cells have the ability to duplicate, as explained in the following sections. In our algorithm (see section 2.5), when a cell divides, it gives birth to two daughter cells, with one staying at the place of the mother cell, and the other one being created in an empty site in the neighborhood of the mother cell.

## 2.2.1 F cell proliferation rules

We define for each F cell a probability $p_F$ of dividing.

## 2.2.2 C cell proliferation rules

For each C cell, we define a 'basal' probability $p_C$ of dividing and an increased probability $p_{C/F}$ of dividing when the C cell has F cell(s) in its neighborhood. This latter probability is justified as our results and those from another study (*Afroj et al., 2021*) show a robust and high increase of C cell proliferation in direct co-cultures of F and C cells.

Consider a C cell located in $s$. To reflect contact inhibition that enables cells to stop proliferating when many of them are in contact with each other, we also introduced a threshold number $\lambda$.

Two cases are distinguished (*Appendix 1—figure 4*):

Case 1: if all the sites of $M(s)$ are occupied (Case 1 a, *i.e.* $V(s) = |M(s)|$), or if all empty s' sites belonging to $M(s)$ have 'many' C neighbors (Case 1b, *i.e.* $V(C)(s') \geq \lambda$, where $\lambda$ will be taken equal to 3) (Case 1b), the C cell does not divide.

Case 2: there is at least one empty site s' belonging to $M(s)$ and such as $V(C)(s') < \lambda$. If there is no F cell in $M(s)$, then the C cell attempts to divide with the probability $p_C$ (case 2 a). If there is at least one F cell in $M(s)$ ($V(F)(s') \geq 1$), the C cell attempts to divide with the probability $p_{C/F}$ (case 2b).

If the C cell divide, the C cell remains in $s$ and we uniformly choose an unoccupied site s' belonging to $M(s)$, such that $V(C)(s') < \lambda$, on which a new C cell is created.

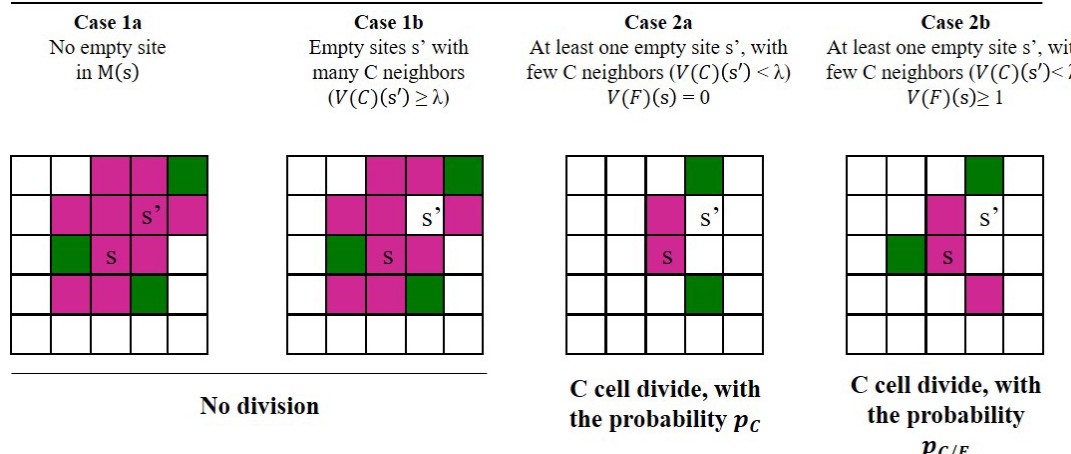

**Appendix 1—figure 4.** The different cases for C cell proliferation rules. $\lambda$ has to be taken equal to three neighbors. F and C cells are indicated by, respectively, green and pink squares. Adapted from *Dupin et al., 2023*.

## 2.3 Cell displacement rules

C and F cells are able to move, as shown previously (*Dupin et al., 2016*; *Mrass et al., 2017*). This process is taken into account in the model, as described below.

Let $s = (i, j)$ and $s' = (i', j')$ be two sites of the lamina propria. We define for each F cell (or C cell) a probability $P_F(s, s')$ (respectively, $P_C(s, s')$) of going from s to s'. A living cell can only move to a site adjacent to the site it occupies:

$$P_F\left(s, s'\right) = P_C\left(s, s'\right) = 0 \text{ if } s' \notin M(s) \cup \{s\}, \text{ or if } s' \in M(s) \text{ and is occupied.}$$

### 2.3.1 F cell displacement rules

Our image analysis on lung tissue indicates that (i) the minimum median distance between two F cells is relatively high (30.6 µm for control subjects, 21.5 µm for patients with COPD) and (ii) F cells are rarely observed in clusters of two or more F cells, suggesting that F cells do not attract each other. We will assume that an F cell is not attracted by another F cell. Then, we applied $f_F$ to the variable $V(C)(s')$, the number of C cells located in the Moore's neighborhood of s'.

A living F cell, located in s, will move to the site $s' \in M(s) \cup \{s\}$, with the probability:

$$P_F(s, s') = \begin{cases} k_F f_F\left(V(C)(s')\right), & \text{if } s' \in M(s), \ s' \text{ is empty and } s' \neq s \\ k_F x_F, & \text{if } s' = s \end{cases}$$

Where:

1. $k_F$ is a constant such that $P_F(s, .)$ is a probability:

$$k_F = \frac{1}{x_F + \sum\limits_{s' \in M(s)} f_F\left(V(C)(s')\right) 1_{\{s' \text{ empty}\}}}$$

2. $x_F > 0$ will be determined later (see section 3.3.3)
3. $f_F$ is a function which will be defined in section 3.3.1

$$f_F(n) = \begin{cases} 1, & \text{for } n \text{ to be defined } (see \ section \ 3.3.1) \\ \varepsilon_F, & \text{for } n \text{ to be defined } (see \ section \ 3.3.1) \end{cases}$$

### 2.3.2 C cell displacement rules

Our image analysis on lung tissue indicate that (i) the minimum median distance between two C cells is low (7,9 µm for control subjects, 4,9 µm for patients with COPD) and (ii) C cells are often observed in clusters of two or more C cells, suggesting that C cells can attract each other. On the other hand, C cells express CCR1, CCR2, CCR4, CCR5, CXCR1, and CXCR2 (*Hombrink et al., 2016*), which are receptors of the chemokines CCL2, CCL3, CCL4, CXCL1, and CXCL8 that can be secreted by F cells (*Dupin et al., 2018*), and the minimum median distance between two C cells

is low (10,9 µm for control subjects, 6,8 µm for patients with COPD), indicating that C cells can be attracted by F cells. Therefore, in contrast to $f_F$ which is applied to $V\left(C\right)\left(s'\right)$, we applied $f_C$ to the variable $V\left(s'\right)$, the total number of C and F cells located in the neighborhood $M\left(s'\right)$.

A living C cell, located in $s$, will move to the site $s' \in M(s) \cup \{s\}$, with the probability:

$$P_c(s, s') = \begin{cases} k_C f_C \left(V\left(s'\right)\right), & \text{if } s' \in M\left(s\right), \; s' \text{ is empty and } s' \neq s \\ k_C x_C, & \text{if } s' = s \end{cases}$$

Where:

1. $k_C$ is a constant such that $P_c(s, .)$ is a probability:

$$k_C = \frac{1}{x_C + \sum\limits_{s' \in M(s)} f_C \left(V\left(s'\right)\right) 1_{\{s' \text{ empty}\}}}$$

2. $x_C > 0$ will be determined later see section (see section 3.3.3)
3. $f_C$ is a function defined on $\{1, \ldots, 8\}$ will be found in section 3.3.2:

$$f_C\left(n\right) = \begin{cases} 1, & \text{for } n \text{ to be defined } \left(\text{see section 3.3.2}\right) \\ \varepsilon_C, & \text{for } n \text{ to be defined } \left(\text{see section 3.3.2}\right) \end{cases}$$

## 2.4 Cell infiltration rules

F and C cells can infiltrate the lungs at a stable state, and this process can be amplified during exacerbations. We will add at the beginning of each 3 min period (see section 1.4), one cell F (resp. C) with the probability $p_{istaF}$ (resp. $p_{istaC}$) to take into account the phenomenon of infiltration during the stable state. These probabilities will be determined from biological considerations (see section 3.4). If a cell is recruited, we randomly and uniformly position it among all the empty sites of the lamina propria (*Appendix 1—figure 5*). If there are no empty sites nearby, no cell is added.

### C and F cell infiltration at the stable state

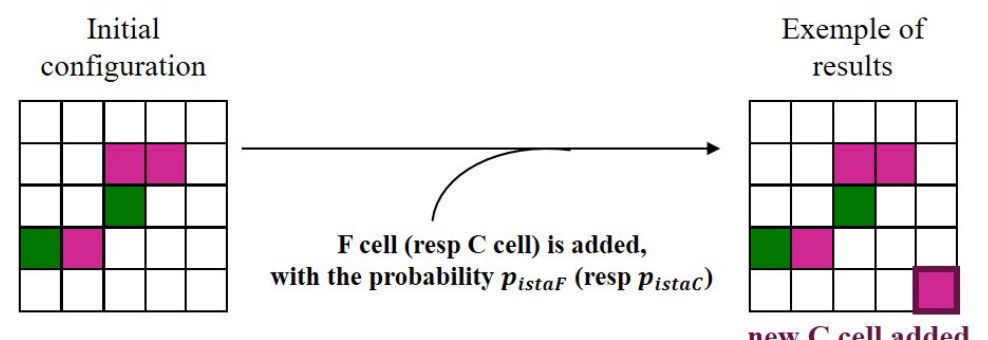

**Appendix 1—figure 5.** C and F cell infiltration rules at the stable state. F and C cells are indicated by, respectively, green and purple squares. Adapted from *Dupin et al., 2023*.

The process of infiltration can be amplified during exacerbations, which is an acute event specific to patients with COPD, and not happening in healthy subjects. To take into account the infiltration during the exacerbations, we will add every year a number $N_{iexaF}$ (resp. $N_{iexaC}$) of F cells (resp. C) with the probability $p_{iexaC}$ (resp. $p_{iexaF}$). If cells are added, they are placed uniformly on the empty sites of the lamina propria.

Concerning $p_{iexaF}$, it depends closely on the condition of the subject (healthy *versus* COPD). For a healthy subject, as there is no exacerbation, this probability is zero. The values of $p_{iexaF}$ and $N_{iexaF}$ for a patient with COPD will be specified in section 3.4.1.

## 2.5 Dynamics of C and F cells

The time steps are denoted $k$, where $1 \leq k \leq T$. The dynamics take place by time step (*Appendix 1—figure 6*). Let us consider the beginning of the time step $k + 1$. They are $N_k\left(F\right)$ F cells and $N_k\left(C\right)$ C cells, then $N_k = N_k\left(F\right) + N_k\left(C\right)$ is the total number of cells in the lamina propria at the beginning of the time step $k$.

If a C or F cell is added by infiltration at the stable state (see section 2.4), it is positioned randomly and uniformly across all vacant sites. This implicitly assumes that all the sites are not occupied. Otherwise, no cell is added. We agree that the added cell is not part of the $N_k$ initial cells, and cannot be drawn at random afterward. It is, therefore, neither subject to death, nor to duplication, nor to displacement during the time step $k + 1$, it is just considered as present.

We divide the time step $k + 1$ into $N_k$ sub-time steps. For each sub-time step, we randomly draw one cell among the $N_k$ cells (with the probability $1/N_k$). Several cases can occur.

1. If the selected cell is dead or if it is a C cell that gave birth to a new cell by division in a previous sub-time step, nothing happens.
2. Assuming in the following that the selected cell is alive and is not a 'mother' cell, we note $(i, j)$ the site occupied by this cell.
   a. An F cell attempts to die with the probability $p_{dF}$ ,
   b. A C cell attempts to die with the probability $p_{dC}$ or $p_{dC+}$ according to the rules defined in section 2.1.2.
3. We hypothesize that in the following the randomly selected cell did not die.
   a. An F cell moves according to the procedure described in section 2.3.1.
   b. A C cell divides according to the procedure described in section 2.2.2. If the C cell divides, we uniformly choose an empty site $s'$ belonging to $M(i, j)$ , such that $V(C)(s') < \lambda$ , on which a new C cell is created. The new C cell is not part of the population of $N_k$ cells, it will be added once the $N_k$ sub-time steps have been completed. If the C cell does not divide, it moves according to the rules described in the section 2.3.2.

When the $N_k$ sub-time steps have been repeated independently, we add to the initial population cells that are either born by proliferation or recruited by infiltration. Dead cells are removed. The number of cells is then $N_{k+1}$ . We start a new cycle of $N_{k+1}$ sub-time steps, as previously described. Therefore, over a time step, a given cell will on average die, move or divide (if it is a C cell).

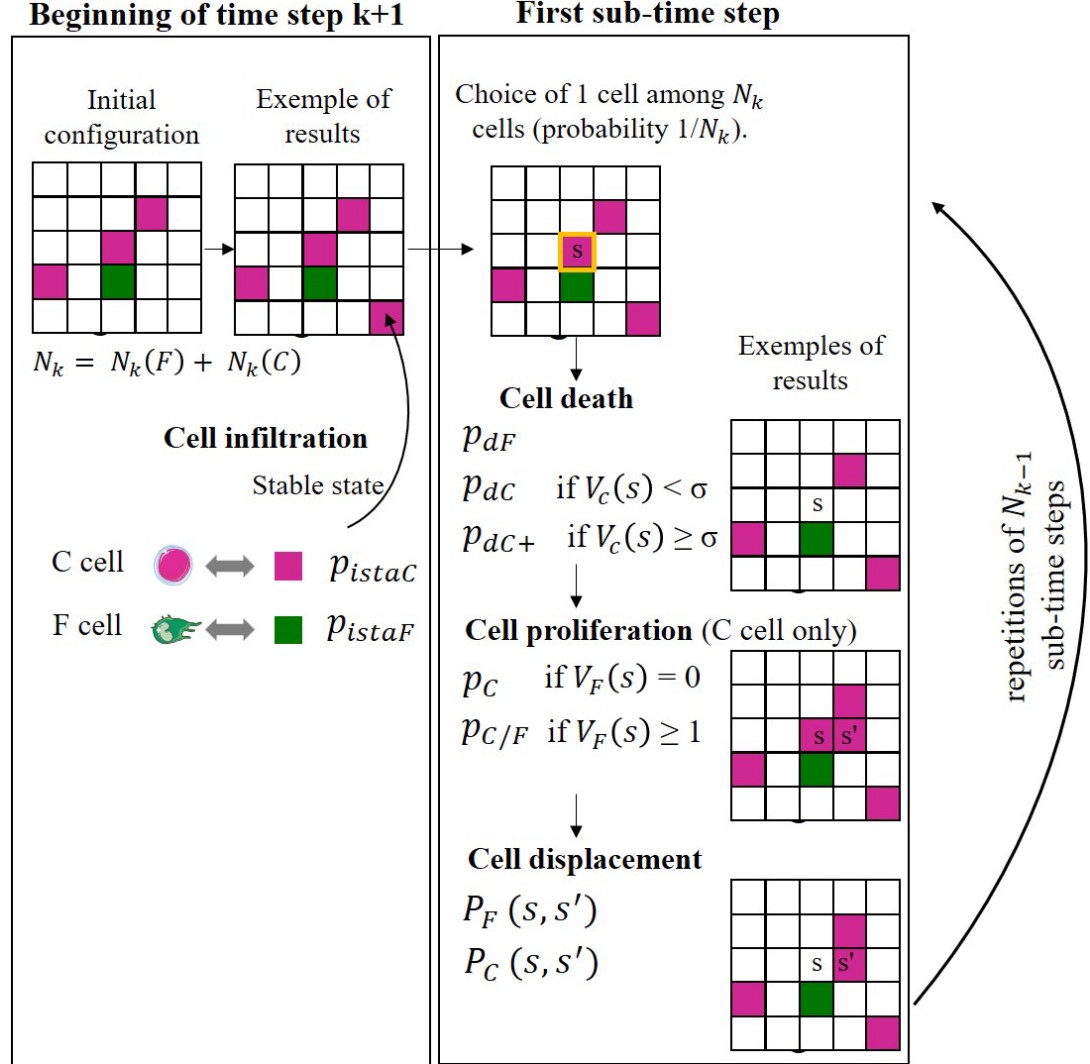

**Appendix 1—figure 6.** Examples of cell infiltration, death, proliferation, and displacement of cells at each time step. We consider the beginning of the time step $k + 1$. This period is divided into $N_k$ sub-time steps, where $N_k$ is the number of cells at the beginning of period $k + 1$. F and C cells are indicated by, respectively, green and purple squares. Adapted from *Dupin et al., 2023*.

## Infiltration during exacerbations

Every year, i.e., after 175,200 time steps, we add a number $N_{iexaF}$ of F cells with the probability $p_{iexaF}$. If F cells are recruited by infiltration during an exacerbation, they are randomly and uniformly positioned among all of the vacant sites. Cells that have infiltrated in the meantime are added to the initial population. A new time step starts again as described above.

## 3 Determination of parameters using biological information

For a control subject as for a patient with COPD, the same model is applied but with different parameters. These parameters are estimated thanks to experiments and data from the literature and are defined below (see *Supplementary file 11* for numerical values). Simulations are performed using these parameter values.

For the estimation of $v_0$, $p_{dC}$, $p_{dF}$, $x_C$, $x_F$, $p_{istaC}$ $p_{istaF}$, $p_{iexaF}$ and $N_{iexaF}$, a streamlined model has been used, in which a mathematical and probabilistic analysis was possible (*Dupin et al., 2023*). The major interest of the streamlined model is that it allows a rigorous mathematical analysis to accurately calculate parameters. We note $\delta^{Ctl}$ and $\delta^{COPD}$ the value of the parameter $\delta$ for, respectively, a control subject and a patient with COPD.

The streamlined model is a particular case of the model, where local interactions, *i.e.*, C cell-induced cell death and contact inhibition of C cell proliferation play no role. To reflect these two properties, σ and λ have been fixed, respectively, to 9 and 0. This means that C and F cell displacement, the infiltration at the stable state, and during exacerbations are identical to the complete model.

## 3.1 Determination of the cell death parameters

### 3.1.1 Determination of $p_{dF}$

Due to the inability to easily label living F cells in vivo, there is no quantitative data on the lifespan of these cells. We will, therefore, approximate that F cells, as cells derived from monocytes, have a lifespan $hl(F)$ similar to those of interstitial macrophages (derived from monocytes) in the lungs. A recent study shows that the half-life of interstitial macrophages derived from monocytes is approximately 10 months in a healthy context (*Schyns et al., 2019*).

For a control subject, we choose the value of the probability $p_{dF}^{Ctl}$ of dying for an F cell over a period of 3 min equal to those calculated in the streamlined model using the $hl^{Ctl}(F)$ lifespan of an F cell in the control case (*Dupin et al., 2023*):

$$p_{dF}^{Ctl} = 1 - e^{-ln2/hl^{Ctl}(F)} \approx 4.8 \times 10^{-6}$$

Our previous work showed that the exposure of fibrocytes to the secretions of the bronchial epithelia of patients with COPD decreases by a factor 2 the percentage of dead cells (*Dupin et al., 2019*). In patients with COPD, we will, therefore, choose for each F cell the probability $p_{dF}^{COPD}$ of dying equal to:

$$p_{dF}^{COPD} = p_{dF}^{Ctl}/2 \approx 2.4 \times 10^{-6}$$

### 3.1.2 Determination of $p_{dC}$

Data obtained in murine lungs show that CD8$^+$ cells have a half-life $hl(C)$ of 14 days in the lung (*McMaster et al., 2015*). For a control subject, we choose the value of the probability $p_{dC}^{Ctl}$ of dying for an C cell over a period of 3 min equal to those calculated in the streamlined model using the $hl^{Ctl}(C)$ lifespan of a C cell in the control case (*Dupin et al., 2023*):

$$p_{dF}^{Ctl} = 1 - e^{-ln2/hl^{Ctl}(C)} \approx 1.0 \times 10^{-4}$$

A study suggests a modification of the C cell death processes in tissues from patients with COPD, with a decrease by about half of the percentage of apoptotic C cells in the distal airways of mild to moderate COPD patients (*Siena et al., 2011*), which constitute the majority of our study cohort. In patients with COPD, we will, therefore, choose for each C cell the probability $p_{dC}^{COPD}$ of dying equal to:

$$p_{dF}^{COPD} = p_{dF}^{Ctl}/2 \approx 5 \times 10^{-5}$$

### 3.1.3 Determination of $p_{dC+}$ and σ

Our image analysis on lung tissues showed that clusters of C cells contain a median value of approximately 4 cells (*Figure 1*). Thus, we choose σ=3 neighbors, to favor cell death for a neighborhood comprising three or more C neighbors.

A recent study demonstrated the existence of a negative feedback loop acting by CTLA-4, to limit the expansion of activated lymphocytes (*Zenke et al., 2020*). Cross-linking of CTLA-4 to the surface of lymphocytes induces apoptosis of previously activated lymphocytes, going from less than 5% of apoptotic lymphocytes to about 90% (*Gribben et al., 1995*). These quantifications were refined in another study, which shows that the cross-linking of CTLA-4 induces an increase in the number of apoptotic lymphocytes of about 4 times (*Scheipers and Reiser, 1998*). We, therefore, choose:

$$p_{dC+} = 4 \times p_{dC}$$

The increased probability $p_{dC+}$ of dying when a C cell has many other C cells in its neighborhood is equal to four times the probability $p_{dC}$, regardless of the condition. We thus obtain:

$$p_{dC+}^{Ctl} = p_{dC}^{Ctl} \times 4 \approx 4.0 \times 10^{-4}$$

$$p_{dC+}^{COPD} = p_{dC}^{COPD} \times 4 \approx 2.0 \times 10^{-4}$$

Our analysis on lung tissues showed that clusters of C cells contain a median value of approximately four cells, in patients with COPD as in control subjects (*Figure 1*), suggesting that σ is unchanged in patients with COPD versus control subjects:

$$\sigma^{Ctl} = \sigma^{COPD} = 3$$

## 3.2 Determination of the cell proliferation parameters

### 3.2.1 Determination of $p_F$

Based on our own unpublished observations and published studies (*Ling et al., 2019*; *Schmidt et al., 2003*), fibrocytes very poorly proliferate in culture, allowing us to consider that an F cell does not divide in lung tissue. We will consider that the probability $p_F$ of dividing for an F cell over a period of 3 min is identical in control subjects and COPD patients and equal to:

$$p_F^{Ctl} = p_F^{COPD} = 0$$

### 3.2.2 Determination of $p_C$

The cell cycle length of a C cell is highly variable, and it is difficult to obtain quantitative data at a steady state. During the 3–6 weeks following an immunization, lung C cells undergo 1–2 cycles of proliferation (*Bivas-Benita et al., 2013*). We will then consider an average duration of 6 weeks=6 × 7 × 24 × 60=60,480 min= 20,160 × 3 min for a cell cycle. We obtain the probability $p_C$ that a C cell divide during the 3 min-time step, for control subjects as well as for patients with COPD:

$$p_C^{Ctl} = p_C^{COPD} = 1/20\,160 \approx 5.0 \times 10^{-5}$$

### 3.2.3 Determination of $p_{C/F}$

Our in vitro experiments show that the presence of fibrocytes multiplies the rate of C cells, which have been divided, by approximately 4, if we consider, respectively, C cells previously activated by beads coated with anti-CD3 and anti-CD28 antibodies. The increased probability $p_{C/F}$ of dividing will, therefore, be taken equal to four times the value of the probability $p_C$. This probability is identical in control subjects and patients with COPD:

$$p_{C/F}^{Ctl} = p_{C/F}^{COPD} = 1/5040 \approx 2.0 \times 10^{-4}$$

## 3.3 Determination of cell displacement parameters

### 3.3.1 Determination of $f_F$

Our chemotaxis experiments show that F cells are almost not attracted towards the secretion of C cells purified from control subjects (*Figure 2*). This justifies an almost zero attraction ($\varepsilon_F$, « small », arbitrarily chosen as $\varepsilon_F = 10^{-3}$).

For control subjects, we have thus chosen for $f_F^{Ctl}(n)$ (*Appendix 1—figure 7*):

$$f_F^{Ctl}(n) = \varepsilon_F, \quad if\, n \in \{0,\, 1,\, 2,\, 3,\, 4\,5,\, 6,\, 7,\, 8\}$$

In the COPD situation, our experiments also show that F cells are significantly attracted toward the secretion of C cells obtained from patients with COPD (*Figure 2*). As this chemotactic effect requires soluble factors that have to be secreted in a sufficient concentrations, this justifies an almost zero attraction for s' such as $V(C)(s') < 4$ cells and a maximal and constant attraction for s' such as $V(C)(s') = 4$ or 5 cells. On the other hand, the attraction of the site s' for an F cell probably decreases when the site is too 'crowded,' because of physical hindrance and/or the secretion of factors that are secreted when many C cells are aggregated. This is in agreement with the theory of quorum sensing (*Antonioli et al., 2018*) and with our observations that the median number of cells in clusters containing F and C cells is relatively low ( for patients with COPD). This leads us to choose an almost zero attraction for s' such as $V(C)(s') > 5$ cells. Thus, a F cell will preferentially and uniformly go to an empty s' site such that $V(C)(s') = 4$ or 5, or will stay on the site s.

For patients with COPD, we have thus chosen for $f_F^{COPD}(n)$(*Appendix 1—figure 7*):

$$f_F^{COPD}(n) = \begin{cases} 1, & if\ n = 4\ or\ 5 \\ \varepsilon_F & if\ n \in \{0,\ 1,\ 2,\ 3,\ 6,\ 7,\ 8\} \end{cases}$$

## F cell displacement

### Examples (Control situation)

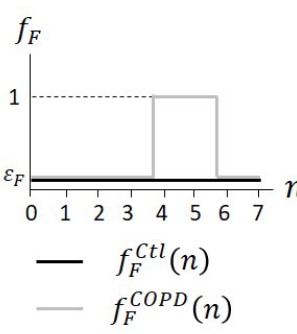 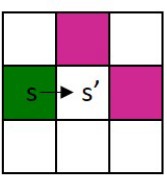 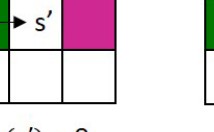 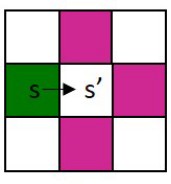 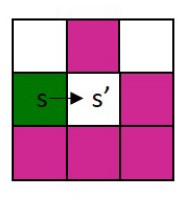

$V_c(s') = 2$

F cell move to s',
with the probability
$P_F(s,s') = k_F\,\varepsilon_F$

$V_c(s') = 3$

F cell move to s',
with the probability
$P_F(s,s') = k_F\,\varepsilon_F$

$V_c(s') = 5$

F cell move to s',
with the probability
$P_F(s,s') = k_F\,\varepsilon_F$

### Examples (COPD situation)

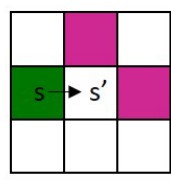 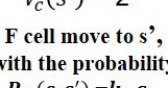 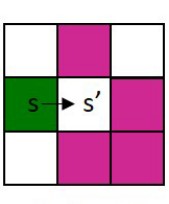 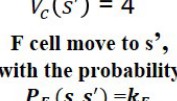 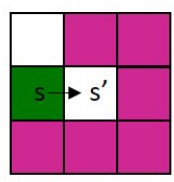

$V_c(s') = 2$

F cell move to s',
with the probability
$P_F(s,s') = k_F\,\varepsilon_F$

$V_c(s') = 4$

F cell move to s',
with the probability
$P_F(s,s') = k_F$

$V_c(s') = 6$

F cell move to s',
with the probability
$P_F(s,s') = k_F\,\varepsilon_F$

**Appendix 1—figure 7.** F Cell displacement rules. F and C cells are indicated by, respectively, green and pink squares. $\varepsilon_F$ has been taken equal to $10^{-3}$.

### 3.3.2 Determination of $f_C$

For control subjects, we have thus chosen for $f_C^{Ctl}(n)$:

$$f_C^{Ctl}(n) = \begin{cases} 1 & if\ n = 4\ or\ 5 \\ \varepsilon_C & if\ n \in \{1,\ 2,\ 3,\ 6,\ 7,\ 8\} \end{cases}$$

with $\varepsilon_C = 10^{-3}$

Based on the same type of justifications than those used for $f_F$, the choice of $f_C$ reflects an almost zero attraction for s' such as $V(s') <4$ cells or $V(s') >5$ cells and a maximal and constant attraction for s' such as $V(s') = 4$ or 5 cells. As $V(s')$ includes the C cell considered at the site s and is, therefore, $\geq 1$, $f_C$ is defined for n $\in \{1,\ 2,\ 3,\ 4,\ 5,\ 6,\ 7,\ 8\}$.

We will consider that $f_C$ is identical in control subjects and COPD patients (**Appendix 1—figure 8**):

$$f_C^{Ctl}(n) = f_C^{COPD}(n)$$

## C cell displacement

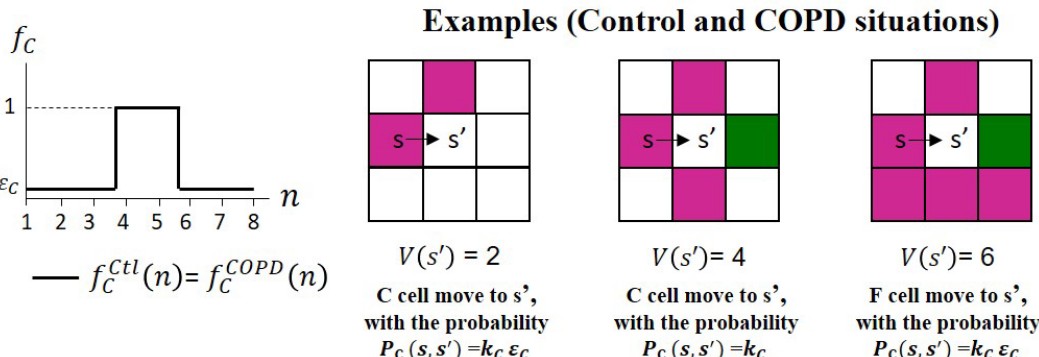

**Appendix 1—figure 8.** C Cell displacement rules. F and C cells are indicated by, respectively, green and pink squares. $\varepsilon_F$ has been taken equal to $10^{-3}$. Adapted from *Dupin et al., 2023*.

### 3.3.3 Determination of $x_F$ and $x_C$

The parameters $x_F$ and $x_C$ are terms involved in the definition of, respectively, $P_F\left(s, s'\right)$ and $P_C\left(s, s'\right)$. They are chosen such as the median speed of F and C cells is equal to 2.3 μm/min. We have previously demonstrated using the streamlined model (*Dupin et al., 2023*) that:

$$k_F = \frac{\alpha/5}{\sum\limits_{s'\in M(s)} f_F\left(V\left(C\right)\left(s'\right)\right) 1_{\{s'\,empty\}}},$$

with $\alpha$ being a real number, whose numerical value is $\alpha = 4.67091$.
This implies that:

$$k_F = \frac{\alpha/5}{\sum\limits_{s'\in M(s)} f_F\left(V\left(C\right)\left(s'\right)\right) 1_{\{s'\,empty\}}}$$

For the C cells, the real numbers $x_C$ and $k_C$ are obtained similarly:

$$x_C = \frac{1-\alpha/5}{\alpha/5}\left(\sum f_C\left(V\left(s'\right)\right) 1_{\{s'\,empty\}}\right), k_C = \frac{\alpha/5}{\sum\limits_{s'\in M(s)} f_C\left(V\left(s'\right)\right) 1_{\{s'\,empty\}}}$$

These parameters are identical for control subjects and patients with COPD.

### 3.4 Determination of cell infiltration parameters

#### 3.4.1 Determination of $p_{istaF}$, $p_{iexaF}$, and $N_{iexaF}$

F cells have a limited lifespan in the lungs, with a half-life that we have estimated at 10 months in the lung, by analogy with the half-life of interstitial macrophages (*Schyns et al., 2019*). Our previous work indicates the presence of F cells in the lungs, at varying densities in control subjects and COPD patients (*Dupin et al., 2019*), suggesting infiltration of F cells at a stable state, in order to maintain this pulmonary pool relatively constant. This leads us to add, for a healthy subject as for a COPD subject, F cells with a probability $p_{istaF}$.

We have demonstrated in the streamlined model (*Dupin et al., 2023*) that the expectation of $N_k\left(F\right)$ is close to $N_0\left(F\right)$ for large $k$ is realized if $p_{istaF} = p_{dF}\,N_0\left(F\right)$. We kept this relation in the full model for the control situation. With $N_0\left(F\right) = 19$, and $p_{dF}^{Ctl} \approx 4.8 \times 10^{-6}$, we obtain for a control subject:

$$p_{istaF}^{Ctl} \approx 9.12 \times 10^{-5}$$

The level of circulating fibrocytes is identical in control subjects and patients with COPD at the stable state (*Dupin et al., 2016*), suggesting that $p_{istaF}$ is identical in control subjects and COPD patients:

$$p_{istaF}^{COPD} = p_{istaF}^{Ctl} \approx 9.12 \times 10^{-5}$$

The choice of $p_{istaC}$ is different, see section 3.4.2.

The value of $p_{iexaF}$ depends closely on the condition of the subject (healthy *versus* COPD). For a healthy subject, as there is no exacerbation, this probability is zero:

$$p_{iexaF}^{Ctl} = 0$$

For COPD patients, this value is non-zero. Indeed, we previously showed that there is an increase in the concentration of F cell levels in the peripheral blood of COPD patients during acute exacerbation (*Dupin et al., 2016*). In addition, in the lungs, the density of F cells is higher in the tissues of COPD patients than in those of healthy subjects (*Dupin et al., 2019*). Since F cells proliferate little or not at all (our own unpublished observations and published studies *Ling et al., 2019*; *Schmidt et al., 2003*), this suggests that for COPD patients, F cells are recruited from the blood to the lungs at the time of exacerbations. To take this into account, for COPD patients, we will add a number $N_{iexaF}$ of F cells in the surface of interest with the probability $p_{iexaF}$ at a frequency corresponding to the average exacerbation frequency of COPD patients, so that after 20 years, on average, the number of F cells is double in COPD patients than in healthy patients. The average number of exacerbations is 0.85, 1.34, and 2 /year in patients with stages 2, 3, and 4 COPD, respectively (*Hurst et al., 2010*). In our cohort for tissue analysis, the majority of patients are stage 2 or 3, so we choose an average number of exacerbations of 1 per year. We will achieve the goal of doubling F cells after 20 years by adding enough F cells each year to compensate for cell death. We agree that the first addition takes place in year 0 and the last is done in the nineteenth year. We do the same assumption as for infiltration at stable state: if we add $N_{iexaF}$ at the start of a year, these cells are not active immediately, it is necessary to wait for the start of the following time step for them to be.

The goal is to choose $p_{iexaF}^{COPD}$ in such a way that the expectation of $N_T(F)$ is the double of $N_0(F)$, where T=20 years=3,504,000 time steps. Let us remind the important fact that the F cells die independently of the environment and do not duplicate themselves. Therefore, we can realize the condition of doubling the number of F cells after 20 years using the streamlined model. We then get (*Dupin et al., 2023*):

$$N_{iexaF} = \left[\frac{K_1}{K_2}\right] + 1 \text{ and } p_{iexaF} = \frac{1}{N_{iexaF}} \frac{K_1}{K_2}$$

With:

$$K_1 = \left(1 - \left(1 - p_{dF}^{COPD}\right)^{a1}\right) \left(2N_0(F) - \left(1 - p_{dF}^{COPD}\right)^{T} \left(N_0(F) - \frac{p_{istaF}^{COPD}}{p_{dF}^{COPD}}\right) - \frac{p_{istaF}^{COPD}}{p_{dF}^{COPD}}\right)$$

$$K_2 = \left(1 - p_{dF}^{COPD}\right)^{a1-1} \left(1 - \left(1 - p_{dF}^{COPD}\right)^{T}\right)$$

And

$$a1 = 1 \text{ year} = 175\,200 \text{ time steps of 3 min}$$

With the values of the previously determined parameters we obtain:

$$N_{iexaF}^{COPD} = 1,\ p_{iexaF}^{COPD} \approx 0.0022$$

### 3.4.2 Détermination of $p_{istaC}$, $p_{iexaC}$ and $N_{iexaC}$

Since C cells have a limited lifespan in the airways, with an estimated half-life of 14 days in the lung (*McMaster et al., 2015*), it has been proposed that the number of memory C cells in the lung tissue is maintained through continuous recruitment either from the systemic pool of cells (*Ely et al., 2006*) or from niches in the pulmonary interstitium (*Takamura et al., 2016*), rather than by

proliferation or prolonged survival. This leads us to add C cells by infiltration in such a way that the number of C cells fluctuates little (at least at the equilibrium). The estimation of the probability of infiltration of C cells at steady state $p_{istaC}$ is described below. We first consider the control situation.

Recall that in the streamlined model (2) if we choose $p_{istaC} = p_{dC} \times N_0(C)$ then the mean number of cells C remains constant at equilibrium. According to the values of $p_{dC}^{Ctl}$, we have $p_{dC}^{Ctl} N_0(C) \approx 1.18 \times 10^{-2}$

We realize $K$ independent simulations of our model up to the time step $T + 1$. We are interested in the $N_T$ sub-steps which follow the first $T$ steps.

For each simulation $1 \leq i \leq K$, $N_{T,i}$ stands for the number of cells at time $T$.

For any $0 \leq l \leq N_{T,i}$, let us introduce:

$\alpha_{l,T,i}^-$ = the number of $C_j-$ cells, computed at the end of the sub-step l, which are alive, did not duplicate and $V(C_j) < \sigma$,

$\alpha_{l,T,i}^+$ = the number of $C_j -$ cells, computed at the end of the sub-step l, which are alive, did not duplicate and $V(C_j) \geq \sigma$,

$\beta_{l,T,i}$ = the number of $C_j -$ cells, computed at the end of the sub-step l, which are alive, did not duplicate, there exists $s' \in M(C_j)$, $s'$ empty, and $V(C)(s') < \lambda$ and there exists no $F$ in $M(C_j)$,

$\beta_{l,T,i}^+$ = the number of $C_j -$ cells, computed at the end of the sub-step l, which are alive, did not duplicate, there exists $s' \in M(C_j)$, $s'$ empty, and $V(C)(s') < \lambda$ and there exists at least one $F \in M(C_j)$.
and

$$A_T^+ = \frac{1}{K}\left\{\sum_{i=1}^{K}\frac{1}{N_{T,i}}\left(\sum_{l=0}^{N_{T,i}-1}\alpha_{l,T,i}^+\right)\right\} \quad A_T^- = \frac{1}{K}\left\{\sum_{i=1}^{K}\frac{1}{N_{T,i}}\left(\sum_{l=0}^{N_{T,i}-1}\alpha_{l,T,i}^-\right)\right\}$$

$$B_T = \frac{1}{K}\left\{\sum_{i=1}^{K}\frac{1}{N_{T,i}}\left(\sum_{l=0}^{N_{T,i}-1}\beta_{l,T,i}\right)\right\} \quad B_T^+ = \frac{1}{K}\left\{\sum_{i=1}^{K}\frac{1}{N_{T,i}}\left(\sum_{l=0}^{N_{T,i}-1}\beta_{l,T,i}^+\right)\right\}$$

We agree that sub-step 0 is at the end of the period $T$.

Under these conditions, we can prove:

$$E(N_{k+1}(C)) \approx E(N_k(C)) + p_{istaC} + p_C B_T + p_{C/F} B_T^+ - p_{dC+} A_T^+ - p_{dC} A_T^-. \tag{1}$$

For any period $k$, where $E(N_k(C)$ and $E(N_{k+1}(C))$ is the mean number of C cells at the time step k (resp. k+1).

## Remark

The coefficients $A_T^+$, $A_T^-$, $B_T$ and $B_T^+$ can be interpreted as follows: during the $N_T$ sub-steps following the period $T$,

- $A_T^+$ (resp. $A_T^-$) is the average number of times C cells can potentially die with probability $p_{dC+}$ (resp. $p_{dC}$).
- $B_T$ (resp. $B_T^+$) is the average number of times C cells can potentially proliferate with probability $p_C$ (resp. $p_{C/F}$).

All the parameters are given by experimental data, the only one which is free is $p_{istaC}$. However, the relation (1) does not permit to determine $p_{istaC}$, because $B_T$, $B_T^+$, $A_T^+$ and $A_T^-$ also depend on $p_{istaC}$. Our strategy is the following: we fix $T$ and we want to determine $p_{istaC} = p_{istaC}(T)$ such that

$$E(N_{T+1}(C)) = E(N_T(C)) \tag{2}$$

From (1) this condition is equivalent to:

$$p_{istaC}(T) = p_{dC+} \times A_T^+ + p_{dC} \times A_T^- - p_C \times B_T - p_{C/F} \times B_T^+. \tag{3}$$

We read (3) as an equation of the form:

$$p_{istaC}(T) = \phi(p_{istaC}(T)) \tag{4}$$

where $\phi$ is a function and we solve it by iteration. We begin with $p_{istaC}^0(T)$ as the value of $p_{istaC}^{ctrl}$ in the streamlined model (*Dupin et al., 2023*), $p_{istaC}^0(T) = 1.18 \times 10^{-2}$.

We simulate $K$ times the complete model over the time period $[0, T+1]$ and we get:

$$p_{istaC}^1(T) = \phi\left(p_{istaC}^0(T)\right). \tag{5}$$

We repeat this process, then we define by induction a sequence $\left(p_{istaC}^k(T)\right)_k$ as

$$p_{istaC}^{k+1}(T) = \phi\left(p_{istaC}^k(T)\right). \tag{6}$$

We stop the iterations as soon as $k \mapsto p_{istaC}^k$ 'seems' constant. Let $p_{istaC}^{ctrl}$ be this value.

Beware that the condition (2) does not imply that $E\left(N_{T'}(C)\right) = E\left(N_T(C)\right)$ for $T' > T$. We verify by simulations that

$$E\left(N_{T'}(C)\right) \approx E\left(N_T(C)\right), \quad for\ any\ \ T \leq T' \leq T_{20y} \tag{7}$$

where $T_{20y} = 20\ years\ \ = 3\ 504\ 000$ time steps.

As $T_{20y}$ represents a large number of time steps we will aggregate them month by month. Let $T_m$ be the number of periods to obtain one month:

$$T_m = \frac{T_{20y}}{240} = 146\ 000\ \ periods.$$

We simulate $K$ times our model (with parameter $p_{istaC}^{ctrl}$ determined as explained above) up to time $T_{20y}$. First, we consider the stochastic fluctuations in the number of $C$ cells. For any simulation $1 \leq i \leq K$ and month $0 \leq j < 240$, let

$$\bar{N}_{jT_m,i}(C) = \frac{1}{T_m}\left(\sum_{l=0}^{T_m-1} N_{jT_m+l,i}(C)\right)$$

where $N_{k,i}(C)$ is the number of $C$ cells at the end of the $k$ time steps, and for the simulation $i$.

It is also interesting to compare the family of curves $\left(\left(\bar{N}_{jT_m,i}(C)\right)_{0 \leq j < 240}, 1 \leq i \leq K\right)$ with the variations of the mean values $\left(\bar{N}_{jT_m}(C)\right)_{0 \leq j < 240}$ where:

$$\bar{N}_{jT_m}(C) = \frac{1}{K}\left(\sum_{i=1}^K \bar{N}_{jT_m,i}(C\ \text{to}\ \bar{N}_T(C))\right), \quad 0 \leq j < 240.$$

We observed that numerically $\bar{N}_T(C)$ is close to $\bar{N}_{T_{20y}}(C)$.

Of course, it remains to choose a value for T. In all the simulations, we found numerically that the stationary state was reached rather quickly, by following the fluctuations of the numbers of cells C and F. We have chosen T=2 years.

Finally, we obtain the following estimation for $p_{istaC}^{Ctl}$:

$$p_{istaC}^{Ctl} = p_{dC}^{Ctl}\, n_0(C) \approx 1.40 \times 10^{-2}$$

There is no biological evidence of difference in the level of circulating C cells between healthy subjects and patients with COPD, suggesting that $p_{istaC}$ is identical in control subjects and COPD patients:

$$p_{istaC}^{Ctl} = p_{istaC}^{COPD} \approx 1.40 \times 10^{-2}$$

Concerning $p_{iexaC}$, the literature shows that there is probably an infiltration of C cells in the lungs, especially in COPD patients (*Freeman et al., 2007*; *Saetta et al., 1999*), but the relationships with exacerbations are not entirely clear. For simplification, we will assume that there is no C cell

infiltration during exacerbations. Thus, for healthy subjects as well as for patients with COPD the value of $p_{iexaC}$ is zero:

$$p_{iexaC}^{Ctl} = p_{iexaC}^{COPD} = 0$$

### 3.5 Determination of parameters upon therapeutic interventions

### 3.5.1 Determination of parameters upon CXCR1/2 inhibition

Our results show reparixin (dual CXCR1/CXCR2 antagonist) treatment completely suppresses the increased chemotaxis induced by the secretions of CD8⁺ T cells purified from COPD lungs (*Figure 2E*). We will thus choose:

$$f_F^{CXCR1/2\ inhibition}\ (n) = f_F^{Ctl}\ (n) = \varepsilon_F\ ,\ if\ n\ \in\ \{0,\ 1,\ 2,\ 3,\ 4\ 5,\ 6,\ 7,\ 8\}$$

All the other parameters remain unchanged and similar to the COPD situation.

### 3.5.2 Determination of parameters upon CD86 or CD54 inhibition

Our results show the treatment with CD54 and CD86-blocking antibodies in the co-culture assay significantly reduced the fibrocyte-induced proliferation of CD8⁺ T cells by a factor, respectively, 1.5 and 1.2 (*Figure 4*).

We will thus choose:

$$p_{C/F}^{CD54\ inhibition} =\ p_{C/F}^{COPD}/1,5 = 1/(5040 \times 1.5) =\approx 1.3 \times 10^{-4}$$

$$p_{C/F}^{CD86\ inhibition} =\ p_{C/F}^{COPD}/1,2 = 1/(5040 \times 1.2) =\approx 1.6 \times 10^{-4}$$

All the other parameters remain unchanged and similar to the COPD situation.

## 4 Simulation procedure

Our algorithm (see section 2.6 and *Appendix 1—figure 6*) is implemented in Julia, which allows to parallelize the computation sequences as well as to use graphic libraries, in order to produce drawings and videos illustrating step by step the evolution of a starting situation according to the biological parameters selected to launch the simulations. Our program is modular, in the sense that it is made up of reusable functions for the benefit of users wishing to test other configurations involving different evolutionary laws. A complete version of the program can be downloaded from the following site: https://plmbox.math.cnrs.fr/d/49bcbc1db63a4654be7e/. It is also included as *Source code 1*.

## 5 Definition of measurements at the final state

### 5.1 Quantification of the density of C cells in interaction with F cells

For each F cell in a site $(i,j) \in P$, the number of C cells belonging to the set of $(i-1,j)$, $(i,j-1)$, $(i,j+1)$, $(i+1,j)$ is counted. The density of C cells in interaction with F cells corresponds to the sum of these numbers of neighboring C-cells divided by the area of the lamina propria (0.179 mm²).

### 5.2 Quantification of the minimal distances between C cells and F cells

For each F cell in a site $= (i,j) \in P$, we set $M^{(0)}\ (s) = M\ (s)$ and for any integer k ≥ 1:

$$M^{(k)}\ (s) = \left\{ (i',j') \in P,\ max\ |i\ -\ i'|,\ |j-j'|\ \right\}\ \leq\ k$$

Obviously $M^{(0)}\ (s)\ \subset\ M^{(1)}\ (s)\ \subset \ldots \subset\ M^{(k)}\ (s)$. $M^{(k+1)}\ (s)$ is obtained by adding another layer of lattice sites around $M^{(k)}\ (s)$.

Consider a C cell located in s'. We define as the minimal distance between the considered F cell and the C cell the number: $k\ \times$ 7m if s $\in M^{(k)}\ (s)$ and s $\notin\ M^{(k+1)}\ (s)$

The same process is repeated until a C cell is founded. For each F, there is a minimal distance associated. For each simulation at the final state, the mean value of the minimal distances can be calculated, and the frequency distributions of minimal distances can also be determined.

## 5.3 Quantification of the clusters

Clusters of cells are defined as a number of cells greater or equal to 2. Cells belong to the same cluster if they are in contact with a corner or a side. Clusters are automatically recorded at each time step and at the final state of the simulations. The distinction between clusters containing exclusively C cells, F cells, or both cell types ('mixed' clusters) is also done. The number of cells in each cluster is also recorded. The mean number of cells per cluster is the average of the number of cells for all the clusters detected at the considered time step. The density of clusters corresponds to the number of clusters divided by the area of the lamina propria ($0.179 \text{ mm}^2$).

