## [Editor Report · eLife assessment]

The manuscript by Eyraud and colleagues examines the role of interactions between fibrocytes and CD8 cells as drivers of disease progression in COPD (chronic obstructive pulmonary disease). The findings that there exist bidirectional interactions between CD8 cells and fibrocytes are supported by **solid** evidence that combines histology of clinical lung samples, in vitro studies obtained from circulating blood fibrocytes and CD8 cells, as well as a computational model that predicts how bidirectional interactions could promote disease progression over the course of 20 years. The study, which is based on patient samples, thus provides **fundamental** insights on COPD progression.

---

## [Referee Report · Reviewer #1 (Public Review)]

The present study combines quantitative histomorphometry, live cell imaging and tracking, functional analyses, and computational modeling to define potentially pathologic interactions between lung CD8 T cells and fibrocytes in human COPD. The authors use multiple technical approaches to establish the close proximity of CD8 T cells with fibrocytes in peri-bronchial tissue in COPD subjects that notably correlate with functional disease parameters (FEV1/FEV). Their follow-on studies identify specific chemokine pathways and inflammatory consequences of these interactions. Collectively, these seminal data acquired in a unified experimental context, provide support for pathogenic interactions between lung CD8 T cells and fibrocytes and now offer the consideration of mediators and pathways that may be amenable to therapeutic targeting. The strength of the study is the integration of the multi-modality approach, the quality of the quantitative data, and the creation of a tenable model for the interaction role in COPD of CD8 T cells and fibrocytes. While both have been previously implicated in COPD, this new study is more definitive by using this integrated approach.

---

## [Referee Report · Reviewer #2 (Public Review)]

The authors use a series of elegant methods to describe the nature of the interrelationship among CD8+ T cells and fibrocytes in the airways of COPD patients. They find an increased presence of these interactions in COPD and show that CXCL8-CXCR2 interactions are crucial for this interaction, leading to increased CD8+ T cell proliferation.

Major strengths of the work include the detailed functional experiments used to describe the nature of the CD8+ T cell - fibrocyte interaction. Another key strength is the translational approach of the work, building on clinical data and connecting back to these same clinical data. The conclusions of the authors are supported by the data. The impact of the work is significant and key to our understanding of the interrelationship between inflammation and tissue remodeling in COPD. Understanding this relationship holds strong potential for the identification of new drug targets and for the identification of patients at risk.

---

## [Referee Report · Reviewer #3 (Public Review)]

Eyraud and colleagues examine how fibrocytes and CD8 cells can interact with each other to promote COPD. The key findings include that CD8 cells and fibrocytes are found to exist in close proximity to each other in COPD lungs using histopathological analysis of patient samples. The authors leverage pre-existing transcriptomic data on CD8 cells to focus on chemokine release by CD8 cells as a potential pathogenic mechanism by which they could affect fibrocyte migration. In vitro studies using peripheral blood-derived CD8 cells and fibrocytes confirm increased fibrocyte migration in the presence of CD8 cells. as drivers of COPD progression. Conversely, in vitro studies show that fibrocytes exert a pro-proliferative effect on CD8 cells. The authors also use a computational model to assess how these interactions could promote the development of fibrocyte-CD8 clusters as COPD progresses over the course of 20 years.

The strengths of the study include:

1. The multi-faceted research approach that integrates histopathology from clinical COPD lung sections, in vitro co-culture studies, and computational modeling.

2. Applying computational modeling to determine how cell-cell interactions of migration and proliferation can result in distribution patterns within the lung that approximate what is found in actual clinical samples

3. Propose a feedback loop of CD8 cells and fibrocytes that could become a potential therapeutic target to interrupt a vicious cycle that promotes COPD

However, there are also some weaknesses:

1. Specificity of the role of CD8 cells: While much of the focus is on the proximity of and interactions between CD8 cells and fibrocytes, it is not clear whether other cells similarly interact with fibrocytes. For example, CD4 cells, dendritic cells, or interstitial macrophages may similarly interact with fibrocytes as several of these also release chemokines. In the absence of a more comprehensive assessment, it becomes difficult to parse out how specific and relevant the fibrocyte-CD8 cell interactions are for COPD progression when compared to other putative interactions.

2. The transcriptomic analysis which in many ways sets the stage for the chemokine studies uses a pre-existing dataset of COPD and non-COPD samples with only n=2. The robustness of such a sample size is limited and the narrow focus on chemokines or adhesion receptors of CD8 cells in this limited sample size does not provide a more comprehensive analysis that would require larger samples sizes, studying the transcriptomes of other cell types and a broader analysis of which pathways are the most likely to be dysregulated in the cells that surround fibrocytes.

3. Specificity of the findings for COPD: The in vitro studies use circulating cells which are different from lung cells and this is appropriately acknowledged by the authors. However, it appears from the description that the cells are all from COPD patients. It is, therefore, not clear whether these interactions between fibrocytes and CD8 cells are unique to COPD, whether they also occur between control CD8 and fibrocytes, or only in cells obtained from patients with inflammatory/pulmonary diseases. This limitation is appropriately acknowledged in the manuscript.

---

## [Author Response]

The following is the authors’ response to the original reviews.

We thank the reviewers for their positive and constructive evaluations. Based upon the reviewers’ helpful comments, we have performed complementary experiments. In particular, we additionally show that:

a complete analysis of CXCR1/2 binding chemokines in the secretions of tissular CD8+ T cells reinforces the key role of CXCL8 in CD8+ T cell-induced fibrocyte chemotaxis (new panel D in Figure 2)a direct contact between fibrocytes and CD8+ T cells triggers CD8+ T cell cytotoxicity against primary basal bronchial epithelial cells (new Figure 6)the interaction between CD8+ T cells and fibrocytes is bidirectional, with CD8+ T cells triggering the development of fibrocyte immune properties (new Figure 7)the characteristic time to reach a stationary state reminiscent of a resolution of the COPD condition was estimated to be about 2.5 years using the simulations. Interfering with chemotaxis and adhesion processes by inhibiting CXCR1/2 and CD54, respectively, was not sufficient to reverse the COPD condition, as predicted by the mathematical model (new Figure 9)the massive proliferation effect induced by fibrocytes is specific to CD8+ T cells and not CD4+ T cells (new Figure 3-figure supplement 2), and that fibrocytes moderately promote the death of unactivated CD8+ T cells in direct co-culture (new Figure 3-figure supplement 3)

We have graphically summarized our findings (new Figure 10) suggesting the existence of a positive feedback loop playing a role in the vicious cycle that promotes COPD. A new table describing patient characteristics for basal bronchial epithelial cell purification has also been added (new Supplementary File 9), the Supplementary Files 7 and S8 have been up-dated to take into account the new experiments.

The mass spectrometry proteomics data have been deposited to the ProteomeXchange Consortium (http://proteomecentral.proteomexchange.org) via the PRIDE partner repository with the dataset identifier PXD041402.

**Reviewer #1 (Recommendations For The Authors):**
The experimental approaches are all rationally designed and the data clearly presented, with appropriate analyses and sample sizes. I could find no technical or interpretative concerns. The interrelationship between the observational data (histology) with the quantitative live cell imaging and the follow-on functional investigations is especially laudable. The data nicely unifies several years of accumulated data regarding the (separate) participation of CD8 T cells and fibrocytes in COPD.

We thank the reviewer for his/her comments.

I have only minor comments:1. Line 79: The observation that T cells may influence fibrocyte differentiation/function was initially made some years earlier by Abe et al (J Immunol 2001; 7556), and should be cited in addition to the follow-on work of Niedermeyer.

This reference has been added to acknowledge this seminal work.

1. Line 632: Corticosteroids originate from the cortex of the adrenal gland. Budenoside and fluticasone are glucocorticoids, not corticosteroids.

This mistake has been corrected in the discussion of the revised manuscript (see line 802 in the revised manuscript).

1. Given the state of T cell immunotherapies, cytokine/chemokine antagonists, and emerging fibrocyte-targeted drugs, can the authors possibly speculate as to desired pathways to target therapeutically?

Chemokine-receptor based therapies could be used to inhibit fibrocyte recruitment into the lungs, such as CXCR4 blockade. We have very recently shown that using the CXCR4 antagonist, plerixafor, alleviates bronchial obstruction and reduces peri-bronchial fibrocytes density (Dupin et al., 2023). Because CXCR4 expression in human fibrocytes is dependent onmTOR signaling and is inhibited by rapamycin in vitro (Mehrad et al., 2009), alternative strategies consisting of targeting fibrocytes via mTOR have been proposed. This target has proven effective in bronchiolitis obliterans, idiopathic pulmonary fibrosis, and thyroid-associated ophthalmopathy, using rapamycin (Gillen et al., 2013; Mehrad et al., 2009), sirolimus (Manjarres et al., 2023) or an insulin-like growth factor-1 (IGF-I) receptor blocking antibody (Douglas et al., 2020; Smith et al., 2017). Inhibiting mTOR is also expected to have effects on CD8+ T cells, ranging from an immunostimulatory effect by activation of memory CD8+ T-cell formation, to an immunosuppressive effect by inhibition of T cell proliferation (Araki et al., 2010). Last, chemokine-receptor base therapies could also include strategies to inhibit the CD8+-induced fibrocyte chemotaxis, such as dual CXCR1-CXCR2 blockade. We were able to test this latter strategy in our mathematical model, see response to point 6 of reviewer 2.

Immunotherapies directly targeting the interaction between fibrocytes and CD8+ T cells could also be considered, such as CD86 or CD54 blockade. The use of abatacept and belatacept, that interfere with T cell co-stimulation, is effective in patients with rheumatoid arthritis (Pombo-Suarez & Gomez-Reino, 2019) and in kidney-transplant recipients (Vincenti et al., 2016), respectively. Targeting the IGF-I receptor by teprotumumab in the context of thyroid-associated ophthalmopathy also improved disease outcomes, possibly by altering fibrocyte-T cell interactions (Bucala, 2022; Fernando et al., 2021).

We also tested this CD86 and CD54 blocking strategy for COPD treatment by simulations, see response to point 6 of reviewer 2.

However, such therapies should be used with caution as they may favour adverse events such as infections, particularly in the COPD population (Rozelle & Genovese, 2007). Additionally, the fibrocytes-lymphocytes interaction has recently been shown to promote anti-tumoral immunity via the PD1-PDL1 immunological synapse (Afroj et al., 2021; Mitsuhashi et al., 2023). Therefore, care should be taken in the selection of patients to be treated and/or timing of treatment administration with regards to the increased risk of lung cancer in COPD patients.

The discussion section has been altered accordingly.

1. The authors may want to consider mentioning (and citing) recent insight into the immune-mediated fibrosis in thyroid-associated ophthalmopathy

These important publications are now cited in a dedicated paragraph about the possible therapeutical interventions (see answer to point 3, and discussion in the revised manuscript).

**Reviewer #2 (Recommendations For The Authors):**
Specific comments1. The rationale for the selection of chemokines overexpressed by CD8+ T cells in COPD is based on literature data of n=2 patients per group. This is limited and risky. I am less concerned about false positives given the selection of chemokines and the available literature but am worried about the possibility that many chemokines may not have been selected based on insufficient power to do meaningful stats on this comparison. For example, many other CXCR1/2 binding CXCL chemokines exist and these could contribute to the migration effect in Fig 2C as well. Given the currently available single-cell resources it should be possible to extend these observations and to investigate CXCL chemokine expression in COPD CD8 T cells to the benefit of Fig 2A in full detail.

We agree with the reviewer that the rationale for the selection of chemokines of interest could be reinforced by the analysis of supplementary single-cell resources. We used data from the COPD cell atlas (Gene Expression Omnibus GSE136831 (Sauler et al., 2022)) to perform such an analysis of chemokine expression by CD8+ CD103+ and CD8+ CD103- T cells. However, the expression level of all chemokines was globally very low, and was not different between control and COPD patients (see Author response image 1).

**Author response image 1. sa4fig1:** Expression of CXC chemokines in lung CD8+ CD103+ and CD8+ CD103- T cells from patients with COPD (n=18 independent samples) in comparison with healthy control subjects (n=29 independent samples) under resting conditions by Single-Cell RNA sequencing analysis (GEO accession GSE136831). The heatmaps show the normalized expression of genes (horizontal axes) encoding CXC chemokines. PF4=CXCL4, PPBP = CXCL7.

The latter results are in discrepancy with those resulting from transcriptomic analysis of microarray data obtained on purified lung CD8+ CD103+ and CD8+ CD103- T cells, showing a significant level of chemokines expression (Hombrink et al., 2016), and a differential expression of CCL2, CCL26, CXCL2, CXCL8 and CCL3L1 between CD8+ T lymphocytes of control and COPD patients (Figure 2A in the revised manuscript). The reason for these differences is unclear, and could be attributed to biological differences (samples obtained from different patients) or, more likely, to differences in sample processing (cell sorting by flow cytometry for microarray analysis, that could activate minimally CD8+ cells) and/or methodological differences (differences of sensitivity between microarray and scRNA seq).

Nevertheless, microarray data regarding CXCL8 expression are in good agreement with our in vitro experiments, showing an enhanced CXCL8 expression by CD8+ T cells purified from COPD lungs, in comparison with that of control subjects. In addition, the CXCL8 blocking antibody fully abrogates the increase of migration induced by secretion of COPD CD8+ T cells, to the same extent as the blocking of CXCR1/2 by reparixin. This suggests that this supplementary chemotaxis is mainly due to CXCL8 and not other CXCR1/2 binding CXCL chemokines, and correlates CXCL8 measurements to functional experiments. This precision has been now added in the results section of the revised version.

1. Equally, it would strengthen the work if multiplex ELISA assays could be provided on the supernatants used in Fig 2D to provide a more comprehensive view of CXCR1/2 binding chemokines.

In order to have a complete view of CXCR1/2 binding chemokines, we have now performed supplementary ELISA assays to measure the concentrations of CXCL1, 3, 5, 6 and 7, in addition of the measurements of CXCL2 and CXCL8 already presented in the previous version of the manuscript (Figure 2D). Results of these new assays are now presented in the revised version of Figure 2. Concentrations of CXCL1, 3, 5, 6 and 7 were unchanged between the control and COPD conditions.

1. In the functional analyses, I missed information on the activation of the fibrocytes. Equally, the focus on CD8 T cells was mainly on proliferation in the functional work. RNAseq analyses on the cells, comparing CD8 T cells and fibrocytes, alone and in co-culture to each other would help to identify interaction patterns in comprehensive detail. Such an experiment would bolster the significance of the studies by providing impact analysis not only on the T cells beyond proliferation but by expanding on the effect of the interaction on the fibrocyte as well.

Regarding the activation state of fibrocytes, we apologize if this was not clear: in our in vitro co-culture experiments, we chose not to activate the fibrocytes. This setting is in agreement with previous findings, demonstrating an antigen-independent T cell proliferation effect driven by fibrocytes (Nemzek et al., 2013), and it is now explicitly written in the results of the revised manuscript.

Regarding the focus of the functional analyses:

First, we have pushed forward the analysis of the consequences of the interaction beyond CD8+ T cells proliferation. In particular, having shown that fibrocytes promote CD8+ T cells expression of cytotoxic molecules such as granzyme B, we decided to investigate the cytotoxic capacity of CD8+ T cells against primary basal bronchial epithelial cells (see new Supplementary File 9 in the revised manuscript for patient characteristics).

Direct co-culture with fibrocytes increased total and membrane expression of the cytotoxic degranulation marker CD107a, which was only significant in non-activated CD8+ T cells (see new Figure 6A-E in the revised manuscript). A parallel increase of cytotoxicity against primary epithelial cells was observed in the same condition (see new Figure 6F-H in the revised manuscript). This demonstrates that following direct interaction with fibrocytes, CD8+ T cells have the ability to kill target cells such as bronchial epithelial cells. This is now included in the results section of the revised manuscript.

Second, we have now performed proteomic analyses on fibrocytes, alone or in co-culture during 6 days with CD8+ T cells either non-activated or activated (see new Figure 7A in the revised manuscript). Of the top ten pathways that were most significantly activated in co-cultured vs mono-cultured fibrocytes, largest upregulated genes were those of the dendritic cell maturation box, the multiple sclerosis signaling pathway, the neuroinflammation signaling pathway and the macrophage classical signaling pathway, irrespective of the activation state of CD8+ T cells (see new Figure 7B in the revised manuscript). The changes were globally identical in the two conditions of CD8+ T cell activation, with some upregulation more pronounced in the activated condition. They were mostly driven by up-regulation of a core set of Major Histocompatibility Complex class I (HLA-B, C, F) and II (HLA-DMB, DPA1, DPB1, DRA, DRB1, DRB3) molecules, co-simulatory and adhesion molecules (CD40, CD86 and CD54). Another notable proteomic signature was that of increased expression of IFN signaling-mediators IKBE and STAT1, and the IFN-responsive genes GBP2, GBP4 and RNF213. We also observed a strong downregulation of CD14, suggesting fibrocyte differentiation, and an upregulation of the matrix metalloproteinase-9 (MMP9) in the non-activated condition only. Altogether, these changes suggest that the interaction between CD8+ T cells and fibrocytes promotes the development of fibrocyte immune properties, which could subsequently impact the activation of CD4+ T cells activation.

Up-regulated pathways identified in proteomic profile of fibrocytes co-cultured with CD8+ T cells are very consistent with a shift towards a proinflammatory phenotype rather than towards a reparative role. The activation of IFN-γ signaling could be triggered by CD8+ T cell secretion of IFN upon fibrocyte interaction, suggesting the existence of a positive feedback loop (see new Figure 10). Additionally, the priming of fibrocytes by CD8+ T cells could also induce CD4+ T cell activation.

1. I suggest rewording the abstract to capture the main storyline and wording more. The abstract is good, but I see so many novelties in the paper that are not well sold in the abstract, particularly the modelling aspects.

As suggested by the reviewer, we revised the abstract, as shown below and in the revised manuscript. The changes are indicated in red:

Revised abstract:

Bronchi of chronic obstructive pulmonary disease (COPD) are the site of extensive cell infiltration, allowing persistent contacts between resident cells and immune cells. Tissue fibrocytes interaction with CD8+ T cells and its consequences were investigated using a combination of in situ, in vitro experiments and mathematical modeling. We show that fibrocytes and CD8+ T cells are found in vicinity in distal airways and that potential interactions are more frequent in tissues from COPD patients compared to those of control subjects. Increased proximity and clusterization between CD8+ T cells and fibrocytes are associated with altered lung function. Tissular CD8+ T cells from COPD patients promote fibrocyte chemotaxis via the CXCL8-CXCR1/2 axis. Live imaging shows that CD8+ T cells establish short-term interactions with fibrocytes, that trigger CD8+ T cell proliferation in a CD54- and CD86-dependent manner, pro-inflammatory cytokines production, CD8+ T cell cytotoxic activity against bronchial epithelial cells and fibrocyte immunomodulatory properties. We defined a computational model describing these intercellular interactions and calibrated the parameters based on our experimental measurements. We show the model’s ability to reproduce histological ex vivo characteristics, and observe an important contribution of fibrocyte-mediated CD8+ T cell proliferation in COPD development. Using the model to test therapeutic scenarios, we predict a recovery time of several years, and the failure of targeting chemotaxis or interacting processes. Altogether, our study reveals that local interactions between fibrocytes and CD8+ T cells could jeopardize the balance between protective immunity and chronic inflammation in bronchi of COPD patients.

1. The probabilistic model appears to suggest that reduced CD8 T cell death may also explain the increase in the pathology in COPD. Did the authors find that fibrocytes reduce cell death of the CD8 T cells?

Taking advantage of the staining of CD8+ T cells with the death marker Zombie NIR, we have quantified CD8+ T cell death in our co-culture assay. The presence of fibrocytes in the indirect co-culture assay did not affect CD8+ T cell death (see new Figure 3-figure supplement 3A-B in the revised manuscript). In direct co-culture, the death of CD8+ T cells was significantly increased in the non-activated condition but not in the activated condition (see new Figure 3-figure supplement 3C-D in the revised manuscript). Of note, these results are in agreement with a recent study showing the existence of CD8+ T cell-population-intrinsic mechanisms regulating cellular behavior, with induction of apoptosis to avoid an excessive increase in T cell population (Zenke et al., 2020). This is taken into account in our mathematical model by an increased probability p_(dC+) of dying when a CD8+ T cell is surrounded by many other T cells in its neighborhood. It also suggests that the reduced CD8+ T cell death evidenced in tissues from patients with COPD (Siena et al., 2011) might not be due to the specific interplay between fibrocyte and CD8+ T cells, but rather to a global pro-survival environment in COPD lungs.

These new data have been described in the results section.

1. Following the modeling in Figure 6, curiosity came to mind, which is how long it would take for the pathology to disappear if a drug would be applied to the patient. How much should the interactions be reduced and how long would it take to reach clinical benefit? Could such predictions be made? I understand that this may be outside the main message of the manuscript but perhaps this could be included in the discussion.

This is a very interesting question, that we have addressed by performing additional simulations to investigate the outcomes of possible therapeutic interventions. First, we applied a COPD dynamics during 20 years, to generate the COPD state, that provide the basis for treatment implementation. Then, we applied a COPD dynamic during 7 years, that mimics the placebo condition (see new Figure 9A in the revised manuscript, and below), that we compared to a control dynamics (“Total inhibition”), that mimics an ideal treatment able to restore all cellular processes. As expected the populations of fibrocytes and CD8+ T cells, as well as the density of mixed clusters, decreased. These numbers reached levels similar of healthy subjects after approximately 2.5 years, and this time point can, therefore, be considered as the steady state (Figure 9B-E).

Monitoring of the different processes revealed that these effects were mainly due to a reduction in fibrocyte-induced CD8+ T duplication, and a transient or more prolonged increase in basal fibrocyte and CD8+ T death (Figure 9C-D).

Then, three possible realistic treatments were considered (Figure 9A). We tested the effect of directly inhibiting the interaction between fibrocytes and CD8+ T cells by blocking CD54. This was implemented in the model by altering the increased probability of a CD8+ T cell to divide when a fibrocyte is in its neighbourhood, as shown by the co-culture results (Figure 4). We also chose to reflect the effect of a dual CXCR1/2 inhibition by setting the displacement function of fibrocyte similar to that of control dynamics, in agreement with the in vitro experiments (Figure 2E). Blocking CD54 only slightly reduced the density of CD8+ T cells compared to the placebo condition, and had no effect on fibrocyte and mixed cluster densities (Figure 9B). CXCR1/2 inhibition was a little bit more potent on the reduction of CD8+ T cells than CD54 inhibition, and it also significantly decreased the density of mixed clusters (Figure 9B). As expected, this occurred through a reduction of fibrocyte-induced duplication, which was affected more strongly by CXCR1/2 blockage than by CD54 blockage (Figure 9C-E). Combining both therapies (CD54 and CXCR1/2 inhibition) did not strongly major the effects (Figure 9B-E). In all the conditions tested, the size of the fibrocyte population remained unchanged, suggesting that other processes such as fibrocyte death or infiltration should be targeted to expect broader effects.

The results section has been altered accordingly.

Using the simulations, we were also able to estimate the characteristic time to reach a stationary state reminiscent of a resolution of the COPD condition. This time of approximately 2.5 years was totally unpredictable by in vitro experiments, and indicates that a treatment aiming at restoring these cellular processes should be continued during several years to obtain significant changes.

We have also investigated the outcomes of more realistic treatments, modifying specifically processes such as chemotaxis or targeting directly the intercellular interactions. The modification of parameters controlling these processes only slightly affected the final state, suggesting that such treatments may be more effective when used in combination with other drugs e.g., those affecting fibrocyte infiltration and/or death.

The discussion section has been altered accordingly.

**Reviewer #3 (Recommendations For The Authors):**
1. Broader assessment of cell types in the lung: Staining for other cell types such as dendritic cells, CD4 cells, and interstitial macrophages, and comparing their proximity to fibrocytes with that of CD8 cells would better justify the CD8 focus.

We agree with the reviewer that multiple stainings would have better justified the focus on CD8+ T cells. However, it is difficult to distinguish fibrocytes, dendritic cells and interstitial macrophages on the basis of immunohistochemistry, as we and others previously showed (Dupin et al., 2019; Mitsuhashi et al., 2015; Pilling et al., 2009). On the other hand, the study of Afroj et al. indicated the possible interaction between fibrocytes and CD8+ T cells in cancer context, with the induction of CD8+ T cell proliferation (Afroj et al., 2021). This T cell-costimulatory function of fibrocytes and CD8+ T cells was further confirmed in a very recent study, together with the antitumor effects of PD-L1 and VEGF blockade (Mitsuhashi et al., 2023). These data, along with the specific implication on CD8+ T cells in COPD, relying mainly on their abundance in COPD bronchi (O’Shaughnessy et al., 1997), their overactivation state (Roos-Engstrand et al., 2009), their cytotoxic phenotype (Freeman et al., 2010; Wang et al., 2020) and the protection against lung inflammation and emphysema induced by their depletion (Maeno et al., 2007) justified the CD8 focus.

To further justify this focus, we have now performed co-culture between fibrocytes and CD4+ T cells, indicating that the massive fibrocyte-mediated proliferation was specific to CD8+ T cells (see answer to comment 3 below). This is in agreement with the results obtained with the simulations, showing that considering fibrocytes and CD8+ T cells only was sufficient to reproduce the spatial patterns in the bronchi of healthy and COPD patients. Altogether, we think that focusing on the CD8+ T cell-fibrocyte interplay was pertinent in the context of COPD. It does obviously not exclude the possibility of other interactions, that could be the focus of otherstudies.

1. Transcriptomic analysis: Using n=2 and only showing the chemokines as well as selected adhesion receptor data narrows the focus but does not provide broader insights into the interactions. Using a more robust sample size and performing a comprehensive pathway analysis would represent an unbiased analysis to determine the most dysregulated pathways. Importantly, the authors could use a single-cell RNA-seq dataset to broadly assess the transcriptomes of several cell types in the lung (such as the data from Sauler et al, Characterization of the COPD alveolar niche using single-cell RNA sequencing).

This very pertinent suggestion has also been raised by reviewer 2, see our answer to comment 1 of reviewer 2, and below:

We agree with the reviewer that the rationale for the selection of chemokines of interest could be reinforced by the analysis of supplementary single-cell resources. We used data from the COPD cell atlas (Gene Expression Omnibus GSE136831 (Sauler et al., 2022)) to perform such an analysis of chemokine expression by CD8+ CD103+ and CD8+ CD103- T cells. However, the expression level of all chemokines was globally very low, and was not different between control and COPD patients (see Figure scRNAseq, in the answer to comment 1 of reviewer 2).

These latter results are in discrepancy with those resulting from transcriptomic analysis of microarray data obtained on purified lung CD8+ CD103+ and CD8+ CD103- T cells, showing a significant level of chemokines expression (Hombrink et al., 2016), and a differential expression of CCL2, CCL26, CXCL2, CXCL8 and CCL3L1 between CD8+ T lymphocytes of control and COPD patients (Figure 2A in the revised manuscript). The reason for these differences is unclear, and could be attributed to biological differences (samples obtained from different patients) or, more likely, to differences in sample processing (cell sorting by flow cytometry for microarray analysis, that could activate minimally CD8+ cells) and/or methodological differences (differences of sensitivity between microarray and scRNA seq).

Nevertheless, microarray data regarding CXCL8 expression are in good agreement with our in vitro experiments, showing an enhanced CXCL8 expression by CD8+ T cells purified from COPD lungs, in comparison with that of control subjects. In addition, the CXCL8 blocking antibody fully abrogates the increase of migration induced by secretion of COPD CD8+ T cells, to the same extent as the blocking of CXCR1/2 by reparixin. This suggests that this supplementary chemotaxis is mainly due to CXCL8 and not other CXCR1/2 binding CXCL chemokines, and correlates CXCL8 measurements to functional experiments. This precision has been now added in the text of the revised version.

1. Inclusion of control/comparison cell types in co-culture studies would help establish that CD8 cells are more relevant for interactions with fibrocytes than for example CD4 cells.

We have now performed co-cultures between fibrocytes and CD4+ T cells, with the same settings than for CD8+ T cells. The results from these experiments show that fibrocytes did not have any significant effect of CD4+ T cells death, regardless of their activation state (see new Figure 3-figure supplement 2A-C in the revised manuscript, and below). Fibrocytes were able to promote CD4+ T cells proliferation in the activated condition but not in the non-activated condition (see new Figure 3-figure supplement 2A-D in the revised manuscript). Altogether this indicates that although fibrocyte-mediated effect on proliferation is not specific to CD8+ T cells, the amplitude of the effect is much larger on CD8+ T cells than on CD4+ T cells.

These new data have been added in the results section.

1. In vitro analysis of cells from non-COPD patients would also help assess whether the circulating cells from COPD patients have a level of baseline activation which promotes the vicious cycle but may not exist in healthy cells.

Regarding circulating cells, the present study relies on the COBRA cohort (COhort of BRonchial obstruction and Asthma), which includes only asthma and COPD patients and, therefore, does not grant access to healthy subjects’ blood samples (Pretolani et al., 2017). Unfortunately, we have no other ongoing study with healthy subjects that would allow us to retrieve blood for research, and fibrocytes can only be grown from freshly drawn blood samples. We agree with the reviewer that it is a limitation of our study, which is now acknowledged at the end of the discussion section.

References

Afroj, T., Mitsuhashi, A., Ogino, H., Saijo, A., Otsuka, K., Yoneda, H., Tobiume, M., Nguyen, N. T., Goto, H., Koyama, K., Sugimoto, M., Kondoh, O., Nokihara, H., & Nishioka, Y. (2021). Blockade of PD-1/PD-L1 Pathway Enhances the Antigen-Presenting Capacity of Fibrocytes. The Journal of Immunology, 206(6), 1204‑1214. https://doi.org/10.4049/jimmunol.2000909

Araki, K., Youngblood, B., & Ahmed, R. (2010). The role of mTOR in memory CD8+ T-cell differentiation. Immunological reviews, 235(1), 234‑243. https://doi.org/10.1111/j.0105-2896.2010.00898.x

Bucala, R. J. (2022). Targeting fibrocytes in autoimmunity. Proceedings of the National Academy of Sciences, 119(5), e2121739119. https://doi.org/10.1073/pnas.2121739119

Douglas, R. S., Kahaly, G. J., Patel, A., Sile, S., Thompson, E. H. Z., Perdok, R., Fleming, J. C., Fowler, B. T., Marcocci, C., Marinò, M., Antonelli, A., Dailey, R., Harris, G. J., Eckstein, A., Schiffman, J., Tang, R., Nelson, C., Salvi, M., Wester, S., … Smith, T. J. (2020). Teprotumumab for the Treatment of Active Thyroid Eye Disease. The New England Journal of Medicine, 382(4), 341‑352. https://doi.org/10.1056/NEJMoa1910434

Dupin, I., Henrot, P., Maurat, E., Abohalaka, R., Chaigne, S., Hamrani, D. E., Eyraud, E., Prevel, R., Esteves, P., Campagnac, M., Dubreuil, M., Cardouat, G., Bouchet, C., Ousova, O., Dupuy, J.-W., Trian, T., Thumerel, M., Begueret, H., Girodet, P.-O., … Berger, P. (2023). CXCR4 blockade alleviates pulmonary and cardiac outcomes in early COPD (p. 2023.03.10.529743). bioRxiv. https://doi.org/10.1101/2023.03.10.529743

Dupin, I., Thumerel, M., Maurat, E., Coste, F., Eyraud, E., Begueret, H., Trian, T., Montaudon, M., Marthan, R., Girodet, P.-O., & Berger, P. (2019). Fibrocyte accumulation in the airway walls of COPD patients. The European Respiratory Journal, 54(3), Article 3. https://doi.org/10.1183/13993003.02173-2018

Fernando, R., Caldera, O., & Smith, T. J. (2021). Therapeutic IGF-I receptor inhibition alters fibrocyte immune phenotype in thyroid-associated ophthalmopathy. Proceedings of the National Academy of Sciences, 118(52), e2114244118. https://doi.org/10.1073/pnas.2114244118

Freeman, C. M., Han, M. K., Martinez, F. J., Murray, S., Liu, L. X., Chensue, S. W., Polak, T. J., Sonstein, J., Todt, J. C., Ames, T. M., Arenberg, D. A., Meldrum, C. A., Getty, C., McCloskey, L., & Curtis, J. L. (2010). Cytotoxic potential of lung CD8+ T cells increases with COPD severity and with in vitro stimulation by IL-18 or IL-15. Journal of immunology (Baltimore, Md. : 1950), 184(11), 6504‑6513. https://doi.org/10.4049/jimmunol.1000006

Gillen, J. R., Zhao, Y., Harris, D. A., LaPar, D. J., Stone, M. L., Fernandez, L. G., Kron, I. L., & Lau, C. L. (2013). Rapamycin Blocks Fibrocyte Migration and Attenuates Bronchiolitis Obliterans in a Murine Model. The Annals of thoracic surgery, 95(5), 1768‑1775. https://doi.org/10.1016/j.athoracsur.2013.02.021

Hombrink, P., Helbig, C., Backer, R. A., Piet, B., Oja, A. E., Stark, R., Brasser, G., Jongejan, A., Jonkers, R. E., Nota, B., Basak, O., Clevers, H. C., Moerland, P. D., Amsen, D., & van Lier, R. A. W. (2016). Programs for the persistence, vigilance and control of human CD8+ lung-resident memory T cells. Nature Immunology, 17(12), Article 12. https://doi.org/10.1038/ni.3589

Maeno, T., Houghton, A. M., Quintero, P. A., Grumelli, S., Owen, C. A., & Shapiro, S. D. (2007). CD8+ T Cells are required for inflammation and destruction in cigarette smoke-induced emphysema in mice. Journal of Immunology (Baltimore, Md.: 1950), 178(12), 8090‑8096. https://doi.org/10.4049/jimmunol.178.12.8090

Manjarres, D. C. G., Axell-House, D. B., Patel, D. C., Odackal, J., Yu, V., Burdick, M. D., & Mehrad, B. (2023). Sirolimus suppresses circulating fibrocytes in idiopathic pulmonary fibrosis in a randomized controlled crossover trial. JCI Insight. https://doi.org/10.1172/jci.insight.166901

Mehrad, B., Burdick, M. D., & Strieter, R. M. (2009). Fibrocyte CXCR4 regulation as a therapeutic target in pulmonary fibrosis. The International Journal of Biochemistry & Cell Biology, 41(8‑9), 1708‑1718. https://doi.org/10.1016/j.biocel.2009.02.020

Mitsuhashi, A., Goto, H., Saijo, A., Trung, V. T., Aono, Y., Ogino, H., Kuramoto, T., Tabata, S., Uehara, H., Izumi, K., Yoshida, M., Kobayashi, H., Takahashi, H., Gotoh, M., Kakiuchi, S., Hanibuchi, M., Yano, S., Yokomise, H., Sakiyama, S., & Nishioka, Y. (2015). Fibrocyte-like cells mediate acquired resistance to anti-angiogenic therapy with bevacizumab. Nature Communications, 6(1), Article 1. https://doi.org/10.1038/ncomms9792

Mitsuhashi, A., Koyama, K., Ogino, H., Afroj, T., Nguyen, N. T., Yoneda, H., Otsuka, K., Sugimoto, M., Kondoh, O., Nokihara, H., Hanibuchi, M., Takizawa, H., Shinohara, T., & Nishioka, Y. (2023). Identification of fibrocyte cluster in tumors reveals the role in antitumor immunity by PD-L1 blockade. Cell Reports, 112162. https://doi.org/10.1016/j.celrep.2023.112162

Nemzek, J. A., Fry, C., & Moore, B. B. (2013). Adoptive transfer of fibrocytes enhances splenic T-cell numbers and survival in septic peritonitis. Shock (Augusta, Ga.), 40(2), 106‑114. https://doi.org/10.1097/SHK.0b013e31829c3c68

O’Shaughnessy, T. C., Ansari, T. W., Barnes, N. C., & Jeffery, P. K. (1997). Inflammation in bronchial biopsies of subjects with chronic bronchitis : Inverse relationship of CD8+ T lymphocytes with FEV1. American Journal of Respiratory and Critical Care Medicine, 155(3), 852‑857. https://doi.org/10.1164/ajrccm.155.3.9117016

Pilling, D., Fan, T., Huang, D., Kaul, B., & Gomer, R. H. (2009). Identification of markers that distinguish monocyte-derived fibrocytes from monocytes, macrophages, and fibroblasts. PloS One, 4(10), e7475. https://doi.org/10.1371/journal.pone.0007475

Pombo-Suarez, M., & Gomez-Reino, J. J. (2019). Abatacept for the treatment of rheumatoid arthritis. Expert Review of Clinical Immunology, 15(4), 319‑326. https://doi.org/10.1080/1744666X.2019.1579642

Pretolani, M., Soussan, D., Poirier, I., Thabut, G., Aubier, M., COBRA Study Group, & COBRA cohort Study Group. (2017). Clinical and biological characteristics of the French COBRA cohort of adult subjects with asthma. The European Respiratory Journal, 50(2), 1700019. https://doi.org/10.1183/13993003.00019-2017

Roos-Engstrand, E., Ekstrand-Hammarström, B., Pourazar, J., Behndig, A. F., Bucht, A., & Blomberg, A. (2009). Influence of smoking cessation on airway T lymphocyte subsets in COPD. COPD, 6(2), 112‑120. https://doi.org/10.1080/15412550902755358

Rozelle, A. L., & Genovese, M. C. (2007). Efficacy results from pivotal clinical trials with abatacept. Clinical and Experimental Rheumatology, 25(5 Suppl 46), S30-34.

Sauler, M., McDonough, J. E., Adams, T. S., Kothapalli, N., Barnthaler, T., Werder, R. B., Schupp, J. C., Nouws, J., Robertson, M. J., Coarfa, C., Yang, T., Chioccioli, M., Omote, N., Cosme, C., Poli, S., Ayaub, E. A., Chu, S. G., Jensen, K. H., Gomez, J. L., … Rosas, I. O. (2022). Characterization of the COPD alveolar niche using single-cell RNA sequencing. Nature Communications, 13(1), Article 1. https://doi.org/10.1038/s41467-022-28062-9

Siena, L., Gjomarkaj, M., Elliot, J., Pace, E., Bruno, A., Baraldo, S., Saetta, M., Bonsignore, M. R., & James, A. (2011). Reduced apoptosis of CD8+ T-lymphocytes in the airways of smokers with mild/moderate COPD. Respiratory Medicine, 105(10), 1491‑1500. https://doi.org/10.1016/j.rmed.2011.04.014

Smith, T. J., Kahaly, G. J., Ezra, D. G., Fleming, J. C., Dailey, R. A., Tang, R. A., Harris, G. J., Antonelli, A., Salvi, M., Goldberg, R. A., Gigantelli, J. W., Couch, S. M., Shriver, E. M., Hayek, B. R., Hink, E. M., Woodward, R. M., Gabriel, K., Magni, G., & Douglas, R. S. (2017). Teprotumumab for Thyroid-Associated Ophthalmopathy. The New England Journal of Medicine, 376(18), 1748‑1761. https://doi.org/10.1056/NEJMoa1614949

Vincenti, F., Rostaing, L., Grinyo, J., Rice, K., Steinberg, S., Gaite, L., Moal, M.-C., Mondragon-Ramirez, G. A., Kothari, J., Polinsky, M. S., Meier-Kriesche, H.-U., Munier, S., & Larsen, C. P. (2016). Belatacept and Long-Term Outcomes in Kidney Transplantation. The New England Journal of Medicine, 374(4), 333‑343. https://doi.org/10.1056/NEJMoa1506027

Wang, X., Zhang, D., Higham, A., Wolosianka, S., Gai, X., Zhou, L., Petersen, H., Pinto-Plata, V., Divo, M., Silverman, E. K., Celli, B., Singh, D., Sun, Y., & Owen, C. A. (2020). ADAM15 expression is increased in lung CD8+ T cells, macrophages, and bronchial epithelial cells in patients with COPD and is inversely related to airflow obstruction. Respiratory Research, 21(1), 188. https://doi.org/10.1186/s12931-020-01446-5

Zenke, S., Palm, M. M., Braun, J., Gavrilov, A., Meiser, P., Böttcher, J. P., Beyersdorf, N., Ehl, S., Gerard, A., Lämmermann, T., Schumacher, T. N., Beltman, J. B., & Rohr, J. C. (2020). Quorum Regulation via Nested Antagonistic Feedback Circuits Mediated by the Receptors CD28 and CTLA-4 Confers Robustness to T Cell Population Dynamics. Immunity, 52(2), 313-327.e7. https://doi.org/10.1016/j.immuni.2020.01.018